



# Attribution of the Australian bushfire risk to anthropogenic climate change

Geert Jan van Oldenborgh[1], Folmer Krikken[1], Sophie Lewis[2], Nicholas J. Leach[3], Flavio Lehner[4,5], Kate R. Saunders[6], Michiel van Weele[1], Karsten Haustein[7], Sihan Li[7,8], David Wallom[8], Sarah Sparrow[8], Julie Arrighi[9,10], Roop P. Singh[9], Maarten K. van Aalst[9,11,12], Sjoukje Y. Philip[1], Robert Vautard[13], and Friederike E. L. Otto[7]

[1]Royal Netherlands Meteorological Institute (KNMI), De Bilt, Netherlands
[2]University of New South Wales, Canberra, ACT, Australia
[3]Atmospheric, Oceanic and Planetary Physics, Department of Physics, University of Oxford, Oxford, U.K.
[4]Institute for Atmospheric and Climate Science, ETH Zürich, Zürich, Switzerland
[5]Climate and Global Dynamics Laboratory, National Center for Atmospheric Research, Boulder, USA
[6]Delft Institute of Applied Mathematics, Delft University of Technology, Delft, Netherlands
[7]Environmental Change Institute, University of Oxford, Oxford, U.K.
[8]Oxford e-Research Centre, University of Oxford, Oxford, U.K.
[9]Red Cross Red Crescent Climate Centre, The Hague, Netherlands
[10]Global Disaster Preparedness Center, Washington DC, USA
[11]Faculty of Geo-information Science and Earth Observation, University of Twente, Enschede, Netherlands
[12]International Research Institute for Climate and Society, Columbia University, New York, USA
[13]Institut Pierre-Simon Laplace, France

**Correspondence:** G. J. van Oldenborgh (oldenborgh@knmi.nl)

**Abstract.** Disastrous bushfires during the last months of 2019 and January 2020 affected Australia, raising the question to what extent the risk of these fires was exacerbated by anthropogenic climate change. To answer the question for southeastern Australia, where fires were particularly severe, affecting people and ecosystems, we use a physically-based index of fire weather, the Fire Weather Index, long-term observations of heat and drought, and eleven large ensembles of state-of-the-art

5 climate models. In agreement with previous analyses we find that heat extremes have become more likely by at least a factor two due to the long-term warming trend. However, current climate models overestimate variability and tend to underestimate the long-term trend in these extremes, so the true change in the likelihood of extreme heat could be larger. We do not find an attributable trend in either extreme annual drought or the driest month of the fire season September–February. The observations, however, show a weak drying trend in the annual mean. Finally, we find large trends in the Fire Weather Index in the ERA5

10 reanalysis, and a smaller but significant increase by at least 30% in the models. The trend is mainly driven by the increase of temperature extremes and hence also likely underestimated. For the 2019/20 season more than half of the July–December drought was driven by record excursions of the Indian Ocean dipole and Southern Annular Mode. These factors are included in the analysis. The study reveals the complexity of the 2019/20 bushfire event, with some, but not all drivers showing an imprint of anthropogenic climate change.



# 1 Introduction

2019 was the warmest and driest year in Australia since homogeneous temperature and rainfall observations began (in 1910 and 1900), following two already dry years in large parts of the country. These conditions as well as a strong Indian Ocean Dipole from the middle of the year onwards and a large-amplitude excursion of the Southern Annular Mode led to weather conditions conducive to bushfires across the continent and so the annual bushfires were more widespread and intense and started earlier in the season than usual (Bureau of Meteorology, Annual Climate Statement 2019). There was unprecedented bushfire activity across the states of Queensland (QLD), New South Wales (NSW), Victoria (VIC), South Australia (SA), Western Australia (WA) and in the Australian Capital Territory (ACT).

In addition to the unprecedented nature of this event, its impacts to date have been disastrous (Reliefweb Australia: bushfires, 2020). There have been at least 34 fatalities as a direct result of the bushfires and the resulting smoke caused hazardous air quality adversely affected millions of residents in cities in these regions with levels higher than twenty times those considered safe by the Australian government. About 5,900 buildings have been destroyed. There are estimates that between 0.5 and 1.5 billion wild animals lost their lives, along with tens of thousands of livestock. The bushfires are having an economic impact (including millions in insurance claims), as well an immediate and long term health impact to the people exposed to smoke and dealing with the psychological impacts of the fires.

It has at times been difficult for emergency services to protect or evacuate some communities due to the pace at which the bushfires have spread, sometimes forcing residents to flee to beaches and lakes to await rescue. Interruptions of the supply of power, fuel and food supplies have been reported and road closures have been common. This has resulted in total isolation of some communities, or only accessible by air or sea when smoke conditions allow. The long and severe drought during 2019 and the two years before is expected to have a strong negative impact on agriculture (Reliefweb Australia: bushfires, 2020).

It is well-established that wildfire smoke exposure is associated with respiratory morbidity (Reid et al., 2016). Additionally, fine particulate matter in smoke may act as a triggering factor for acute coronary events (such as heart attack-related deaths) as found for previous fires in southeast Australia (Haikerwal et al., 2015). As noted by Johnston and Bowman (2014), increased bushfire-related risks in a warming climate have significant implications for the health sector, including given measurable increases in illness, hospital admissions, and deaths associated with severe smoke events.

Based on the recovery of areas following previous major fires, such as Black Saturday in Victoria in 2009, these impacts are likely to affect people, ecosystems and the region for a substantial period to come.

The satellite image in Fig. 1 shows the severity of the fires since July, with two regions with particularly severe events in the South West and South East. Due to the fact that in the South East many population centres were affected and the region was also strongly affected by drought we focus our analysis on this region.

Wildfires in general are one of the most complex weather-related extreme events (Sanderson and Fisher, 2020) with their occurrence depending on many factors including the weather conditions conducive to fire at the time of the event and also on the availability of fuel, which in turn depends on rainfall, temperature and humidity in the weeks, months and sometimes even years preceding the actual fire event. In addition ignition sources and type of vegetation as factors largely outside the meteorology


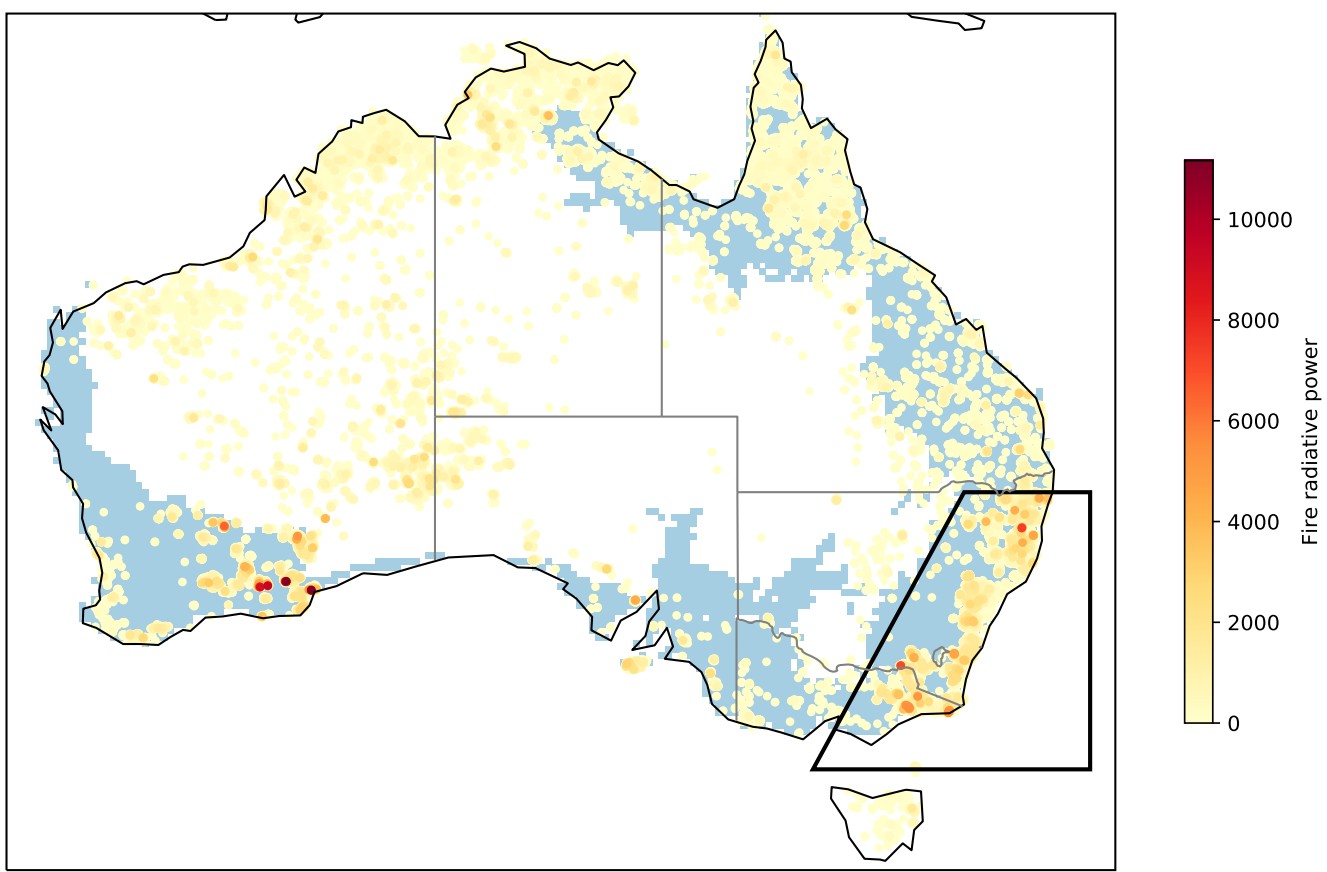

**Figure 1.** Modis active fire data (Collection 6, near real time and standard products) showing the severity of bushfires from 1 July 2019 to 10 January 2020 with the most severe fires being depicted in red. The image also shows the forested areas of Eastern Australia in blue. The polygon shows the area analysed in this article.

play an important role. In this analysis we only consider the influence of weather and climate on the fire risk, excluding ignition
sources. There is not one definition of what fire weather consists of as the relative importance of different factors depends on the climatology of the region. For instance, fires in grasslands in semi-arid regions behave very differently than those in temperate forests. There are a few key meteorological variables that are important: temperature, precipitation, humidity and wind (speed as well as direction). Fire danger indices are derived from these variables either using physical models or empirical relationships between these variables and fire occurrence, including observed factors such as the rate of spread of fires and measurements
of fuel moisture content with different sets of weather conditions. In Australia the Forest Fire Danger Index (FFDI McArthur, 1966, 1967; Noble et al., 1980) is commonly used for indicating dangerous weather conditions for bushfires, including for issuing operational forecasts during the 2019/20 summer. The index is based on temperature, humidity and wind speed on a given day as well as a drought-factor which is based on antecedent temperature and rainfall. Southeast Australia experiences a temperate climate and on the eastern seaboard hot summers are interspersed with intense rainfall events, often linked with





'east-coast lows' (Pepler et al., 2014). Bushfire activity historically commences in the Austral spring (September-November) in the north and summer (December-February) in the south Clarke et al. (2011). Bushfire weather risk, as characterised by the FFDI, has increased across much of Australia in recent decades (Clarke et al., 2013; Dowdy, 2018; Harris and Lucas, 2019). Similar, increasing trends in fire weather conditions over southern Australia have been identified in other studies, both for FFDI (e.g., Dowdy, 2018) and for indices representing pyroconvective processes (Dowdy and Pepler, 2018). These observed trends

over southeast Australia are broadly consistent with the projected impacts of climate change (e.g., Clarke et al., 2011; Dowdy et al., 2019). For individual fire events studies have shown that it can be difficult to separate the influence of anthropogenic climate change from that of natural variability (e.g., Hope et al., 2019; Lewis et al., 2019). An alternative index is the physically based Canadian Fire Weather Index (FWI) that also includes the influence of wind on the fuel availability (Dowdy, 2018). The latter is achieved by modelling fuel moisture on three different depths including the influence of humidity and wind speed on

the upper fuel layer (Krikken et al., 2019). While the FWI was originally developed specifically for the Canadian forests, the physical basis of the models allows it to be used for many different climatic regions of the world (e.g., Camia and Amatulli, 2009; Dimitrakopoulos et al., 2011), and has been shown to provide a good indication of the occurrence of previous extreme fire events in the South Eastern Australian climate (Dowdy et al., 2009). A study on the emergence of fire weather anthropogenic signal from noise indicated that this is expected around 2040 for Southern Australia (Abatzoglou et al., 2019) using the FWI.

In this study we also consider the Monthly Severity Rating (MSR), which is derived from the FWI and reflects better how difficult a fire is to suppress (Shabbar et al., 2011).

As the fire risk indices depend on heat and drought, and these were also extreme in 2019/2020, we also consider these factors separately. Previous attribution studies on Australian extreme heat at regional scales has generally indicated an influence from anthropogenic climate change. The 'angry summer' of 2012/2013—which until 2018/2019 was the hottest summer on record—

was found to be at least 5 times more likely to occur due to human influence (Lewis and Karoly, 2013). The frequency and intensity of heatwaves during this summer were also found to increase (Perkins et al., 2014). Other attribution assessments that found an attributable influence on extreme Australian heat include the May 2014 heatwave (Perkins and Gibson, 2015), the record October heat in 2015 (Hope et al., 2016), and extreme Brisbane heat during November 2014 (King et al., 2015). However, at small spatial scales such as in-situ sites, human influence on extreme heat is less clear (Black et al., 2015). It is

worth noting that Lewis et al. (2019) found that the temperature component of the extreme 2018 Queensland fire weather had an anthropogenic influence, while no clear influence was detected on the February 2017 extreme fire weather over Eastern Australia (Hope et al., 2019). We are not aware of any attribution studies on Australian drought.

Thus, while it is clear that climate change does play an important role in heat and fire weather risk overall, assessing the magnitude of this risk and the interplay with local factors has been difficult. Nevertheless it is crucial to prioritise adaptation

and resilience measures to reduce the potential impacts of rising risks.

We perform the analysis of possible connections between the fire weather risk and anthropogenic climate change in three steps. First, we assess the trends in extreme temperature and conduct an attribution study using a seven-day moving average of annual maximum temperatures corresponding to the time scale chosen for the Fire Weather Index. Second, we undertake the same analysis but for drought defined as purely lack of rainfall in two time windows, the annual precipitation as well as





the driest month in the fire season, with the latter again roughly corresponding to the input time scale of the FWI. Third, we conduct an attribution study on the Fire Weather Index (FWI) and the Monthly Severity Rating (MSR). These three attribution studies follow the same protocol used in previous assessments (Heat waves: Kew et al. (2019); low precipitation: Otto et al. (2018b); Fire Weather Index: Krikken et al. (2019). We continue the analysis with a discussion of other large scale drivers, such as El Niño Southern Oscillation (ENSO), the Indian Ocean Dipole (IOD) or the Southern Annular Mode (SAM). Finally we consider non-climate factors (such as exposure and vulnerability) that have contributed to the extreme fire season of 2019/20.

## 2  Data and methods

### 2.1  Event definition

In this article we trace the connection between anthropogenic climate change and the likelihood and intensity of dangerous bushfire conditions as parametrised by the FWI in the region with the most intense fires in 2019/20 in southeastern Australia. We defined this region as the land area in the polygon 29 °S, 155 °E; 29 °S, 150 °E; 40 °S, 144 °E and 40 °S, 155 °E (as shown in Fig. 1). This corresponds with the area between the Great Dividing Range and the coast.

To capture the variations in the start of the fire season in the region described above we take for most quantities first the annual maximum per grid point over the fire season September–February and next the spatial average over the region defined above. This way the events do not need to be simultaneous at separate grid points within the region. We therefore investigate the question how anthropogenic climate change influences the chances of an intense bushfire season, rather than focusing on a single episode of intense bushfires.

The FWI Index provides a reasonable proxy for the burnt area in the extended summer months, with the strongest relationship observed from November to February. Fig. 2 shows both the Spearman rank based correlation and the Pearson correlation when a log-transform of the burnt area was taken. The 95% confidence intervals are also shown. Given the similarity in the confidence intervals, the log-linear relationship appears to explain equal variability to that of the ranks.

In most years only very small areas are burnt, but the observational record also includes events with extremely large areas. Given this, we checked if the burned area observations were heavy-tailed (Pasquale, 2013). We found that monthly burned area was not Pareto-distributed and instead is reasonably approximated using a log-normal distribution. This supports using the log-transformation and extrapolating this relationship to the 2019/20 fire season. Temporal detrending of the observations did not alter these conclusions.

### 2.2  Observational data

The observational data used in this study are described in Sect. 3.3, 4.2 and 5.3 for heat, drought and the fire weather index respectively. For the Global Mean Surface temperature (GMST) we use GISTEMP surface temperature (Hansen et al., 2010).


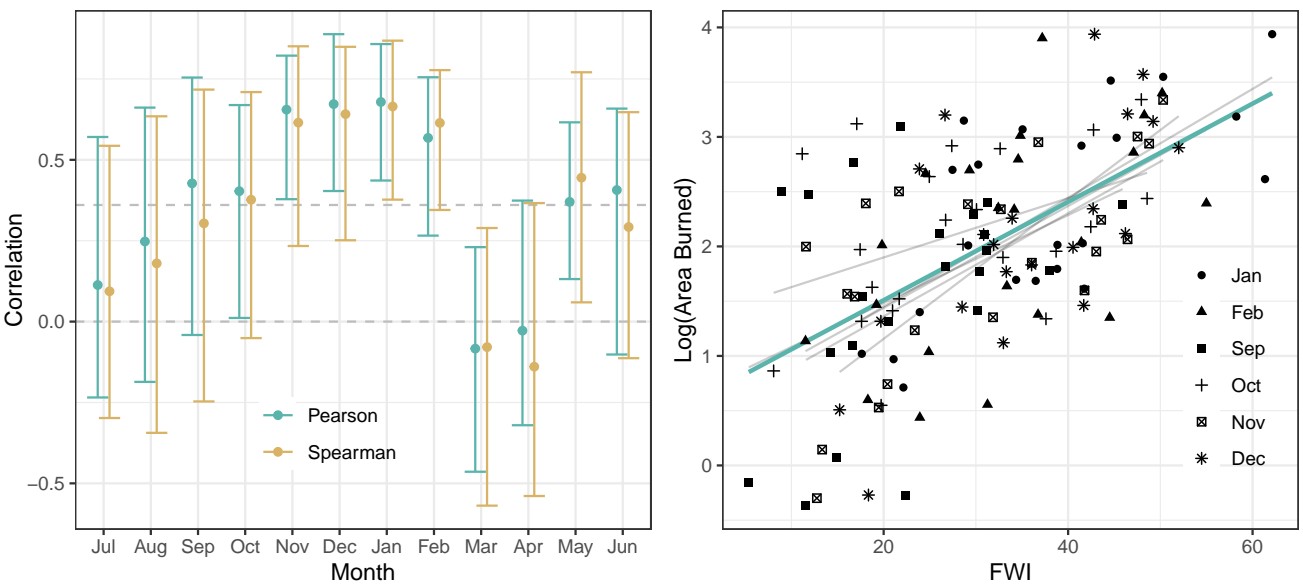

**Figure 2.** Left: correlation and 95% two-sided interval based on the bootstrap of the logarithm of area burnt (MODIS, Collection 6) in the index area as a function of the 7-day maximum Fire Weather Index for each month of the year. The horizontal line denotes the one-sided 95% confidence interval around zero. Right: scatterplot of the values for the fire season, September–February. The correlations are based on the years 1997 to 2018.

## 2.3   Model and experiment descriptions

Attributing observed trends to anthropogenic climate change can only be done with physical climate models as they allow isolating different drivers. For this purpose we use a large set of ocean-atmosphere coupled climate models. A selection of large ensembles of CMIP5 models has been used: CanESM2, CESM1-CAM5, CSIRO Mk3.6.0, EC-Earth 2.3, GFDL CM3, GFDL ESM2M and MPI ESM. In addition, the HadGem3-A N216 attribution model developed in the EUCLEIA project, the weather@home distributed attribution project model and the ASF20C seasonal hindcast ensemble have been used. These

three models are uncoupled and forced with observations of historical Sea Surface Temperature (SST) fields and estimates of SST fields as they might have been in a counterfactual climate without anthropogenic climate change. Finally, we used the coupled IPSL-CM6A-LR low-resolution CMIP6 ensemble. The GFDL-CM3 and MPI-ESM models that did not have daily data were not used for the extreme heat analysis. Given that for the FWI daily data of relative humidity (RH), temperature, precipitation and wind speed are necessary, the list is shortened to CanESM2, CESM1-CAM5, EC-Earth, IPSL-CM6A-LR and

weather@home (HadAM3P) for that part of the analysis.




| name | context | resolution | members | time | reference |
|---|---|---|---|---|---|
| ASF20C | seasonal hindcasts | T255L91 (0.71°) | 51 | 1901–2010 | Weisheimer et al. (2017) |
| CanESM2 | CMIP5 | 2.8° | 50 | 1950–2099 | Kirchmeier-Young et al. (2017) |
| CESM1-CAM5 | CMIP5 | 1° | 40 | 1920–2100 | Kay et al. (2015) |
| CSIRO-Mk3-6-0 | CMIP5 | 1.9° | 30 | 1850–2100 | Jeffrey et al. (2013) |
| EC-Earth | CMIP5 | T159 (1.1°) | 16 | 1860–2100 | Hazeleger et al. (2010) |
| GFDL-CM3 | CMIP5 | 2.0° | 20 | 1920–2100 | Sun et al. (2018) |
| GFDL-ESM2M | CMIP5 | 2.0° | 30 | 1950–2100 | Rodgers et al. (2015) |
| HadGEM3-A | attribution | N216 (0.6°) | 15 | 1960–2015 | Ciavarella et al. (2018) |
| IPSL-CM6A-LR | CMIP6 | 2.5×1.5° | 32 | 1950–2019 | Boucher et al. (2020) |
| MPI-ESM | CMIP5 | 1.9° | 100 | 1850–2099 | Maher et al. (2019) |
| weather@home | attribution | N96 (1.8°) | 1520 × 2 | 1987–2017 | Guillod et al. (2017) |

**Table 1.** List of climate model ensembles used.

## 2.4 Statistical methods

Changes in the frequency of extreme events are calculated by fitting the data to a statistical distribution. In this study the temperature extremes and fire risk-related variables (FWI, MSR) are assumed to follow a GEV distribution, while the low precipitation values are fitted using a Generalized Pareto Distribution (GPD).

The GEV distribution is:

$$P(x) = \exp\left[ -\left(1 + \xi \frac{x-\mu}{\sigma}\right)^{-1/\xi} \right],$$ (1)

where $x$ the variable of interest, e.g., temperature or precipitation, $-\infty < \mu < \infty$, $\sigma > 0$, $-\infty < \xi < \infty$. Here, $\mu$ is the location parameter, $\sigma$ is the scale parameter, and $\xi$ is the shape parameter. The shape parameter determines the tail behaviour: a negative shape parameter gives an upper bound to the distribution. The scale parameter corresponds to the variability in the tail.

The GPD gives a 2-parameter description of the tail of the distribution above a threshold, where the low tail of precipitation is first converted to a high tail by multiplying the variable by $-1$. The GPD is then described by:

$$H(u-x) = 1 - \left(1 - \frac{\xi x}{\sigma}\right)^{(-1/\xi)},$$ (2)

with $x$ the temperature or precipitation, $u$ the threshold, $\sigma$ the scale parameter, and $\xi$ the shape parameter determining the tail behaviour. For the low extremes of precipitation, the fit is constrained to have zero probability below zero precipitation

($\xi < 0, \sigma < u\xi$). Calculations have been done on the lowest 20% and 30% of the data, which gives a first-order estimate of the influence of using more or less extreme events. We cannot use fewer points as the fits do not converge anymore, and using more than 30% does not qualify as the 'lower tail'.





Drought is particularly difficult to model using the existing extreme value framework (Cooley et al., 2018). While minima can be modelled by multiplying by $-1$ (Coles, 2001), the applicability of the underlying extreme value theory assumptions

still needs to be checked. In the case of low precipitation, autocorrelations are a concern. In southeastern Australia, these serial autocorrelations are approximately $r \approx 0.2$, so although non-zero, do not dominate the drought characteristics. Despite these theoretical limitations, in practice the diagnostic plots show the Generalised Pareto models explain the data reasonably well. In general this is a difficult problem, and the statistical extremes community are still developing the solutions necessary for modelling drought events (Naveau et al., 2016).

To calculate a trend in transient data, some parameters in these statistical models are made a function of the 4-yr smoothed global mean surface temperature (GMST) anomaly, $T'$. The covariate-dependent function can be inverted and the distribution evaluated for a given year, e.g., a year in the past (with $T' = T'_0$) or the current year ($T' = T'_1$). This gives the probabilities for an event at least as extreme as the observed one in these two years: $p_0$ and $p_1$, or expressed as return periods $\tau_0 = 1/p_0$ and $\tau_1 = 1/p_1$. The change in probability is called the Probability Ratio (PR): $\mathrm{PR} = p_1/p_0 = \tau_0/\tau_1$.

For temperature we assume that the distribution shifts with GMST: $\mu = \mu_0 + \alpha T'$ or $u = u_0 + \alpha T'$, and $\sigma = \sigma_0$ with $\alpha$ the trend that is fitted together with $\mu_0$ and $\sigma_0$. The shape parameter $\xi$ is assumed constant. For drought and FWI related variables we assume the distribution scales with GMST, the scaling approximation (Tebaldi and Arblaster, 2014). In a GEV fit this gives:

$$\mu = \mu_0 \exp(\alpha T'/\mu_0), \sigma = \sigma_0 \exp(\alpha T'/\mu_0),$$

and in a GPD fit

$$u = u_0 \exp(\alpha T'/u_0), \sigma = \sigma_0 \exp(\alpha T'/\mu_0),$$

with fit parameters $\sigma_0, \alpha$ and $\xi$. The threshold $u_0$ is determined with an iterative procedure and the shape parameter $\xi$ is again assumed constant.

For all fits we also estimate 95% uncertainty ranges using a non-parametric bootstrap procedure, in which 1000 derived time series generated from the original one by selecting random data points with replacement are analysed in exactly the same

way. The 2.5 and 97.5 percentile of the 1000 output parameters (defined as $100i/1001$ with $i$ the rank) are taken as the 95% uncertainty range. For some models with prescribed SSTs or initial conditions the ensemble members are found to not be statistically independent, defined here by a correlation coefficient $r > 1/e$ with $e \approx 2.7182$. In those cases the same procedure is followed except that all dependent time series are entered together in the bootstrapped sample, analogous to the method recommended in Coles (2001) to account for temporal dependencies.

When using a GEV to model tail behaviour, note that taking the spatial average of the annual maxima, does not have the same statistical justification as taking the annual maximum of the spatial average (Coles, 2001). Given this, the impact of the order of operations in the event definition was examined. For heat, we compared the annual time series for the event definition we use, first taking the annual maximum and next the spatial average, to the definition with the order reversed, which can be approximated with a GEV. The Pearson correlation was $r = 0.95$, which is likely due to strong spatial dependence and the

concentration of heatwaves at the peak of the seasonal cycle. Therefore in practice, an approximation with a GEV is not entirely unsuitable for temperature, but caution should be exercised. For the FWI the order of operations makes a difference. Indeed





we find that the whole distribution is not described well by a GEV for most models. In those cases we take block maxima over 5- or 10-ensemble member blocks, effectively looking only at the most extreme events, until the GEV fit agrees with the data points in the return time plot, as expected from taking block maxima.

We evaluate all models on the fitting parameters. For the extremes, we check whether the fit parameters of the distribution from model data agree within uncertainties with those of the observations. We allow for an overall bias correction, additive for temperature, multiplicative for precipitation and Fire Risk variables.

Finally, observations and all models that pass the evaluation test are combined to give a synthesized attribution statement. To obtain a single synthesised attribution statement we first combine the observational results. The spread of the observed estimates stems from the representation uncertainty. This is added to the natural variability as an independent source of uncertainty. In the synthesis figures (Figs. 6, 11, 12 and 16) the solid light blue bars indicate uncertainties due to natural variability, the black outline boxes show natural variability and the dark blue bar represents the consolidated value for observations (reanalyses).

Next, we combine the results from the model-based analysis, which is the main attribution step. We have two estimates of the uncertainty: the uncertainty range expected from the internal natural variability from the individual fits and the spread of the different model results. We check whether these are compatible by computing the $\chi^2/\mathrm{dof}$ statistic. If $\chi^2/\mathrm{dof}$ is greater than one, natural variability alone cannot explain the model spread. We therefore add the model spread to the natural variability, in quadrature as they are independent. In the synthesis figures the model spread is denoted by the white boxes. We next compute a weighted mean by weighing the models by the inverse square of their uncertainties due to natural variabilities, which minimises the uncertainties in the mean. The total uncertainty of the models is shown as a bright red bar in these figures. This total uncertainty consists of a weighted mean using the uncorrelated natural variability plus an independent model spread term added to the uncertainty if $\chi^2/\mathrm{dof} > 1$.

Finally, observations and models are synthesised into a single mean and uncertainty range. This can only be done when they appear to be compatible. We show two combinations. The first one is computed by neglecting model uncertainties beyond the model spread. The optimal combination is then the weighted average of models and observations, shown as a magenta bar. However, the total model uncertainty is unknown and can be larger than the model spread. We therefore also show the more conservative estimate of an unweighted average of observations and models. This is indicated in Figs. 11 and 12 by a white box.

## 3    Extreme heat

### 3.1    The heat of 2019/20

Australia started 2019 during an extreme summer that was the country's hottest on record in terms of both seasonal mean and mean maximum temperatures. Both variables broke the previous records set in the 2012/13 season by almost one degree. The summer mean maximum temperature for the 2018/19 season was 2.61 °C warmer than the 1961–1990 average. However, many of the temperature records set in early 2019 were eclipsed by the extreme heat during December 2019. This was the hottest month on record in terms of national mean and mean maximum temperature anomalies, respectively at 3.21 and 4.15





°C above the 1961–90 December average. The peak of the heat occurred in the week ending the 24th December, which was the country's hottest week on record, at a national mean maximum temperature of 40.5 °C. During this week, the highest national mean maximum temperature was recorded on the 17th at 41.9 °C, 1 °C higher than the previous record, which was set the previous day. In terms of national mean maximum temperatures, eight of the ten hottest days on record occurred in December 2019. While January 2020 was not as extreme as December 2019, it still ranked as the third warmest January on record, with

many individual stations in New South Wales observing their highest January temperature on record on the 4th or 5th of the month (Bureau of Meteorology: Australia in January 2020).

This record summer occurred, at least in part, in Australia's warmest and driest year on record and directly after the current hottest summer on record (2018/2019 was 1.52 °C above the seasonal average). Overall, Australia has warmed by 1 °C since 1910, however, most of this warming has occurred since 1950. The frequency of extreme heat events in Australia outnumber ex-

treme cool events by 12:1 (Lewis and King, 2015), and the frequency of heatwaves have also increased since 1950 (Perkins and Alexander, 2013). Increasing trends in heatwave intensity, frequency and duration are projected throughout the 21st Century (Cowan et al., 2014), with a clear link between global warming thresholds and overall heatwave changes (Perkins-Kirkpatrick and Gibson, 2017).

## 3.2 Temporal Event Definition

For this analysis, we choose an event definition that represents the impacts of extreme heat on the fire risk: the annual (July-June, in order to ensure a continuous summer season) maximum of a 7-day moving average application to daily maximum temperatures, TX7x. Therefore, in this section of the study we aim to answer the question of whether and by how much the probability of a 7-day average maximum temperature at least as high as observed in the study region in 2019 has changed as a result of anthropogenic climate change.

## 240 3.3 Observational temperature data and methods

We use a number of datasets developed using independent methodologies to assess observed daily maximum temperatures. The first is the Berkeley Earth climate analysis (Rohde et al., 2013), a gridded dataset derived statistically from available station data. Although maximum daily temperature data is available from 1880 onwards, here we only use data from 1910 onwards, since the use of Stevenson huts in Australia was only standardised throughout from this time and earlier measurements are likely

biased high by several degrees (Trewin, 2013). Berkeley Earth uses large decorrelations lengths that are more appropriate for annual than daily data. The next is the Australian Water Availability Project (AWAP) analysis 1910–now, which is constructed by imposing anomalies from station data on a high-resolution climatology. This is augmented by a simple average of a set of quality-controlled Australian Climate Observations Reference Network – Surface Air Temperature (ACORN-SAT) stations (Trewin, 2013). These include a large number of coastal stations. The ACORN-SAT daily analysis fields were not yet available

at the time of writing.

We also considered reanalysis data, both long-term reanalyses that are based only on Sea Surface Temperature (SST) and sea-level pressure (SLP), the NOAA Twentieth Century Reanalysis version 3 (20CRv3 Slivinski et al., 2019) and the ECMWF
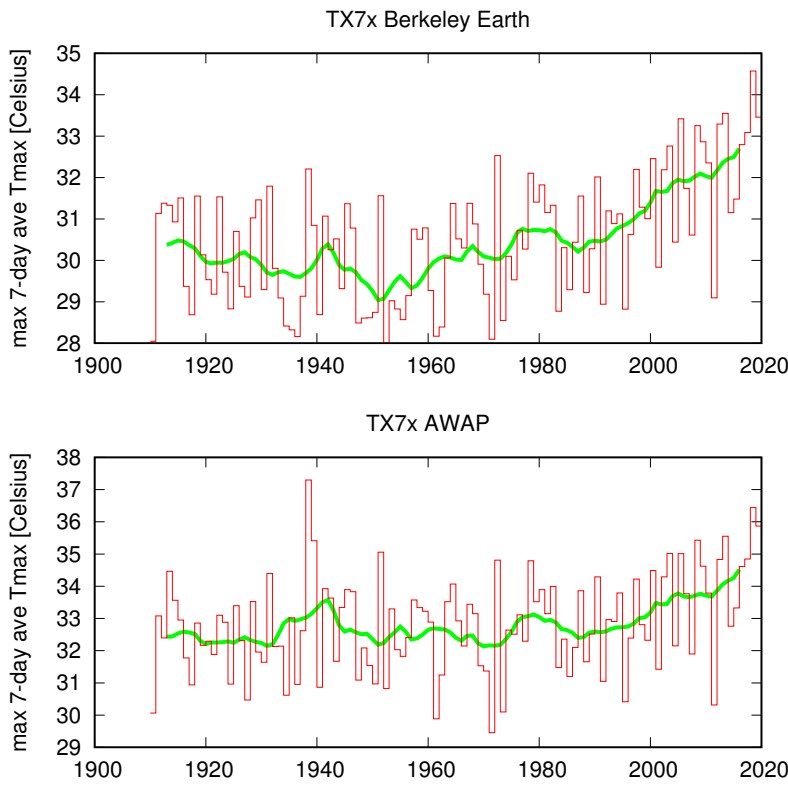

**Figure 3.** The highest 7-day running mean of daily maximum temperature of the July–June year in (top) Berkeley Earth and (bottom) the AWAP analysis. The green line indicates a 10-yr running mean.

Coupled ReAnalysis of the Twentieth Century 1900-2010 (CERA-20C Laloyaux et al., 2018). Finally we used the Japanese ReAnalysis (JRA-55 Kobayashi et al., 2015), a reanalysis product from JMA using 4D Variational data assimilation in their
TL319 global spectral model spanning 1958-2019 at the time of writing.

A comparison of the observational analyses reveals striking differences (Fig. 3). The trend in the Berkeley Earth analysis is higher than in the AWAP analysis and high extremes are suppressed, most notably in 1938/39. We looked into this event on 8–14 January 1939 in more detail and ACORN-SAT station data confirms its reality. It even appears in the 20CRv3 reanalysis, which apparently captures the extraordinary circulation that led to the very high temperatures in southeastern Australia that
week without assimilating near-surface temperatures. We therefore disregard the Berkeley analysis for the heat extremes in the region of this study. The CERA-20C reanalysis also does not capture this event, which is very relevant for the statistical distribution of heat extremes, so we also do not consider it further.

As described in section 2.4, the trend and return period are calculated using the properties of the fit of a Generalized Extreme Value (GEV) distribution, in which the location parameter is a linear model of smoothed Global Mean Surface Temperature
(GMST). As the regional event definition has not been selected on the basis of high temperatures we include the year 2019/20





in the fit when available in the datasets. We note that extreme heat GEV distributions have a negative shape parameter; an upper bound exists for the distribution. Hence if the observed 2019/20 event lies above the upper bound of the distribution in 1900, the probability of the event occurring without the GMST trend is zero, and the increase in likelihood due to global warming is formally infinite, although the 95% uncertainty interval usually has a finite lower bound.

### 3.4  Observational analysis: return time and trend

In the AWAP series the warmest 7-day period so far for the regional index in 2019/20 is 35.9 °C, the third-highest value after 1938/39 and 2018/19. It has a return time of about 8 yr (5 ... 35 yr). For the reanalyses and model results, which have biases, we use a rounded return time of 10 yr. The GEV fit of the AWAP data gives a return time of 85 yr (35 ... $\infty$ yr) in 1900 (see Fig. 4, which implies that the probability has increased by a factor of about 11 (3 ... $\infty$) from 1900 to 2019 in this statistical 275 model. The temperature of TX7x has increased by about 1.7 °C (0.8 ... 2.6 °C) in this period. JRA-55 tells a similar story, with a significant temperature increase of 1.5 (1.3-3.4) °C extrapolated to 1900–2019.

### 3.5  Model evaluation

We consider a set of eight model ensembles that had daily maximum temperatures available to carry out the attribution analysis. To investigate whether the models represent extreme heat well we compare the fit parameters of the tail of the TX7x distribution 280 of the models with those of the observations. In this GEV fit we take the smoothed observed global mean temperature as covariate. The results are shown in Fig. 5. This shows that most models overestimate the scale parameter $\sigma$. This corresponds to the models having too much variability in hot weeks. The same problem was found in the Mediterranean (Kew et al., 2019) and northwestern Europe (Vautard et al., 2020). The only exception is the CESM1-CAM5 model, which has too small a scale parameter. This model also has a shape parameter $\xi$ that is incompatible with the fit to observations, all other models agree 285 with the observations in this parameter.

The discrepancy implies that we cannot give quantitative results for the attribution of heat extremes in southeastern Australia, as the heat extremes in the climate models are too different from the observed heat extremes. This affects especially the change in probability, which depends strongly on the variability. For the trend estimates the influence of this shortcoming is smaller. We continue with all models apart from CESM1-CAM5, keeping these limitations in mind.

### 3.6  Multi-model attribution and synthesis

We computed trends in the models by either comparing the actual climate 1987–2017 to an estimate of a counterfactual climate of the same period with anthropogenic emissions (weather@home) or by fitting a scaled distribution to the transient data in the same way as for the observational estimates, using the observed smoothed GMST (all other models) as covariate. This revealed two outliers: the ASF20C ensemble has a negative trend over the full 1901-2010 period, so we only use data from 295 1960 onwards.

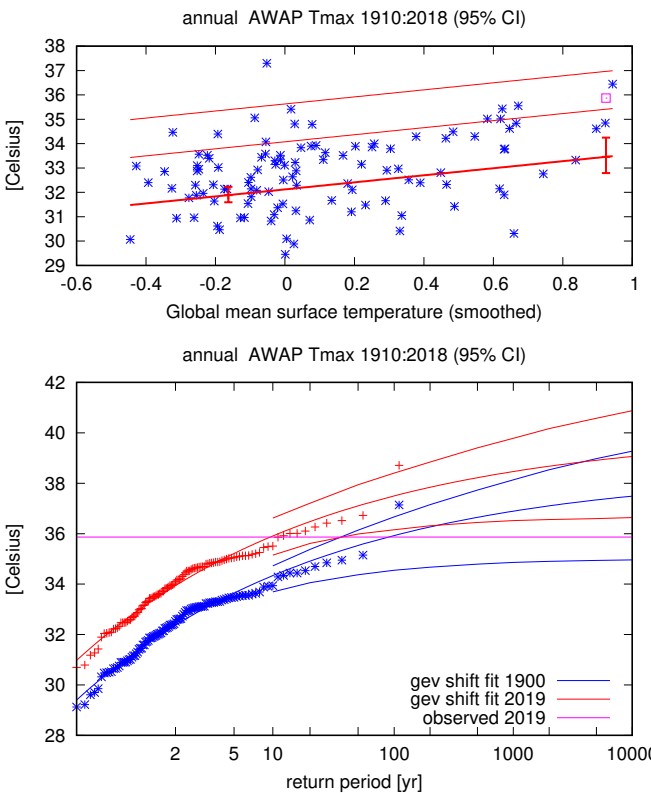

**Figure 4.** GEV fit to the AWAP TX7x averaged over the bushfire region. The position parameter $\mu$ is assumed linearly dependent on the smoothed GMST and the scale and shape parameters constant. Top: observations (blue symbols), location parameter $\mu$ (thick line) and the 6 and 40 yr return values (thin lines). Bottom: return time plot for the climates of 1900 (blue) and 2019 (red), the purple line denotes the 2019/20 event.

Fig. 6 summarises the change in probability and in intensity since 1900 for the 2019 event (observations) and a 10-yr event (the remaining seven models). The observations indicate a 1 to 2 °C temperature increase, with a return time of about 10 years. In contrast, the models only simulate about 1 °C.

Several observational and reanalysis datasets (ACORN stations, CERA-20C) and one model (ASF20C) display what appears

to be a non-stationary relationship between TX7x and GMST; as the starting time of a linear regression between them is varied from 1910 onwards, the best-estimate trend increases. For the ACORN stations this is probably due to the varying station coverage, with the trend over the stations active over the early part smaller, maybe more coastal, than over the later part. CERA-20C was excluded for not reproducing the 1929 event. ASF20C is initialised from ocean reanalyses. Due to increasing numbers and quality of observations over the 20th century these change from closer to the model climatology to closer to

the real state. This gives time-varying initialisation shocks, which is equivalent to a bias in the trend. Finally in the period before 1950 the global mean temperature was affected as much by volcanic and other natural forcings forcings as it was by



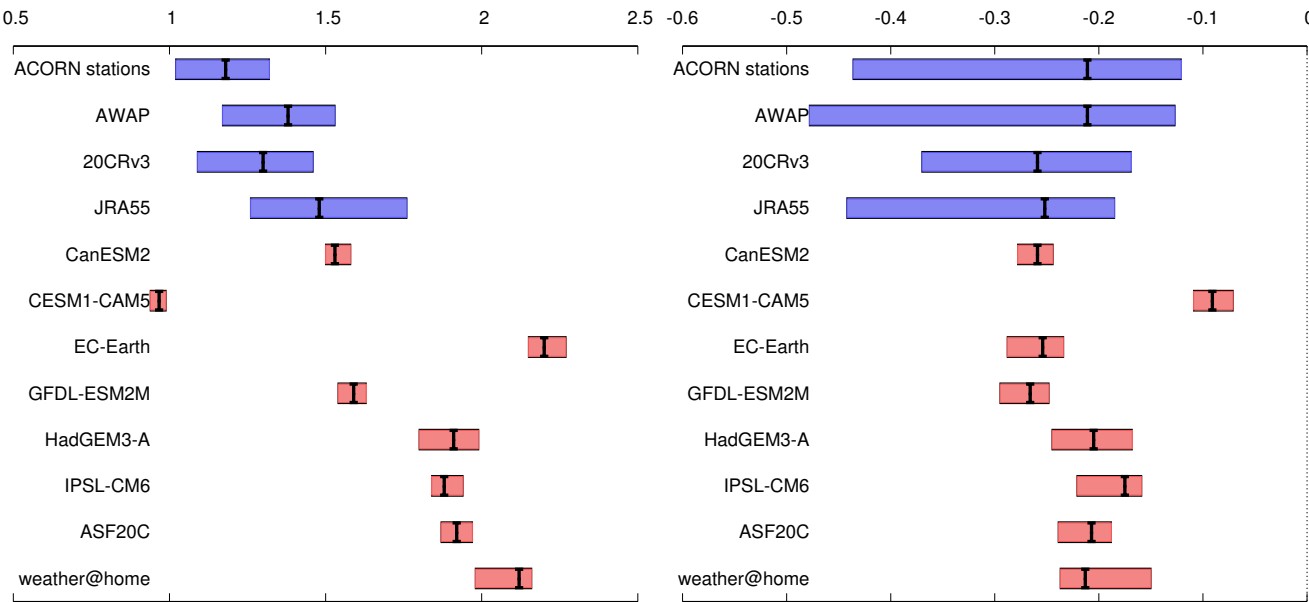

**Figure 5.** Left: scale parameter $\sigma$ (K) in GEV fits of TX7x in observations, reanalyses and climate models. Right: same for the shape parameter $\xi$.

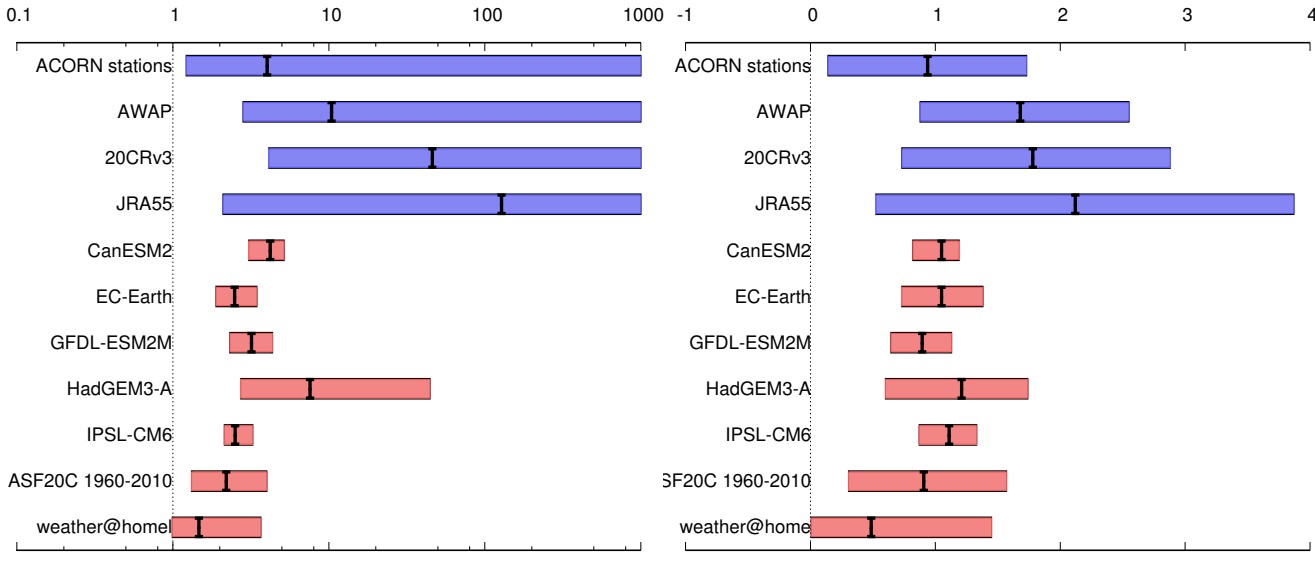

**Figure 6.** Synthesis plots of the Probability Ratio PR (left) and change in temperature $\Delta T$ (right) between 1900 and 2019 for the observations (blue), models (red). We do not attempt a synthesis as the models disagree too much with the observations.


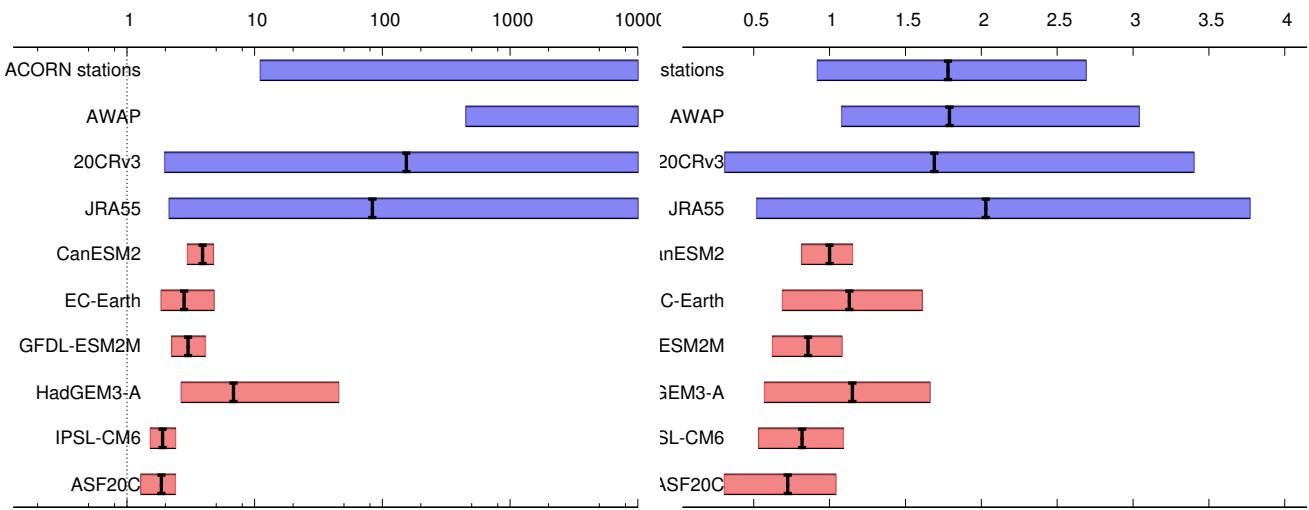

**Figure 7.** As Fig. 6 but using only data starting in 1950.

greenhouse gases, with possibly different effects from the anthropogenic forcings on circulation. We therefore also show a figure with results from 1950, Fig. 7, for which more consistent observations are available, noting that this is a better estimate of the greenhouse-gas driven trend with better observations than the whole period since 1900.

There are two interpretations of this discrepancy: either the observations are influenced by another driver than anthropogenic climate change that caused the rapid rise in extreme temperatures or the models have problems simulating the response of external forcing on these events and their related processes (or a combination of these two). As long as it is unknown which of the two explanations is correct we can only quote a lower bound on results, keeping in mind that the true increases could be much higher.

The Probability Ratios are very roughly ten in the observational datasets, with lower bounds as low as a factor two increase in probability (Fig. 6). The model results are heavily influenced by the overestimated variability: the high variability in the model, together with the low trends, induces lower probability ratios than in the observations. As there is no overlap between the observed and simulated values we do not attempt to synthesise the results but only quote a lower bound. The spread in the models is compatible with their estimates of natural variability ($\chi^2/dof \approx 1$) so we compute a weighted average. This has an

increase in probability between 1900 and 2019 of a factor three with a lower bound of a factor two.

**3.7    Conclusions extreme heat**

We analysed the highest 7-day mean maximum temperatures of the year averaged over the region south of 29 °S between the Great Dividing Range and the sea, the area with most intense bushfires in 2019/20. Observations show that a heatwave as rare





as observed in 2019/20 would have been 1 to 2 °C cooler at the beginning of the 20th century. Similarly, a heatwave of this

intensity would have been less likely by a factor of about 10 in the climate around 1900.

While eight climate models simulate increasing temperature trends they all have some limitations for simulating heat extremes: the variability is in general too high and the trend in these heat extremes is only 1 °C. We can therefore only conclude that anthropogenic climate change has made a hot week like the one in December 2019 more likely by at least a factor of two. Given the larger trend in observations in the models we suspect that climate models underestimate the trend due to climate

change. Coupled with the high variability of the models, the increase in the likelihood of such an event to occur is likely much higher than the models simulate.

## 4   Meteorological drought

### 4.1   Temporal Event Definition

Next we analyse meteorological drought, that is, low precipitation. The formulation of the Fire Weather Index only considers

precipitation over the last 52 days, as a proxy for this we also analyse the driest month in the fire season September–December. December 2019 was one of the driest months on record in our study region in southeastern Australia since 1900 (third in GPCC, ninth in AWAP). Using monthly data means we can utilise the models described above, thus sampling model spread as well as possible.

The January–December annual mean 2019 was the driest year on record since 1900 (Bureau of Meteorology, Annual Climate

Statement 2019). This was also the case in our study region in southeastern Australia, which could play a role in the bushfire risk that is not parameterised by the FWI. The two previous years had also been very dry, but it is unclear whether this still affects the 2019/20 bushfires. We therefore also analyse annual mean drought but not multi-year drought.

### 4.2   Observational precipitation data and methods

We considered three observational datasets of monthly precipitation: GPCC v18 1900–2018 (Schneider et al., 2018b) extended

with the monitoring analysis (Schneider et al., 2018a) up to November 2019 and the first guess analysis (Ziese et al., 2011) up to January 2020, CRU TS 4.03 1901–2017 (Harris et al., 2014) and AWAP 1900–January 2020 (Bureau of Meteorology data). As the distributions of annual mean precipitation and the driest month in the fire season are both not described well by a Gaussian we use a GPD fit to the lowest 20% or 30% for the observations, demanding that it has a lower bound ($\xi < 0$) that is larger than zero ($\sigma < -\xi u$) so that there is no probability for negative precipitation.

### 4.3   Observational analysis: return time and trend

For the annual mean low precipitation analysis the fit for AWAP data using the lowest 20% is shown in Fig. 8. The year 2019 is not included in the fit. This fit shows a significant trend towards more dry extremes over the period 1900-2018. The return time of 16 yr (3 . . . 550 yr) is the lowest in the observational datasets. The fit should be independent of the threshold, but this is not

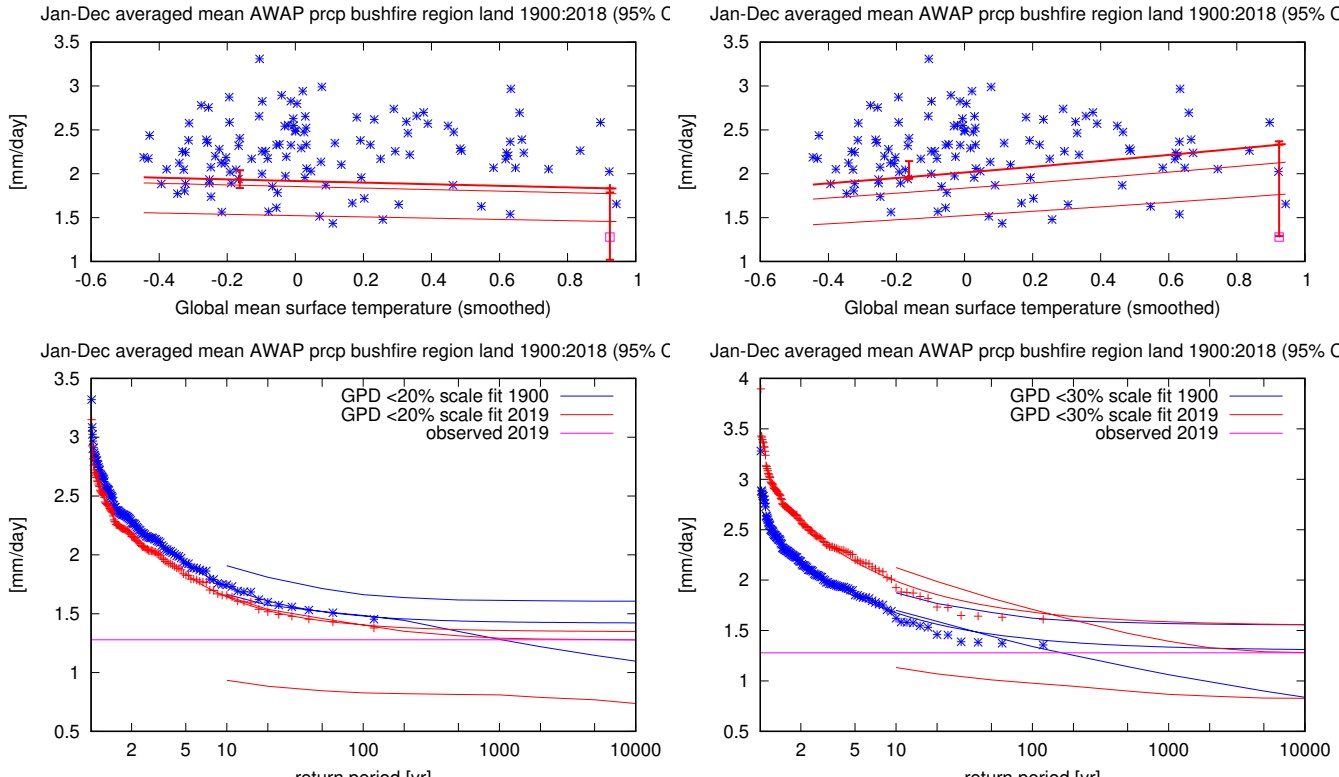

**Figure 8.** GPD fit to the AWAP estimate of annual mean precipitation in the bushfire region. The position and scale parameters depend exponentially on the observed smoothed global mean surface temperature such that their ratio is constant. The scale parameter is forced to be negative and the cut-off zero or higher. Left: using the lowest 20%. Right: using the lowest 30%. Top: observations (blue symbols), location parameter $\mu$ (thick line) and the 6 and 40 yr return values (thin lines). Bottom: return time plot for the climates of 1900 (blue) and 2019 (red), the purple line denotes the 2019/20 event.

the case: whereas the lowest 20% show a significant downward trend, the GPD fit to the lowest 30% has an upward trend that
is not significant at $p < 0.05$ (two-sided) in the AWAP dataset (Fig. 8, right). The lowest 10% does not contain enough data to
fit a GPD. We report both the 20% and 30% choices in the following.

  The return time of the low 2019 precipitation depends strongly on the observational dataset and the cut-off in the GPD fit
and ranges from 25 yr (3 . . . 4000 yr) in GPCC 30% to infinity in the AWAP fits with large uncertainty ranges starting at 3 yr,
with GPCC 20% intermediate at 120 yr (3 . . . 3000 yr). The uncertainty ranges are in fact more similar than the best fit values.
Also given that it was the lowest value in 120 years we evaluate the models at the rounded return time of 100 yr.

  We also fit a GPD to the lowest 20% and 30% of monthly precipitation amounts in the fire season September–February. For
some of the models we only had the driest month per season and the threshold is taken over the driest month in all ensemble
members. These extremes are more difficult to fit because the values approach the lower boundary of zero. In fact, in some

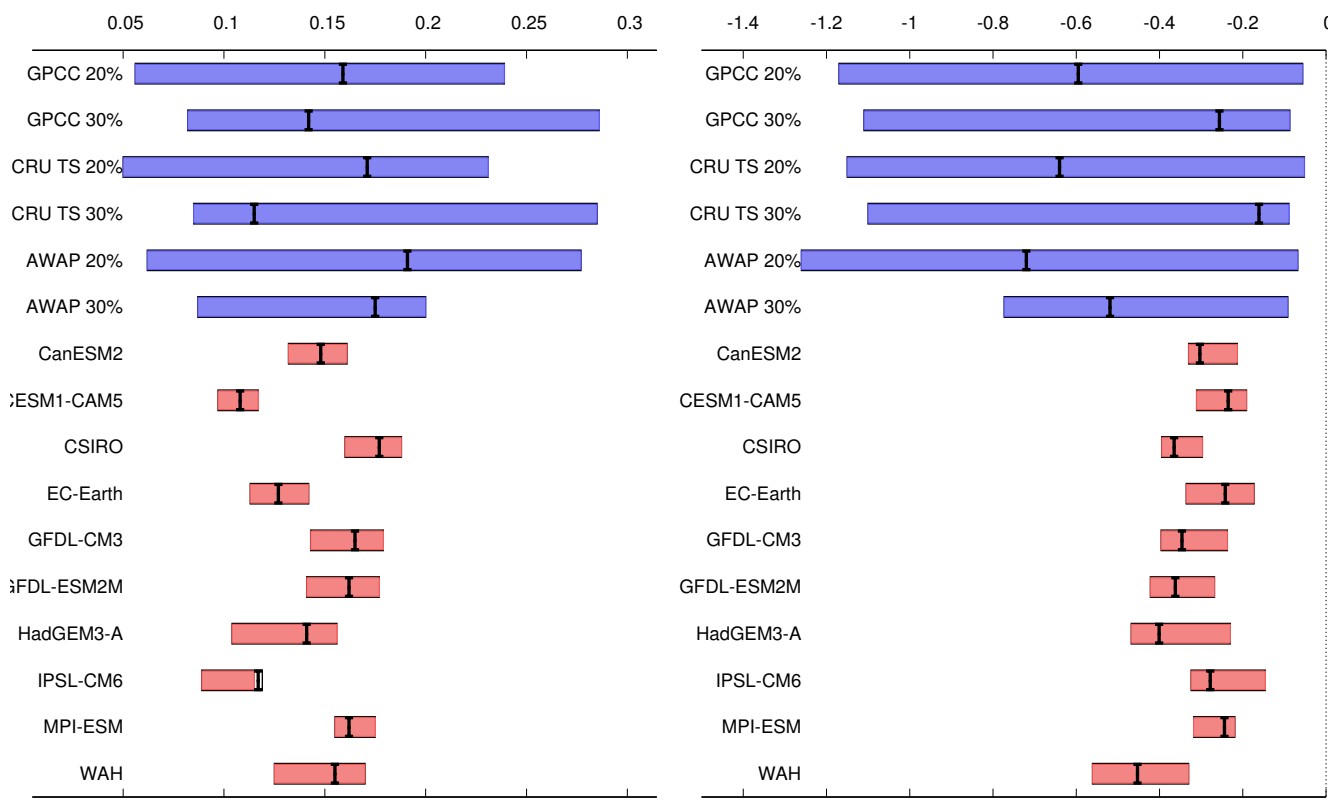

**Figure 9.** The dispersion parameter $\sigma/\mu$ (left) and shape parameter $\xi$ (right) of GPD fits to the low tail of the annual mean precipitation distribution in observations (blue, lowest 20% and 30%) and models (red, lowest 20%).

bootstrapped time series no initial conditions for the fit routine that satisfy all constraints could be found. The result is that the uncertainty range is too small for some fits and does not encompass the best fit.

The driest month in 2019/20 was December 2019. The return times obtained from the fits again vary widely, from 75 yr (15 . . . 200 yr) to 800 yr (10 . . . 250 yr), both for the GPCC analysis for the 20% and 30% thresholds. For the models we use a return time of 100 yr.

### 4.4 Model evaluation

We have data for the annual mean drought and driest month of the fire season for ten climate model ensembles. The ASF20C model only has 4-month runs starting four times per year and therefore cannot provide annual mean precipitation nor the driest month in the fire season and is not included here.

Fig. 9 shows that the statistical description of the low tail of annual mean precipitation of all models agrees with the same quantities in the observations within the (large) uncertainties due to natural variability.




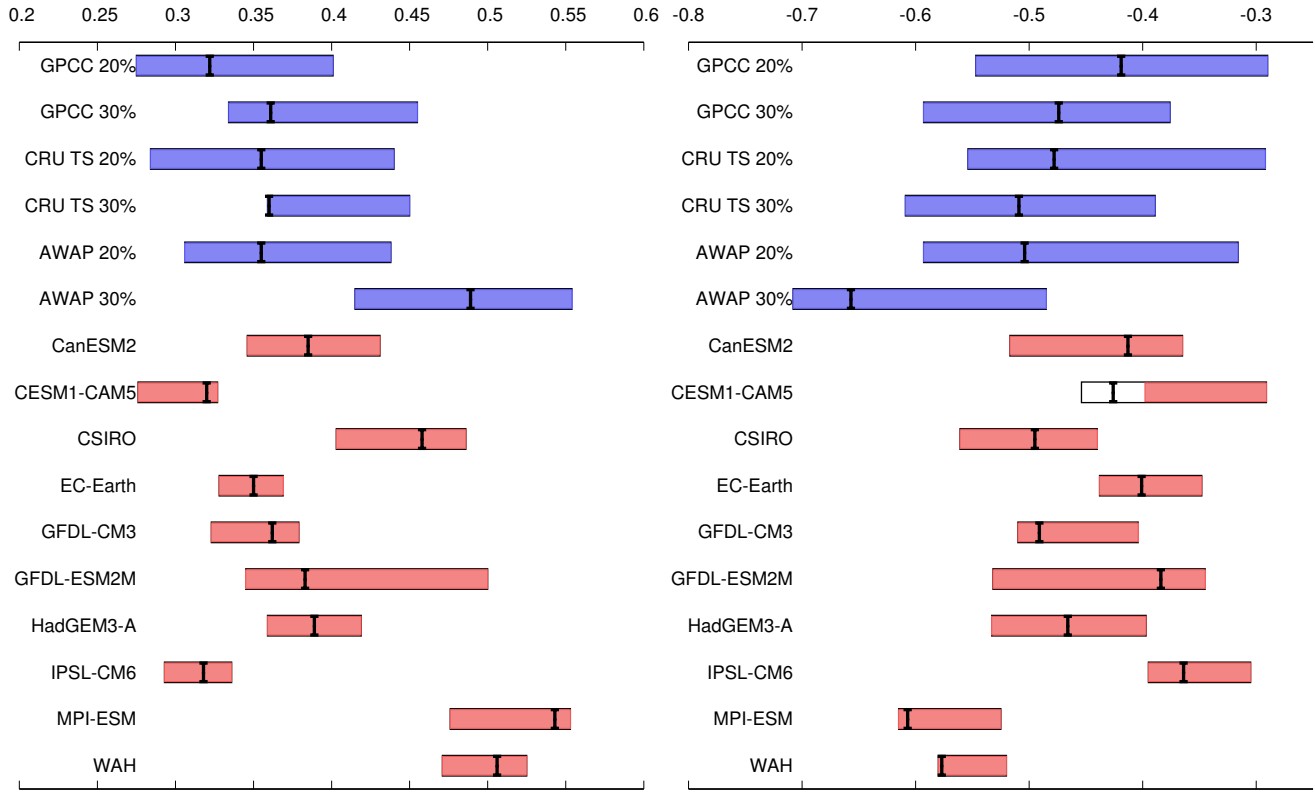

**Figure 10.** As Fig. 9 but for the driest month in the fire season September–February. The bootstrapped uncertainty intervals are sometimes underestimated due to fitting problems.

Fig. 10 shows that also for the driest month in the fire season all models have statistical descriptions of the low tail of the distribution that agree with the description of the observed tail within the large uncertainties of the various estimates of the observed climate.

### 4.5 Multi-model attribution and synthesis

Fig. 11 shows the change in probability and intensity of annual mean drought averaged over the bushfire region in southeastern Australia. The observations show a positive probability ratio PR and smaller intensities of low precipitation events, which implies a shift towards a larger probability for drought over the last 120 years. However, the models show no trend, so we cannot attribute these shifts to anthropogenic climate change. The observed trends can be due to natural variability (note that the 95% uncertainties encompass no change and these ignore the year-to-year autocorrelations, the lag-1 autocorrelation is borderline significantly different from zero, $a_1 \approx 0.2$). Another possibility is that they are due to other drivers not included





**Figure 11.** PR (left) and ΔP (%) (right) for annual mean low precipitation. The purple bar indicates the weighted average under the assumption that the model spread is equal to the model uncertainty, the white box around this purple bar gives equal weight to the observations and models.





in this analysis. Finally, the difference could again be due to shortcomings in the climate models. Note that the large natural variability can hide these shortcomings in the model evaluation in section 4.4.

Following Otto et al. (2018a) we investigate whether the lack of trend in the models is just due to high natural variability masking the trend by repeating the analysis using data up to 2100 for all models that also have future data (CanESM2, CESM1-CAM5, CSIRO Mk3.6.0, EC-Earth, GFDL CM3, GFDL ESM2M, MPI ESM), using the model ensemble-averaged GMST as

covariate. The reduced natural variability reveals a larger model uncertainty, but taking this into account the model trend is between $-7\%$ and $+7\%$, showing no systematic trend towards more dry extremes (not shown).

It should be noted that some of the models show large trends towards more drought in mean precipitation. What we find here is that 1 in 100 yr dry extremes do not follow this trend but stay relatively constant.

The summary figures for the driest month of the fire season are shown in Fig. 12. Although the fits are not very good,

both observations and models indicate a small, non-significant increase in precipitation, which is equivalent to a decrease in probability to observe as dry a month as December 2019 (PR < 1). Due to the fact that the uncertainties in the PR and $\Delta$P encompass zero, there is no attributable trend in the occurrence of very dry months in the fire season.

### 4.6   Conclusions meteorological drought

Observations show non-significant trends towards more dry extremes like the record 2019 annual mean and a non-significant

trend towards fewer dry months like December 2019 in the fire season. All ten climate models we considered simulate the statistical properties of the observations well. Collectively they show no trend in dry extremes of annual mean precipitation nor in the driest month of the fire season (September–February). We conclude that there is no evidence for an attributable trend in either kind of dry extremes like the ones observed in 2019.

## 5   Fire risk indices

### 5.1   The fire weather of 2019/20


As discussed in the introduction, the fire risk as parametrised by the definitions below were extreme in the study area in the 2019/20 fire season. This was reflected by disastrous fires during the season. The region was chosen to encompass these fires and therefore cannot be included in the fits.

### 5.2   Temporal Event Definition

For this analysis we choose two event definitions in order to represent two important aspects of the event, namely the intensity and the duration. For the former, we first select the maximum FWI of a 7-day moving average over the fire season (September–February) for every grid point over the study region, after which we compute the spatial average. This event definition will tell us more on changes in intensity of the bushfire season, and is hereafter labeled as FWI7x-SM (seasonal maximum). For the latter we compute the monthly severity rating (MSR). The MSR is the monthly averaged value of the daily severity rating




**Figure 12.** PR (left) and ∆P (%) (right) for the driest month in September–February. The purple bar indicates the weighted average under the assumption that the model spread is equal to the model uncertainty, the white box around this purple bar gives equal weight to the observations and models. For most observational datasets and some models the uncertainty range is underestimated due to problems fitting the data, this is taken care of in the averages by increasing the representation error and model spread terms (white boxes).





(DSR), which in turn is a transformation of the FWI ($\mathrm{DSR} = 0.0272\mathrm{FWI}^{1.71}$). The DSR reflects better how difficult a fire is to suppress, while the MSR is a common metric for assessing fire weather on monthly time scales (Van Wagner, 1970). For this event, we select the maximum value of the MSR of the fire season over the study area (MSR-SM). In contrast to the FWI7x-SM, we first apply a spatial average of the study area and then select the maximum value per fire season. This event definition provides information more on changes in extreme fire weather for both longer time scales and larger areas.

## 5.3 Observational analysis: return time and trend

For the observational analysis we use the ERA5 reanalysis dataset 1979–January 2020 (Hersbach et al., 2019). This reanalysis dataset is heavily constrained by observations, thus providing one of the best estimates of the actual state of the atmosphere for all the variables needed to compute the FWI over the study area.

Fig. 13 shows the time series of the highest 7-day-mean FWI averaged over the study area. Both for the FWI7x-SM and

MSR-SM the event is the highest over the 1979–2020 time period. Note that for the MSR-SM, the value is considerably more extreme than for the FWI7x-SM. The GEV-fits (Fig. 13, right) illustrate this further, with return times in excess of 1000 years.

A fit allowing for scaling with the smoothed GMST gives a significant trend in the highest 7-day mean FWI, averaged over the study area (Fig. 14). This fit gives a return time of the maximum in the 2019/20 fire season of about 31 yr (4 to 500 yr) in the current climate and more than 800 yr extrapolated to the climate of 1900. This corresponds to an infinite PR, with a lower

bound of four.

For the model analysis we will use a return time of 31 years for the FWI7x-SM. Because the return time for MSR-SM is undefined, we select a return time of 100 years for the models analysis.

## 5.4 Model evaluation

For the model analysis we use four climate models with large ensembles, leaving out CESM1-CAM5 because of its failure to

represent heat extremes (see section 3). This is fewer than for the drought and heat analysis, because the FWI requires four daily input variables, which are not available for all models. In contrast to the heat extremes and drought analyses, the models use as covariate the model GMST, and take as reference climates, where the years at which it is evaluated is taken from the 1.1 °C temperature increase for the present-day climate and the 2 °C increase for the future reference climate and not 2019 and 2060. As the fits are invariant under a scaling of the covariate this does not make much difference.

First the models are evaluated on how well they represent the extremes of FWI7x-SM and MSR-SM. This is quantified by the dispersion parameter $\sigma/\mu$ and shape parameter $\xi$, of the GEV fit for the present-day climate. We do not check the position parameter $\mu$, assuming a multiplicative bias correction can be applied.

Fig. 15 gives an overview of these parameters. Preferably, we would like the parameters to lie within the observational uncertainty (ERA5). For the dispersion parameter CanESM2 and weather@home fall within the observational uncertainty of

the FWI. The other two models (EC-Earth and IPSL CM6) show too much variability relative to the mean. The same holds for the shape parameter. This implies that it is difficult to draw strong conclusions from the model data, given that they do not
**Figure 13.** Left: time series with 10-yr running mean of the area-average of the highest 7-day mean Fire Weather Index in September–February (top) and MSR September-February maximum (bottom). Right: stationary GEV fit to these data.



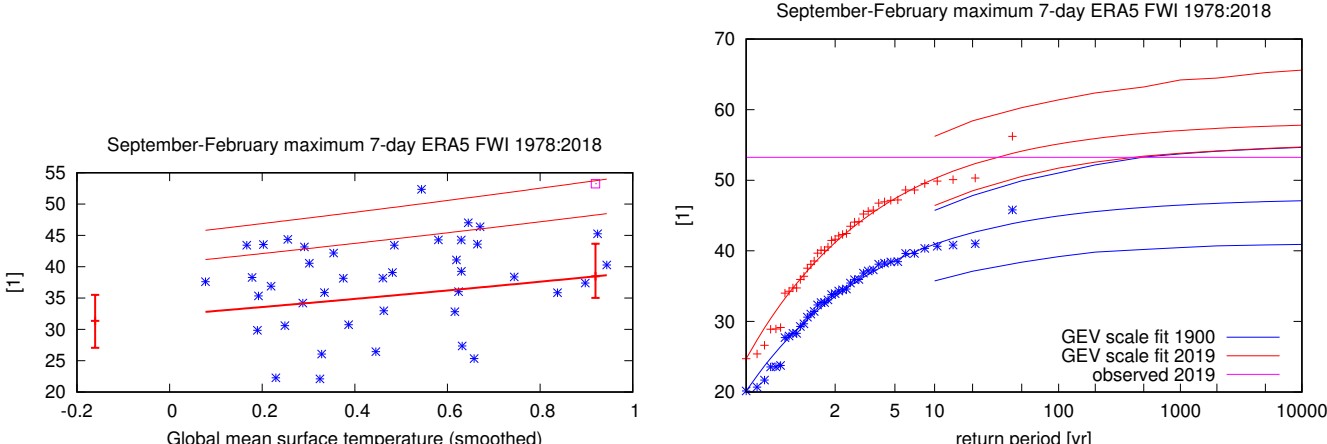

**Figure 14.** Fit of a GEV that scales with the smoothed GSMT of the highest 7-day mean FWI, averaged over the index region.

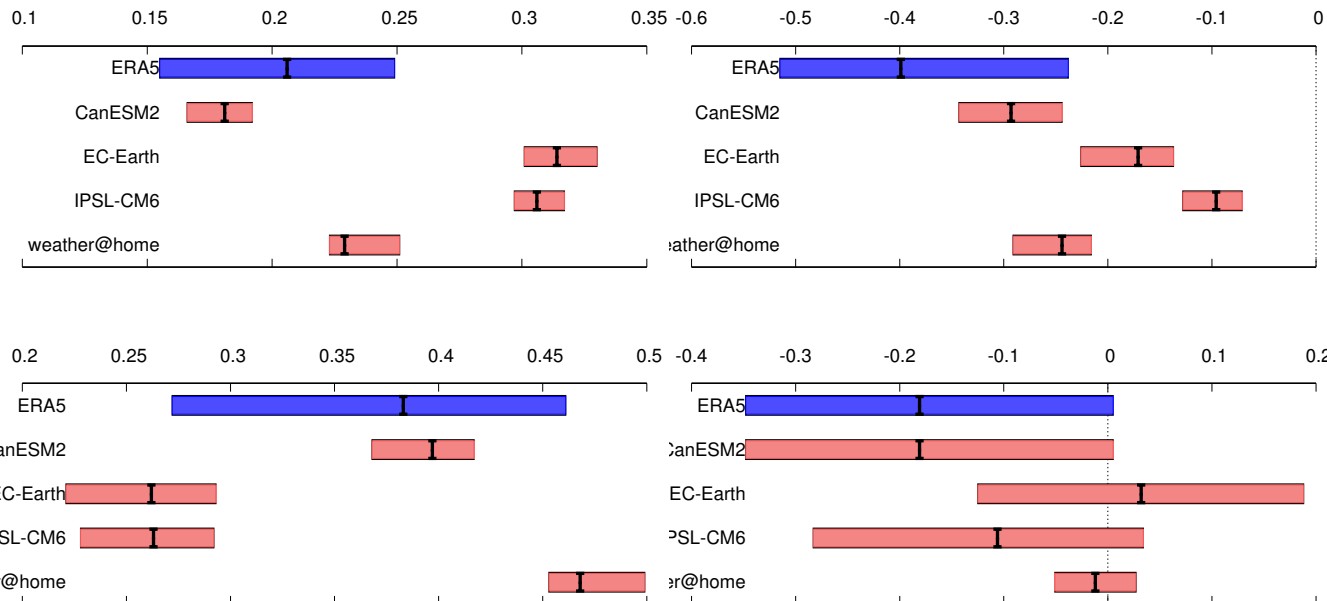

**Figure 15.** Model verification for FWI (top) MSR (bottom). The left figures show the dispersion parameter $\sigma/\mu$ and the right figures the shape parameter $\xi$.




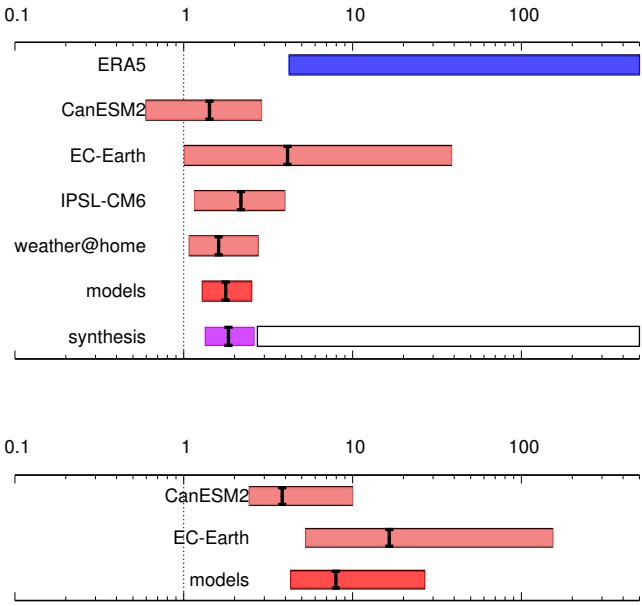

**Figure 16.** Top: PR for a FWI as high as observed in 2019 or higher, (top) from 1920 to 2019 and (bottom) from 1900 to a climate 2 °C warmer than late 1920. The last row is the weighted average of all models, the spread of which is consistent with only natural variability. Bottom: same for a 2 °C climate (GMST change from late 19th century).

accurately represent the extreme fire weather events. In particular, the models with too much variability will underestimate the probability ratios. We continue with all four models keeping these problems in mind.

The MSR is simulated better: all model dispersion and shape parameters lie within the large observational uncertainties, although they largely disagree with one another in the dispersion parameter.

## 5.5 Multi-model attribution and synthesis

The model results are summarized by their probability ratio, i.e., how more or less likely such an event will be for present or future climate, relative to early 20th century.

Figs. 16 and 17 show the change in probability for both the FWI7x-SM, and the MSR-SM. For the FWI7x-SM, all models agree on an increased risk for this event for the present climate relative to early 20th century, although the trend is not significant at $p < 0.05$ two-sided for one of the models, CanESM2. As the spread of the models is compatible with natural variability ($\chi^2/\mathrm{dof} < 1$) we take a weighted average. This shows that such an event has become about 80% more likely in the models, with a lower bound of 30%. Note that all models severely underestimate the increased risk compared to ERA5, which has a lower bound PR of a factor four, above the upper end of the model average.




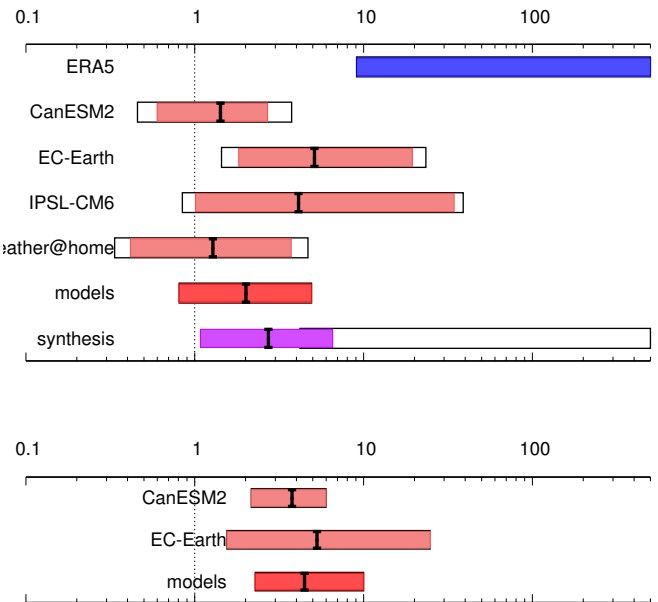

**Figure 17.** As Fig. 16 but for the Monthly Severity Rating (MSR).

For a future climate of 2 °C warming above pre-industrial we find that such events become about eight times more likely in the models, with a lower bound of about four times more likely. Note that the estimate of future climate is only based on two climate models, CanESM2 and EC-Earth.

For the MSR-SM we find on average for the present climate relative to early 20th century climate an increased risk of about two times more likely. However, this trend is not significant as the lower bound is 0.8, i.e., a decreased risk is also possible within the two-sided 95% uncertainty range. For future climate relative to early 20th century climate we find an increased risk of about four times more likely with a lower bound of two. Again, the fit to ERA5 data (including 2019) shows much higher probability ratios, with no overlap with the model results.

### 5.6 Interpretation

The underestimation of the observed trend in all models and tendency for too much variability in some models is reminiscent of the extreme heat results in section 3. In order to better understand which input variables cause the increased FWI, we select the input variables during FWI7x-SM for each model. For precipitation we select the cumulative precipitation (90 days) prior to the maximum value. By comparing these values for the early 20th century climate and present day we get the change in the input parameters during high FWI events. To analyse their individual impact on the FWI, we then subtract these changes of the input parameters calculated from model data to the ERA5 2019 data, and recompute the FWI for the adjusted 2019 event. The results are shown in Fig. 18.




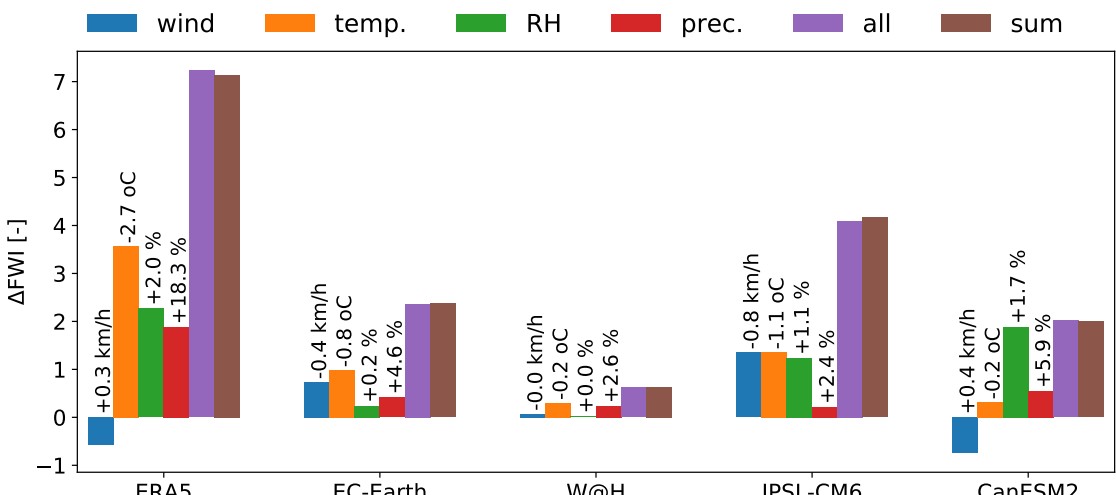

**Figure 18.** Sensitivity analysis for the FWI on the individual contribution of RH, wind, temperature and precipitation for all models used. The relative increases or decreases for the individual variables of the models are based on the average change in input variables during the seasonal maximum FWI7x between present day and early 20th century climate. These values are plotted above the bars. For ERA5 the changes are based on a linear regression of the input variables during FWI7x-SM with GMST for the years 1979 to 2018. These changes are subtracted from the 2019 ERA5 data, after which the FWI is recomputed, where $\Delta$FWI is the original FWI minus the altered FWI. In the 'all' experiment all input parameters are changed simultaneously. 'Sum' is the sum of all the individual changes in FWI.

First it should be noted that the sum of the individual contributions matches with the total effect, so the individual components add linearly. Second, the underestimation of the heat trends in these models carries over into this analysis, so the temperature bars are underestimated. Even with this underestimation, for EC-Earth and weather@home temperature is most important, as it explains roughly half of the increase in FWI. For IPSL, the simulated temperature increase explains about a third of the increase, together with wind and RH. CanESM2 behaves differently, where it is mainly the decrease of RH that explains the higher FWI. Most, but not all models analysed here therefore derive the increase in FWI to a large extent from the increase in temperature of heat extremes. For ERA5 the increase in temperature also appears to be most important in explaining the increased FWI, and to a lesser extent a decrease of RH and precipitation.

These results are in agreement with the findings from the heat attribution (section 3), given that we find an increase in temperature during high FWI events for all models. The underestimation of these models of the trends in heat extremes is therefore part of the explanation that all models underestimate the increase in FWI compared to the observations.

.





## 5.7 Conclusions fire risk indices

The observations of the Fire Weather Index show that the 2019 values were exceptional. They have a significant trend towards
higher fire weather risk since 1979. Compared with the climate of 1900, the probability of Fire Weather Index as high as in
2019/20 has increased by more than a factor of four. For the Monthly Severity Rating the probability has increased by more
than a factor of nine.

The four climate models investigated show that the probability of a Fire Weather Index this high has increased by at least
30% since 1900 as a result of anthropogenic climate change. As the trend in extreme heat is one of the main factors behind this
increase and the models underestimate the observed trend in heat, the real increase could be much higher. This is also reflected
by a larger trend in the Fire Weather Index in the observations.

The Monthly Severity Index increased by a factor of two in the models, compared with 1900, but this is not significantly
different from no change. Again, the real increase is likely much higher.

Projected into the future, the models simulate that a Fire Weather Index at the 2019/20 level would be at least four times
more likely with a 2 °C temperature rise, compared with 1900. Due to the model limitations described above this is likely an
underestimate.

## 6 Other drivers

There are known associations between large-scale climate drivers, such as El Niño—Southern Oscillation (ENSO), the Indian
Ocean Dipole (IOD) and the Southern Annular Mode (SAM), and fire risk (e.g., Williams and Karoly, 1999; Cai et al., 2009;
Harris and Lucas, 2019; King et al., 2020). These drivers of interannual variability can dominate the risk of fire weather over
the trend in individual years (Harris and Lucas, 2019).

### 6.1 ENSO

ENSO variations are linked to Australia's climate variability, with generally warm and dry conditions in eastern Australia
during El Niño, although the signal is weaker on the east coast. ENSO was considered neutral during 2019/20, however, the
western Pacific was anomalously warm and the Niño4 index indicated El Niño conditions of the Central Pacific 'flavour' (Ren
and Jin, 2011) with the relative Niño4 index in which the climate change signal is removed (van Oldenborgh et al., 2020) around
+0.6 °C during the 2019/20 fire season. The index that connects best with global teleconnections, relative Niño3.4 anomalies,
was neutral during the period. However, the atmospheric state was also typical for weak El Niño conditions with the Southern
Oscillation Index (SOI) of the National Centers for Environmental Prediction (NCEP) at around −0.6 until December 2019,
perhaps encouraging an enhancement of the IOD via atmospheric teleconnections (Cai et al., 2011).



## 6.2 Indian Ocean Dipole

The IOD is an interaction between the ocean and atmosphere in the tropical Indian Ocean basin with climatic influences around this basin (Saji et al., 1999). In general, IOD events develop in the Southern Hemisphere winter through spring, and break down with the start of the Indonesian monsoon in early summer. It is thus not an important driver through summer. In
the positive phase, colder than normal sea surface temperatures are observed in the eastern Indian Ocean (around Sumatra and Java). Positive IOD events are typically associated with below average winter-spring rainfall in Southern Australia and warmer than average conditions (White et al., 2014). As a result, IOD positive phases are associated with severe bushfire conditions (Harris and Lucas, 2019; Cai et al., 2009).

   In 2019, an unusually strong positive IOD event was observed, which, together with the Southern Annular Mode, was argued
to be primarily responsible for the precipitation deficit over South-Eastern Australia ((King et al., 2020)). This event started to emerge in June but matured and strengthened during early austral Spring. The late onset of the monsoon off Sumatra also led to the unusual persistence of the IOD into early summer in 2019. Variations in the IOD (as measured by the Dipole Model Index, DMI) explain only a small fraction of variance in observed temperatures in SON. In particular, the correlation with the AWAP TX7x index defined in section 3 is only $r = 0.22$. The correlation with the AWAP precipitation in July–December is
much higher, $r = -0.4$.

   This does include some double-counting in the statistical relation: as the IOD is often partially forced by ENSO, a fraction of these correlations are in fact ENSO teleconnections (although the influence was minor in 2019). When the influence of ENSO teleconnections, expressed as the regression on the relative Niño3.4 index, is linearly removed from the DMI these correlations become much lower, with no significant connection with TX7x and a correlation of $r = -0.22$ with July–December rainfall
(Fig. 19). However, even with this low correlation, the very high positive IOD conditions observed in July-December 2019 account for approximately one third of the July-December low rainfall, see Fig. 19. We chose this half year as it precedes the fire season September–February and therefore represents some of the drought preconditioning of the fire weather risk in this season. As the Niño3.4 index was neutral it did not contribute to the drought in the second half of 2019, although as mentioned above western Pacific SST and the atmospheric circulation did indicate a weak El Niño.

## 6.3 Southern Annular Mode and stratospheric preconditioning

The Southern Annular Mode (SAM) is an important mode of climate variability in the Southern Hemisphere related to variations in the large-scale atmospheric circulation. The positive phase of the SAM is associated with a stronger and more contracted stratospheric polar vortex and a stronger and more poleward located mid-latitude storm track. The impact of the SAM on the Australian climate variability is seasonally dependent and opposite precipitation responses are found in different regions
of the continent. Based on ERA-Interim reanalysis data, a strong negative subtropical precipitation response during spring and summer in southeast Australia has been found for the negative phase of the SAM (Hendon et al., 2014). During most of the second half of 2019 (July–December) the SAM phase was negative, with strong negative index values in November 2019. This negative phase of the SAM during 2019 has contributed roughly another one third to the extreme drought conditions in

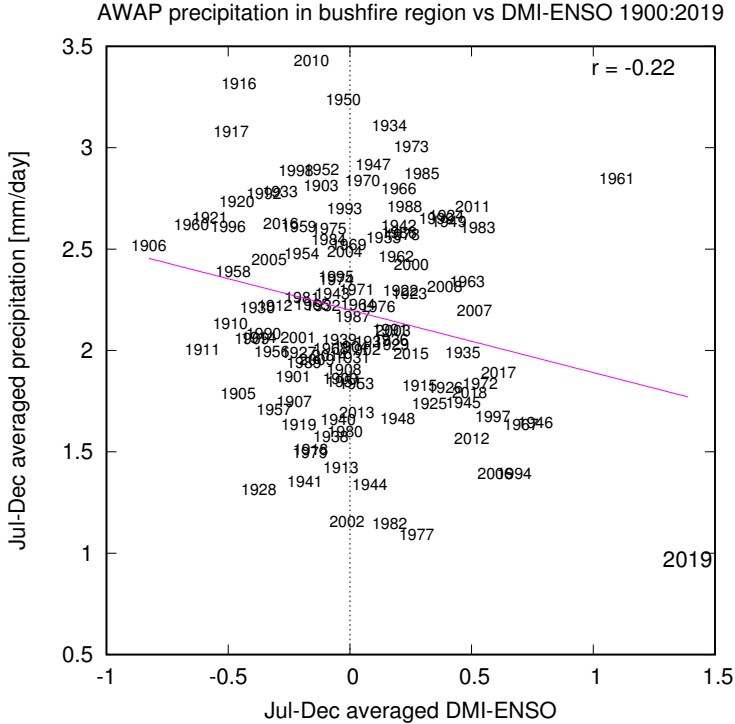

**Figure 19.** Scatterplot of AWAP precipitation averaged over the bushfire region in southeastern Australia preceding the fire season, July–December as a function of the DMI index with the monthly linear regression on the relative Niño3.4 index subtracted.

southeast Australia from July to December (Fig. 20). We did not find a significant correlation with our measure of heat waves,
TX7x, in agreement with Perkins et al. (2015).

The strong SAM excursion of 2019 was very well predicted. One cause was a Sudden Stratospheric Warming (SSW) event,
where the winter stratospheric vortex over Antarctica breaks down and the stratosphere warms rapidly. These are rare events,
with only two major events recorded, in 2002 and 2019. In 2019, the negative phase of the SAM from October onwards has
been preconditioned by the preceding SSW event through a downward coupling of the weakened polar vortex to tropospheric
levels as highlighted by Lim et al. (2020). SSW events are thus associated with warm and dry conditions over eastern Australia
(Lim et al., 2019).

Another interconnected cause of the record SAM state was the strong positive ozone anomaly in September, associated with
a relatively warm polar stratosphere and an exceptionally weak ozone hole season (UNEP, 2019). This was followed by a
period with a negative phase of the SAM from October onwards which in line with the ozone-SAM correlations found by Son
et al. (2013); Lim et al. (2018); Byrne and Shepherd (2018).

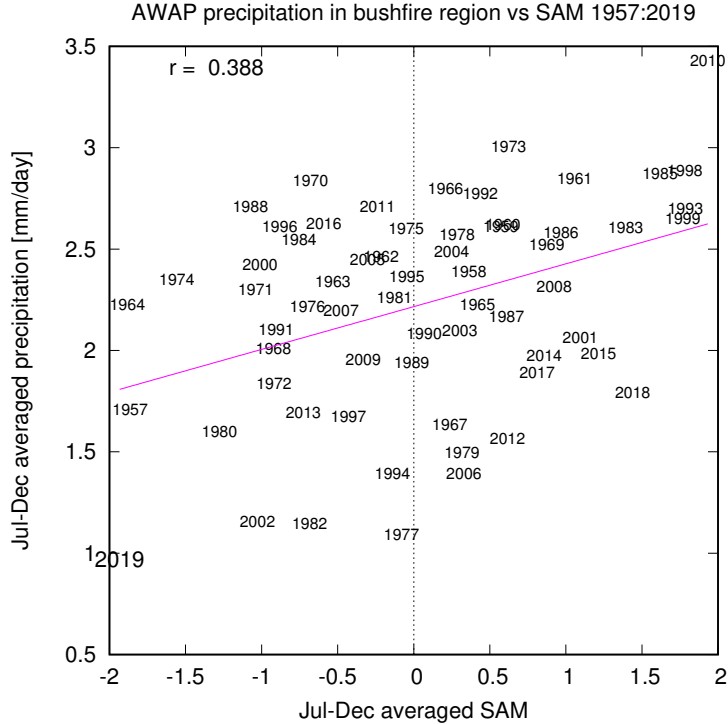

**Figure 20.** Scatterplot of AWAP precipitation averaged over the bushfire region in southeastern Australia preceding the fire season, July–December as a function of the SAM index.

## 6.4 Conclusions other drivers

The attribution statements presented in this paper are for events defined as meeting or exceeding the threshold set by the 2019/20 fire season and thus assessing the overall effect of human-induced climate change on these kind of events. In individual years, however, large scale climate system drivers can have a higher influence on fire risk than the trend.

Besides the influence of anthropogenic climate change, the particular 2019 event was made much more severe by a record excursion of the Indian Ocean Dipole and a very strong anomaly of the Southern Annular Mode, which together explain more than half of the amplitude of the meteorological drought (precipitation deficit). We did not find a connection of either mode to heat extremes.

## 7 Vulnerability and exposure

At least 19.4 million hectares of land have burned as a result of the Black Summer Bush Fires of 2019/20 (CDP, 2020). This has resulted in 34 direct deaths and the destruction of 5,900 residential and public structures (IFRC, 2020). Nearly 80% of Australians reported being impacted in some way by the bushfires (The Conversation, 17 February 2020). In Sydney, Canberra





and a number of other cities, towns and communities air quality levels reached hazardous levels (NY Times, 3 January 2020).
Over 65,000 people registered on Australian Red Cross' reunification site to look for friends and family, or to let loved ones
know that they were all right (Australian Red Cross, 2020). It is estimated that over 1.5 billion animals have died nationally
(Reliefweb Australia: bushfires, 2020). These impacts are not only hazard-related but also related to various vulnerability
and exposure factors that each play a role in increasing or decreasing risk and impacts. Vulnerability is 'The propensity or
predisposition to be adversely affected. Vulnerability encompasses a variety of concepts and elements including sensitivity or
susceptibility to harm and lack of capacity to cope and adapt' (Agard et al., 2014). It can also be defined as 'the diminished
capacity of an individual or group to anticipate, cope with, resist and recover from the impact of a natural or man-made hazard'
(IFRC, 1 March 2020). Exposure is defined as 'The presence of people, livelihoods, species or ecosystems, environmental
functions, services, and resources, infrastructure, or economic, social, or cultural assets in places and settings that could be
adversely affected' (Agard et al., 2014).

Bushfires have been a part of the Australian landscape for millions of years, and are an ever-present risk for people living
in rural and peri-urban areas surrounded by vegetation/bush/grasslands. Significant bushfires in the recent past include: Black
Saturday in 2009 where 450,000 hectares burned and 173 people died (National Museum of Australia, 23 May 2019); the
2002/03 bushfire season where 38 million hectares burned and seven people died (Ellis et al., 2004); the Ash Wednesday
bushfires in 1983 where 400,000 hectares burned and 75 people died (ABC Emergency, 14 February 2013); and the 1974/75
bushfires where 117 million hectares burned, largely in the less-populated center of the country, where it is estimated that three
people died (Australia Institute for Disaster Resilience). This frequent occurrence of severe bushfires, with records extending
back to the 1850s, has resulted in robust preparedness and emergency management systems which serve to reduce risk and
aid in swift response. Comprehensive risk assessments are undertaken at local council level, and bushfire preparedness and
contingency plans have been in place in most high-risk areas for decades. However, these systems were severely strained in the
Black Summer Bushfires.

## 595  7.1  Excess morbidity and mortality

The time of publication is too soon for a robust estimate of excess morbidity and mortality specific to the 2019/20 Australian
Bushfires. Such analysis is typically available weeks to years following the end of an event. However, the combined impacts of
extreme heat and air pollution are deadly. Those most at risk are the elderly, people with pre-existing cardiovascular, pulmonary
and/or renal conditions as well as young children. The 2010 wildfire and heatwave in Russia, for example, resulted in nearly
11,000 excess deaths. In that event wildfires raged for three months with a 44-day heatwave occurring in parallel (Shaposhnikov
et al., 2014), 300,000 hectares burned and 54 people died directly from the wildfires (Reuters, 17 Aug 2010, Nature, 12 Aug
2010). It has also been estimated that the 2015 wildfires in Indonesia, which lasted two-months, led to approximately 100,000
excess deaths (Koplitz et al., 2016).

In addition to deaths, exposure to wildfire smoke can have acute respiratory effects. Health officials in New South Wales
reported a 34% spike in emergency room visits for asthma and breathing problems between 30 December 2019 and 5 January
2020 (Washington Post, 12 Jan 2020). One study of hospital admissions in Sydney, Australia from 1994 to 2010 found that





days with extreme bushfires air pollution (as measured by PM10) resulted in 1.24% admission increase for every 10 $\mu\mathrm{gm}^{-3}$ (Morgan et al., 2010). On the other hand, it should be noted that Australia is a country with a robust healthcare system which significantly reduces vulnerability to the short and long-term consequences of smoke and extreme heat.

There is also a need for increased mental health services in the days, weeks and years following the bushfires. As of January 2020 the Australian Government announced 76 million AUD in Mental Health funding (ABC, 12 January 2020). A study of the 2009 Back Saturday bushfires found that while the majority of affected people demonstrated psychological resilience in the long-term aftermath of the fires, a significant minority of people in highly affected communities reported mental health impacts 3-4 years following the event (Bryant et al., 2014).

## 7.2   Early Warning

There is not a nationally standardized system for Bushfire Warnings in Australia.. However recommendations from the Royal Commission tasked with reviewing the 2009 Victoria Bushfire (Teague et al., 2010) have helped to drive forward efforts to establish a national system. In 2014 a National Review of Warnings and Information was undertaken. It recommended the establishment of a dedicated, multi-hazard National Working Group for Public Information and Warnings. Part of the task of

this group would be to ensure greater national consistency of early warning information. One outcome of this recommendation is a handbook on Public Information and Warnings which has been issued to provide guidance to actors across National, State and Territory governments in issuing warning information (AIDR, 2018).

Bushfire warnings in Australia are issued by State and Territory fire authorities and generally follow the 'Prepare, Stay and Defend or Leave Early' approach. A Fire Danger Rating system is also widely applied as a way to communicate fire risks.

The system, originally developed in 1967, contained five risk levels ranging from 'low-moderate', where fires can generally be controlled, to 'extreme', where evacuation is recommended but home defence may be possible if the home is located in specific places, specially designed and people have been trained on how to respond and practiced in drills. Following the 2009 Black Saturday bushfires a sixth 'catastrophic' level was added where evacuation is deemed the only survival option (bushfire warning levels). This was adopted by all states, except in Victoria where it is called 'Code Red' (BoM, 2020). In 2017, the

system was revised again, with an aim to improve back-end predictions by updating the metrics used in forecasting the most appropriate level (ABC News, 12 Dec 2017). Threat level information is provided via radio, television, social media and via signs on all major rural roads. Government websites also provide information which is updated every few minutes and includes maps of fires and associated threat levels. In addition, phone calls are made house-to-house when evacuation is recommended.

While these efforts help to reduce vulnerability and exposure to the wildfires, significant barriers to early action still remain.

People in bushfire areas, are frequently not aware of their risk, unprepared to manage risk, wait until the final moments to evacuate or, at times, even return to fire affected areas to defend property. (Whittaker et al., 2020) This is particularly relevant in peri-urban areas which are not as frequently exposed to bushfire risks. A 2020 study of people's reactions to bushfire warnings during the 2017 bushfires in New South Wales found that people largely understood warnings however they did not respond to the warnings in the ways intended by fire services. Consistent with other studies, researchers found that people tend

to seek confirmation of the bushfire threat before evacuating, in order to avoid unnecessary evacuation and associated expenses.





Researchers recommend that rather than further refining messages, first responders need to build in confirmation mechanisms into early warning approaches. Researchers also found that many people do not accept key messages tied to the catastrophic fire level. Specifically, people frequently do not follow recommendations to leave early, before there is a fire, and often do not believe that their house is not defendable at this level of danger (Whittaker et al., 2020). This is compounded by personal desires to keep pets and animals safe, or a determination to personally protect property in the fear that the emergency services will not succeed. There is also a 'hero' culture around those who succeed in defending their homes. These sociological barriers to life-saving measures increase the risk of deadly impacts.

Furthermore, an inquiry into the 2009 Black Saturday bushfires found that the 'Prepare, Stay and Defend or Leave Early' approach assumed that individuals had a fire plan in-place however many people did not. Therefore people were left in a position to make complex decisions without adequate guidance.

## 7.3 Controlled burning and relation to weather conditions

During the 2019/20 fire season, there has been wide discussion about prior fire hazard management strategies, in particular the management of vegetation cover including through bush reduction, manual removal of undergrowth and controlled burning, which is carried out by various actors, and coordinated by various authorities (mainly at state and local level). For this study, we did not assess vegetation cover and condition (dryness) ahead of the season in comparison with earlier years.

Note that the effectiveness of such measures depends on type of vegetation, but also the specific conditions of the bushfire. For instance, prior controlled burning may be somewhat effective to suppress fire risk under average weather conditions, but much less so in cases of very high temperatures, low humidity and strong wind; when the fire risk is no longer dominated by the type, condition and quantity of fuel but by weather conditions. For instance, for the case of the 2009 fires in Victoria, it was shown that recently burnt areas (up to 5–10 years) may have reduced the intensity of the fires, but not enough to increase the chance of effective suppression given the severe weather conditions at the time (Price and Bradstock, 2012).

In addition, it should be noted that controlled burning requires a window during the cooler parts of the year when conditions allow controlled burning to take place, but are not yet so volatile that controlled burning becomes too dangerous. In the case of the 2019 season in Queensland for instance, the Queensland Fire and Emergency Services (QFES) noted that controlled burning is highly dependent on weather conditions, and that not all planned 2019 burns had been completed, given that in some areas, it rapidly became too dry to burn safely (ABC Fact Check, 20 December 2019). Recognising the highly non-linear relationship between weather conditions through the season (and in fact across several years) and anticipatory risk management strategies, in this attribution study we have not assessed the impact of these early-season weather conditions on the ability to reduce risk, and thus on fire risk itself.

## 7.4 Infrastructure and land use planning

Aging electricity infrastructure may play an role in increasing the risk of bushfire outbreaks. For example, an inquiry report following the 2009 Black Saturday bushfires estimated that 200 fires per year are started in Victoria due to the ageing electricity grid (Teague et al., 2010). Electric grid fires are primarily due to elastic extension and fatigue failures, and are made increasingly





worse by high wind speeds (Mitchell, 2013). A 2017 study found that fires sparked by electricity failures are more prevalent
during elevated fire risk and tend to tend to burn larger, making them worse than fires due to other causes (Miller et al., 2017).
Interestingly a 2013 report also notes that while electricity operators have the ability to disconnect electricity grids when there
is a high-risk posed to the public, only South Australia has legislation in place to protect the operator from prosecution (Energy
Networks Association, 2013). All of these factors coupled together increase Australia's vulnerability to bushfire outbreaks.

In contrast, stringent building codes have helped to reduce vulnerability to fire risks. The Bushfire Attack Level (BAL), and
associated building codes, is the guiding resource for assessing and managing risks of a building exposed to heat, embers or
direct fire. The BAL is applied nationwide, however the Fire Danger Index, one of the key metrics used in calculating the
site specific BAL, is under state and local level jurisdiction. The first BAL level is an indication of insufficient risk to warrant
special construction requirements. There are five additional levels of potential risk, each with increasingly strict building
code requirements. The highest of these requirements, BAL-FZ, was established following the 2009 Black Saturday bushfires.
Illustrative examples of building restrictions at this level include window and door systems that can withstand up to 30 minutes
of fire exposure and construction materials for decking, walls and roofs must be non-combustible, such as stone or bricks.
Property owners can also exceed these codes by installing items like sprinkler systems, wind protection and enlarged defendable
space (Country Fire Authority, 2012).

Furthermore, land use planning at a community level is also crucial in reducing bushfire risk, particularly for rural and per-
urban areas which face the highest bushfire risks. This is recognised and addressed through each state and local government
planning processes, which includes ensuring accessible bushfire evacuation routes and spaces. For example, the 2009 Victorian
Bushfires Royal Commission cited a need for planning which 'prioritized human life over all other policy objectives'. This
led to relevant policy changes through an amendment to Victoria's Planning Provisions. The Bushfire Management Overlay,
and associated guidelines, are among the principle aspects of this amendment. They provide direction for approval of new
construction locations as well as siting and layout requirements of approved spaces, although these guidelines do not apply to
existing property which puts a limitation on their overall positive impact (Country Fire Authority, 2012).

### 7.5 Volunteer fire response

The Australian states of Victoria, New South Wales, South Australia and Western Australia rely on a century-old model of
volunteer fire services to combat bushfires. This model has historically worked due to the episodic and fragmented nature of
bushfires in Australia. In the 2019/20 bushfire season the dedication of this volunteer force saved countless lives and reduced
what could have been significantly larger impacts. This largely unparalleled volunteer force is a crucial asset to managing
bushfire response in Australia. For example, the New South Wales Rural Fire Service has over 72,000 volunteer fire responders,
roughly 900 paid staff and provides emergency fire services to 95% of New South Wales (NSW RFS, 30 June 2018).

However the intensity and longevity of the 2019/20 bushfires has also raised questions regarding the sustainability of the
current volunteer fire response model in a changing climate. During this bushfire season, volunteers faced long hours and
extended periods of leave, in many cases depleting their various forms of paid time-off, and/or taking leave without pay. In
the case of unemployed volunteers who were receiving social security benefits, a waiver mechanism was activated for 13



weeks to avoid benefit cuts due to not being able to attend job interviews and complete other required processes (Social
Security Guide, 10 Feb 2020). In the case of New South Wales for example many volunteers exceeded 100 days of volunteer
service, which can take a significant toll financially as well as from a physical and mental health perspective (ABC News, 12
Dec 2020). In December 2019, federal employees were made eligible for four weeks of paid leave and, following political
pressure, compensation was extended to non-Federal employees for up to 20 days. While this compensation helps, questions
remain regarding how to bolster this volunteer force as Australia's bushfire seasons become longer and more intense. Proposals
include formalising compensation schemes, provision of professional training, emergency driving tests, physical fitness tests
and increased budget allocations for protective equipment (as many volunteers buy their own).

### 7.6   Conclusions vulnerability and exposure

Bushfires are a natural phenomenon, but their impact is also strongly influenced by human choices. The bushfire warning sys-
tem in place in Australia worked well, but research shows that many people do not heed warnings in the ways intended. The
risk of bushfires also includes many other anthropogenic factors, for instance increasing risk due to aging electricity infrastruc-
ture, and risk reduction for instance through building codes and land use planning. The effectiveness of some risk management
options to mitigate against bushfire risk, such as using controlled burning, can also be affected by weather conditions. And
some systems to respond, such as voluntary fire response, may come under strain under extreme conditions such as this fire
season. Overall, however, Australia is one of the most prepared countries in the world to manage bushfires and thus the impacts
from this season's bushfire outbreaks could have been dramatically worse if not for the systems in place. This underscores the
urgent need to adapt to changing risks in all places, with a special focus on the most vulnerable, but also highlights the limits
to risk reduction and preparedness.

As a result of the Black Summer bushfires, formal inquiries have been launched in Victoria (Victoria Government, 14 Jan
2020), New South Wales(New South Wales Government, Jan 2020), Queensland (Queensland Government, 18 Dec 2019) and
South Australia (South Australia Government, 21 Jan 2020). A Federal Royal Commission has also been announced with
an aim to improve resilience, preparedness and response to disasters across all levels of government. The Commission will
also seek to improve disaster management coordination across local government, improve relevant legal frameworks. These
inquiries will undoubtedly shed additional light on the vulnerability and exposure elements of the Black Summer Bushfires,
and hopefully help mitigate future risk.

## 8   Conclusions

We investigated changes in the risk of bushfire weather in southeastern Australia due to anthropogenic climate change, un-
derpinned by changes in extreme heat and extreme drought. The latter have longer time series and are covered by many more
climate models, leading to more robust conclusions. The fire risk is described by the Fire Weather Index, which was shown to
correlate well with the area burnt in this part of Australia.





The first conclusion is that current climate models struggle to represent extremes in the 7-day averaged maximum tem-
perature which was chosen as the most impact-relevant definition of heat as well as the Fire Weather Index. They tend to
overestimate variability and thus underestimate the observed trends in these variables. Both of these factors give an underesti-
mation of the change in probability due to anthropogenic climate change (PR). We therefore do not give best fit values but only
lower bounds for these variables.

We find that the probability of extreme heat has increased by at least a factor two. We do not find attributable trends in
extreme drought, neither on the annual time scale nor for the driest month in the fire season, even when mean precipitation
does have drying trends in some models. Commensurate with this we find a significant increase in the risk of fire weather as
severe or worse as observed in 2019/20 by at least 30%. Both for extreme heat and fire weather we think the true chance in
probability is likely much higher due to the model deficiencies.

The fire weather of 2019 was made much more severe by record excursions of the Indian Ocean dipole, even when the
ENSO teleconnection was removed from this. The average effect of this mode is small, but the anomaly was so large that this
factor explains about one third of the anomalous drought in July–December 2019. The other factor was the Southern Annular
Mode, which was also anomalous during this time, explaining another one third of the July–December drought. Both factors
were predicted well and gave good warning of the high fire risk in late 2019. The variability due to these modes is included in
our analysis, although the simulated fidelity of the modes themselves and their trends has not been assessed in detail here. It
should be noted that only a small fraction of the natural variability is described by these modes.

Of course the full fire risk is also affected by non-weather factors. The bushfire warning system in place in Australia worked
well, but research shows that many people do not follow the guidelines as intended. The risk of bushfires is increased due
to anthropogenic factors like aging electricity infrastructure. Efforts to mitigate against that risk using controlled burning
are hampered by the very high fire risk due to weather factors shrinking the window in which controlled burning can be
safely executed. Overall, however, Australia is one of the most prepared countries in the world to manage bushfires and thus
the impacts from this season's bushfire outbreaks could have been dramatically worse if not for the systems in place. This
underscores the urgent need to adapt to changing risks in all places, and especially the most vulnerable.

Although we clearly identify a connection between climate change and fire weather and ascertain a lower bound, we also
find, in agreement with other studies, that we need more understanding of the biases in climate models and their resolution
before we can make a more quantitative statement of how strong the connection is and how it will evolve in the future.

**Data availability**

Almost all data used in the analyses can be downloaded from the KNMI Climate Explorer at https://climexp.knmi.nl/bushfires_
timeseries.cgi. This also contains the scripts that we used to do all the extreme value fits. These can also be performed using
the graphical user interface of the Climate Explorer.



*Acknowledgements.*   We thank Pandora Hope, Andrew Dowdy and Mitchell Black of the Bureau of Meteorology and Sarah Perkins-Kirkpatrick of the University of New South Wales for substantial contributions to the article. G.J.v.O was supported by the ERA4CS projects SERV_FORFIRE and EUPHEME (grant #690462). The contribution of F. K. was supported by the Belmont Forum Project PREREAL (grant # 292-2015-11-30-13-43-09 to I.D.). F.L. is supported by the Regional and Global Model Analysis (RGMA) component of the Earth and Environmental System Modeling Program of the US Department of Energy's Office of Biological & Environmental Research (BER) via NSF IA 1947282,

and by a Swiss NSF Ambizione Fellowship (Project PZ00P2_174128). We acknowledge the use of data and imagery from LANCE FIRMS operated by NASA's Earth Science Data and Information System (ESDIS) with funding provided by NASA Headquarters. We would like to thank all the volunteers who have donated their computing time to climateprediction.net and weather@home.





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
