# Peer review of "Attribution of the Australian bushfire risk to anthropogenic climate change"

_Natural Hazards and Earth System Sciences, 2020_

## Short Comment (SC1) · 17 Mar 2020

In my view, three key questions need to be addressed before we can attribute the bushfire risk to climate change, at least quantitatively.

1. The predominant cause of the bushfires were unquestionably the exceptionally (unprecedented) dry conditions in 2019. These were linked to the IOD (and other remote factors). Did climate change makes such dry conditions more likely? A pre-requisite to answering such questions is that we can simulate extremes of the IOD, the regional heat waves and the teleconnnections between the two. Studying 7-day heat waves does not address this issue adequately. We should be looking at model fidelity on monthly to seasonal timescales, where teleconnection biases are known to be substantial. The dry conditions of 2019 persisted into 2020 partly through continuation of the IOD, and partly because of the negative SAM.

2. The Fire Index has a dependency on temperature which presumably relates to the fact that vegetation dries out more at higher temperatures. Of course, in general terms this is entirely reasonable. However, at the beginning of summer 2020, the vegetation was already extremely dry due to the exceptional conditions of 2019. It is not clear to me that under these circumstances additional temperature increased the fire risk much further than the critical value it was already at. In this sense the dependence of the fire index on temperature for these exceptionally dry conditions may not be correct.

3. The one factor where anthropogenic climate change may have been important, but has not been taken into account here, is the $CO_2$ fertilisation effect, i.e there was simply more vegetation to burn.

Unless we can answer these questions, then I do not think we can, or indeed should, be making quantitative estimates of the impact of anthropogenic climate change on these bushfires.

Instead we must focus effort on developing a next generation of model where the regional dynamical effects of climate change can be simulated with much more confidence than is currently possible (Palmer and Stevens, 2019). Such models are likely to require much greater resolution than we have now - in particular allowing the convective rainfall anomalies associated with the IOD to be represented with the laws of physics rather than with relatively crude parametrisations.

Palmer, T.N. and B. Stevens, 2019: PNAS, 116, 24390-24395.

---

## Referee Comment (RC1) · Anonymous Referee #1 · 27 Mar 2020

Review – *Attribution of the Australian bushfire risk to anthropogenic climate change* by van Oldenborgh et al.

This paper describes an attribution analysis of a number of factors that are known to either contribute to or reflect wildfire risk using observational data, observationally constrained data products (reanalyses), and a collection of CMIP5 model simulations. It also considers the impacts of internal climate variability as reflected in the large-scale modes of variability that influence the Australian climate, and it includes a discussion of vulnerability and exposure factors associated with the impacts of the summer 2019/2020 wildfires.

I found the paper frustrating to read and evaluate. One clear impression is that the authors were in a terrible hurry, producing text that often appears not to have been carefully proofread, not thinking carefully about how to describe their methods in a clear way, not always justifying methodological choices, not justifying choices of data products or evaluating those products with a sufficiently critical eye, and attempting to be overly comprehensive. Reading the paper is a bit like being forced to "drink from a firehose" – there are so many details and so many small aspects that can be criticized, that is difficult to know exactly how and what to criticize in a review. The fact that all code is being made available doesn't really reassure me very much. Readers who want to understand what was done, sufficiently so that the work can be replicated, shouldn't be placed in a position of having to read code but rather, should be provided with explanations in the paper that are clear enough so that they can develop and implement their own code.

Some specific comments:

1-14: The abstract does not mention the long section on vulnerability and exposure factors, and there is no reference to vulnerability and exposure in the title. Does that section really belong in the paper?

16: The very first sentence of the paper starts by being sloppy in the way in which Australian station data are characterized. The word "homogeneous" has a very clear and well understood meaning in the context of observational data products (i.e., meaning that observations have been carefully evaluated and adjusted to ensure that they are free of artefacts resulting from changes in instrumentation, instrument siting, instrument housing, observing and reporting practices, etc., etc.), and surely the claim here is not that Australian station data is homogeneous in that sense. Clearly, avoiding the obvious inhomogeneity due to the lack of proper instrument shielding early in Australian instrumental record is necessary, but we shouldn't just accept that all of the subsequent record is homogeneous.

26-27: What is the source of this estimate? Is it possible to have any confidence in that number or the range that is given?

27-29: Again, what are the sources?

Figure 1: Is there a URL and a date for where this image was obtained?

93: I imagine daily maximum temperatures are meant. There are many instances in the paper where a second reading of the words, just to see if the connect logically, would have helped enormously. There are also a large number of run-on sentences in the paper that are difficult for readers to parse and understand.

102: This subsection is entitled "Event definition", but it doesn't talk specifically about event definition at all. I think what is needed is a clear statement that the event of interest will be defined using the FWI. This section gives some justification for doing that by considering the relationship between FWI and area burned, but event definitions per se are not discussed in this subsection.

Figure 2 caption: Please tell me what is meant by a "one-sided confidence interval about zero". I assume you mean the interval from -1 to the expected 95$^{th}$ quantile for the correlation coefficient under the null hypothesis that the correlation is zero. If this is correct, then it would be better to call this the 5% significance critical value for a one-sided test of the null hypothesis that the correlation is zero against the alternative hypothesis that the correlation is positive.

129: Often, acronyms like ASF20C appear before they are defined.

137-152: Some careful justification for the distributional choices would seem to be in order. These distributions emerge in statistical extreme value theory as limiting distributions under idealized conditions, where the limit is taken either as block length increases without bound in the case of the GEV, or as the exceedance threshold increases without bound in the case of the GPD. Given the way the data are processed, we are likely a long way from being able to be satisfied that the actual distributions are well approximated by these limiting distributions. Indeed, it seems likely that the relative quality of the fit will diminish as you go deeper into the tail, even if quantile plots look to be ok. In particular, one should be worried about extrapolating beyond the available data. Some aspects of this are discussed later in the paper, but those limitations don't really seem to prevent the authors from referring to values that appear to correspond to very long return periods in some instances. In the case of precipitation deficit, any of a number of possible candidate distributions could presumably be considered if using as much as 30% of sample values. These would have different deep tail characteristics, affecting calculations of probability ratios, but might not be discernably better or worse than the GPD based on standard diagnostics of the fit. So how does one proceed in a careful way take this source of structural uncertainty into account? It might be as important as the structural uncertainty represented by the spread between models.

160: Why 4-years and not some other degree of smoothing? Exactly how is the smoothing done, and how is time referenced to the smoothed values? For example, if using a 4-year running mean, which year is the value associated with in covariate dependent functions?

165-172: Choices for how the GEV and GPD distributions are parameterized should be justified and carefully argued, not just stated. For precipitation, exponential scaling might make sense at the upper end of the precipitation distribution, but why would I consider that to be reasonable at the lower end of the distribution, and why, in that case, should the scale parameter be linked to GMST? Building in something that scales like Clausius-Clapeyron might not be the best idea for the dry end of the precipitation distribution.

187-189: Is it obvious that this is the best way to proceed? If the analysis was literally performed as described here, the effective block size for the models would be 5- or 10-times the block size used for the observations. That means that for the models, the block maxima used to fit GEV distributions would sample a much deeper part of the tail than is possible with the observations since the distributions for the model output would have been fitted to what are effectively 5-year or 10-year blocks rather than 1-year blocks as for the observations. How then, can I make sense of differences in parameter estimates between fits to observed and simulated parameter estimates.

191-192: I think this is all that is said about bias correction in the paper except for another brief mention at line 442, but surely this is important and should be discussed (and defended) in some detail. Exactly what was done, and how does this avoid overusing the observational data?

200: Exactly what do you mean by the $\chi^2$/dof statistic (what is calculated, and what is the basis for the interpretation given to this statistic)?

240: For each observational product, the paper should draw attention to the key limitations that would affect the analysis in this paper. For example, although Stevenson screens begin to be used in 1910, there could be many other reasons to be concerned about the homogeneity of temperature observations, such as variations in station coverage over time (e.g., spatial sampling in 1910 would undoubtedly have been different than in the 1970s). Also, the paper should make a clear distinction between observational products on the one hand, and observationally constrained products (re-analyses) on the other. The latter are clearly non-homogeneous, with inhomogeneity due to changes in data sources, quality and quantity over time being of particular concern in the southern hemisphere where the observational constraint is much weaker. Ensemble reanalysis products, such as the 20th century reanalysis may be able to provide information about the strength of the observational constraint and how it varies in space and time (if the spread between ensemble members is large, the constraint is obviously weak or non-existent; if the spread is small, one has further work to do to determine if it is small because the analysis is being effectively constrained by the observations or whether this is coming about for another reason). Further, it should be noted that surface variables are often not very well constrained in reanalyses. The classification of variables by strength of observational constraint that is given in Appendix A of the Kalnay et al paper describing the original NCEP 40-year reanalysis (BAMS, 1996, [https://doi.org/10.1175/1520-0477(1996)077<0437:TNYRP>2.0.CO;2](https://doi.org/10.1175/1520-0477(1996)077<0437:TNYRP>2.0.CO;2)) still largely holds and should be considered.

Figure 3: Use the same vertical scale on both panels (or better yet, plot the two timeseries on the same graph).

251-255: A number of reanalysis products are mentioned here, but the paper also uses others (e.g., ERA-5).

240-269: An overview of the strategy for using the different observational and reanalysis products would be useful. This would demonstrate that there is some overarching reasoning that knits the selection of products together and that has informed the choice of products. I have to say that the choices are really confusing, both for reanalyses and for the observational products. For example, GMST is apparently from GISTEMP (mentioned at line 123, but not in this observational data section), but the gridded global surface temperature dataset that is used is Berkeley Earth (line 242), and other well studied and documented global gridded temperature data products such as HadCRUT4 are not mentioned at all. Why these particular choices? For the gridded products, the infilling strategy and error models, which vary between choices, are presumably important considerations, particularly in the southern hemisphere and especially when considering a relatively small land area in the southern hemisphere that is sandwiched between ocean to the east and a very dry, sparsely observed continent to the west.

270: I find it very surprising that the entire observational discussion for TX7x, including results from AWAP and mention of one of the reanalyses, is limited to only 6 lines of text. Statistical model fitting results are shown in Figure 4, but are really not discussed in any meaningful way – and Figure 4 itself is not explained in a way that most readers would be able to understand. Specifically, cumulative frequency distributions for 1900 and 2019 are shown, but there is no explanation in the text or in the figure caption explaining how the points that are shown are derived from the observations. Evidently observations are adjusted to particular years using smoothed GMST values for those years to make adjustments via the fitted distribution. Shouldn't one be concerned that this could induce some circularity, particularly if one of the intents of the figure is to illustrate the fit of the statistical model to the observations? Results from one reanalysis are mentioned, but silence concerning other reanalyses begs a question about whether they did not "tell a similar story" – do they tell a similar story?

281: See my comment concerning lines 187-189. What explains the apparently much narrower uncertainty bounds on the climate model-based parameter estimates as compared to the model-based estimates? Is the explanation that the model-based analysis actually uses annual blocks rather than blocks constructed by pooling data for a particular simulated "year" across ensemble members (which is literally what lines 187-189 appear to say)? In this case, samples of annual maxima are 5- or 10-times as large as from observations, which, all else being equal, should result in confidence intervals that are about $5^{-0.5}$ or $10^{-0.5}$ as wide as for observations (i.e., ~45% or ~32% as wide, respectively). But this interpretation also doesn't seem quite right because the model confidence intervals seem narrower than these expectations.

I have many largely similar comments about sections 4 and 5 that I won't repeat here. Hopefully the message that the paper needs to document the work and justify choices and interpretations much more carefully has come across.

Regarding Section 6 – a very strong conclusion is drawn on lines 565-566, but it is not obvious to me that the strong quantitative evidence and supporting modelling experiments that would be required for such statement has really been presented. Quantitative evidence seems to be restricted largely to estimates of correlation coefficients which, if considered as simple regression diagnostics (i.e, focusing on $r^2$ rather than r), would correspond to explained variance amounts of the order of 5-15%.

---

## Short Comment (SC2) · 3 Apr 2020

1. The analysis in this manuscript uses a range of observational data, reanalysis products and model simulations. The authors chose to include or exclude data sets based on criteria that are neither fully documented nor properly justified. A transparent approach would be most useful.

For example, two long observation-based datasets (Berkeley Earth analysis and ECMWF's coupled reanalysis of the 20th Century CERA-20C) were dismissed because of their performance during one particular week (in January 1939) out of the 110-year period dating back to 1910. The argument here was that the results were highly sensitive to that one week which might have seen very warm temperatures. CERA-20C is

a reanalysis based on the latest state-of-the-art coupled assimilation techniques which aims to provide a physically consistent estimate of the coupled atmosphere-ocean state and has been demonstrated to perform in general very well. If the results of the presented attribution study crucially depend on a specific week in a specific year, that should warn us about the robustness of the findings and questions the brute dismissal of the CERA-20C data set altogether.

No explanation has been given why ERA-20C, ECMWF's first atmospheric reanalysis of the 20th Century, has been excluded as a dataset.

Section 3.3. says that the ACORN-SAT station data were not available at the time of writing. But they are included in Figures 5-7. Clarifications should be given which data exactly are used in these figures and for which time periods.

For the Fire Weather Index analysis, no other reanalysis products than ERA-5 from 1979 onwards have been used – why? This presents another non-transparent choice of data sets and time periods used.

2. A related question is that of common periods between the data sets. The time periods spanned by the data sets vary considerably with some temperature data starting in the mid-19th Century, some in the early decades of the 20th Century and some only after 1950, see Table 1. The data in the FWI analysis only start around 1980 though. It is not clear to me what the impacts on the results are that stem from artefacts due to analysing data over vastly different periods, e.g. in Figure 5. If temperature trends were calculated using GEV estimates with a linear covariate relationship with global mean temperature, for data like JRA-55 assumptions must have been made for periods before the 1950s – was the temperature increase from 1900 extrapolated? Given the nonlinear nature of the trends from 1900 to today, that seems an overly strong and questionable assumption to me. The finding that ACORN, CERA-20C and ASF-20C reveal a non-stationary relationship between TX7x and GMST (presumably the others don't?) is interesting and would be worth more investigations regarding possible

implications. Perhaps the authors can comment on this.

3. As the main author of the ASF-20C atmospheric seasonal hindcast data set that has been used in this manuscript, I should clarify that the statements made around line 303 about ASF-20C are not correct. ASF-20C is not initialised from ocean reanalysis. Instead, it uses prescribed SST fields and is an atmosphere-only forecast product. The subsequent sentences are misleading at best and provide no justification for the non-stationary relationship between TX7x and GMST, as mentioned in line 300.

In line 370 it is argued that ASF-20C cannot be used for the drought analysis as the data cannot provide annual mean precipitation nor the driest month in a fire season. This is also not correct; ASF-20C does exist for all months of the year (although originating from different initialisation months).

---

## Referee Comment (RC2) · Anonymous Referee #2 · 19 May 2020

Summary comments:

I was really looking forward to reviewing this paper, but it was a terrible disappointment. Overall, this comes across as a very poorly prepared manuscript, possibly related to an effort to do the work quickly and publicize it prior to peer review. The work lacks clear text to describe the analysis and clear and considered messaging of the main results; it lacks a robust assessment of attribution of fire risk is even feasible with current resources; it lacks a considered approach to climate processes relevant to fires in Australia; and it lacks clear, precise and visually appealing figures that help readers to understand the analysis and outcomes. As a reviewer it was an incredibly frustrating experience to assess this manuscript and a clear demonstration of why careful science and the review process should be prioritized over releasing rapid results for media

attention. The topic of this paper is very important, but the very poor job that has been done here undermines efforts of climate science communication and attribution science in general. My recommendation is that this manuscript should be rejected and that the authors should do a much more careful analysis and writing prior to submitting this work again.

Specific comments:

In all cases it is important to state the direction of excursions of the IOD and SAM, as opposite phases have different impacts. E.g. line 17: "...a strong Indian Ocean Dipole from the middle of the year..." should be "...a strong positive Indian Ocean Dipole from the middle of the year..."

Lines 16-18: It isn't correct to describe the IOD and SAM conditions as being in addition ("as well as") to the warmest and driest year on record. These drivers of variability were part of why conditions were so hot and dry, not an additional factor.

Line 20: Was fire activity unprecedented across all of these states individually (that is how it reads), or was the combined level of fire unprecedented? And in what way was it unprecedented? 1974/75 saw a far larger area burnt. 2009 saw a far greater number of lives lost.

Lines 23-41: The information in here is poorly supported by references. It is also not central to the background of this attribution study. Suggest shortening substantially and refocusing on the introduction required for setting up the background for this specific study.

Figure 1: This is a very misleading and inaccurate figure. The size of the dots on this image may misrepresent fire size. The blue shading doesn't only show forested areas of eastern Australia as the caption says. Why is none of Tasmania shade blue? The blue shading seems to be a very loose definition of "forest". Why start this analysis in July 2019 where it will capture many of the normal winter savannah fires which are

unrelated to the fire crisis being discussed here?

Paragraph starting line 45: This is an enormous paragraph that moves between many different concepts. Editing required.

Line 55-76: It is stated that FFDI is the index that is operationally used in Australia. Yet this index isn't used in this study. No explanation is given as to why FFDI isn't used in this study, and this would seem the most sensible choice given that it is the index used for fire danger in Australia.

Line 84: What does "in-situ sites" mean? Some of the examples given in the previous sentence seem pretty small scale (e.g. Brisbane heat during November 2014).

Line 92: How do you take a 7-day moving average of annual maximum temperatures?

Line 96: Why are these two time windows chosen for looking at precipitation, and are these choices defendable? The FFDI uses instead an accumulated precipitation deficit. Process-wise, soil moisture is important in Australia for determining the dryness of living fuels, while temperature and humidity are important for drying of dead fuels, so it isn't clear that the precipitation indices chosen are the best options for relating to fire risk. Also, how is the "fire season" defined?

Figure 2: The "Log(burned area)" is not clear. What are the units? It would also be better to show burned area in true units but give the y-axis a log scale. 2019/20 conditions should be added to figure 2-right, so that readers can see whether an extrapolation of this relationship to 2019/20 is appropriate (line 199). What are the green and grey lines in Figure 2-right?

Section 2.2: Why have an observational data section that then only says that the observation data used are described elsewhere in the manuscript? It would be appropriate to describe all of the data sources, and choices made, here.

Section 2.3: It needs to be described why these models were chosen and why others were not. These choices need to be objective, and the reader needs to know what

these objective choices were. My guess is that the choice is based on starting by selecting all models CMIP5 and CMIP6 models where large ensembles (n>?) exist, but that is just a guess and needs to be explicit in the manuscript. Is single forcing or omitted forcing within these ensembles also a requirement?

Line 156: Are these autocorrelations based on monthly data, annual data, seasonal data?

Line 160: Why 4-year smoothed? What guides this choice of length?

Line 202: Are the model spread and natural variability necessarily independent? Wouldn't this assumption be broken if model spread is at least partly due to differences in how models represent modes of variability. Wouldn't forced changes of the modes of variability (of which there is evidence) also violate this assumption?

Lines 220 to 226: It is notable that neither of the intervals of extreme heat discussed here coincided with the worst fire events that fell around New Years Day. . ..

Lines 217 and 228: Numbers for temperature anomalies in summer 2018/19 are given of 2.61oC and 1.52oC. I think that the second may be related to mean temperature and the first related to maximum temperature – but the text is not clear.

Line 232: What is the threshold behaviour of global warming referred to here?

Figure 3: Is this data averaged for the SE Australia region? Why is there data through to 2020 if this is showing July-June years? This needs to be specified in the caption. It would also be nice if a bit of extra effort went in to making this figure attractive.

Line 258: It is also notable that 1938/39 was a big bushfire year in southeast Australia. Line 271: Make it explicit here that you are talking now about southeast Australia. I was initially confused because earlier a value of 40oC was given for the hottest week, but this was nation-wide.

Lines 286 to 289: Does this then imply that previous studies that have attributed heat

[Figure]

extremes in Australia, and have given quantitative results, are flawed/affected by this same model capability problem? If so, this is an important finding that should be made clearer.

Figure 4-7: How sensitive are these results to the choice of the 4-year smoothing for the global mean temperature dataset? I think that it is important to show results with a longer smoothing that could better account for differences in interannual to interdecadal variability, which will be random in models and cancel out over a large ensemble, compared with the single realisations of observations/reanalysis where the influence of variability will be a real part of the signal.

Lines 301 to 302: Should ACORN be included if there are issues from changing data coverage? Surely then it would be better just to use AWAP for the Australian observational dataset as it specifically resolves this issue.

Line 303: The 1929 event????

Figures 6 and 7 have data labels cut off.

Lines 335-342: This is an area where I have questions over the suitability of FWI for this study, which only considers precipitation over the last 52 days. The FFDI used operationally in Australia uses a longer-term drought index, which is particularly relevant given the multi-year timescale of droughts in Australia and the changes in soil moisture that influence fuel dryness. Multi-year drought was a factor in the 2019 extreme conditions and so should not be ignored in this way.

Line 348: The suitability of only using the lowest 20% or 30% of annual mean precipitation observations needs to be better justified. It is not clear that this is a suitable measure to be using for the attribution testing. Figure 8 still seems to show all annual data, but these should only show the 20% and 30% of data points that are actually being used in the analysis.

Line 350: What is the justification for scaling precipitation relative to smoothed global

mean surface temperature? I understand this for looking at temperature attribution, but it isn't well justified in the text that this should also apply for precipitation.

Figure 8: again text is cut off. Titles are poorly designed. My frustration at the lack of care in preparing the manuscript is rising...

Lines 357 to 359: This long sentence is not well constructed and hard to read/understand.

Figures 5, 6, 7, 9 , 10, 15, 16 and 17 all have unlabelled x-axes.

Section 4.5: The precipitation fails to take into account the seasonal patterns of rainfall change that are well described by Australia's Bureau of Meteorology (increasing warm season rainfall in northern regions, decreasing cool season rainfall in southern regions). Decreases in cool season rainfall may not directly influence the months when fire occurs, but is important in the context of multi-year droughts and soil moisture deficits that influence fuel dryness.

Section 4.6: Reiterating earlier concerns that this analysis doesn't actually do anything related to meteorological drought. In Australia droughts are multi-year events and so aren't and can't be described by a rainfall deficit over the scale of months to a year. The analysis also fails to address the way droughts in this region are being viewed, with southeast Australia being a region that is usually dry (often in drought) and occasionally experiences drought-breaking rain events.

Section 5: Part of the reason why the 2019/20 fires were so devastating was because of the number of extreme fire events where pyrocumulonimbus activity occurred. The metrics here do not account at all for the factors that influence pyroCb risk.

Section 5.2: The equation for DSR is given, but I don't see anywhere the equations given for FWI or for MSR.

Section 5.5, including Line 459 and Figure 16 and 17: Because all models severely underestimate the risk compared to ERA-5, surely this is an indication that accurate

quantification of attributed changes in fire risk is not possible. It seems very misleading to then go on and give percent increases for 2019 and for 2oC warming based on the model output. Similarly, at line 464, it seems unwise to make a statement about the trends being non-significant when there is such disagreement between observations and models, with the models being a severe underestimate.

General comment: I didn't specifically keep track, but it seems that many acronyms are not defined.

Paragraph at line 476: The precipitation contribution can also be seen to be very much underestimated in these models compared with ERA-5. This should not be overlooked.

Paragraph at line 489: The factor of four and factor of nine numbers I think come from the lower end of ERA-5? But these values aren't the focus of the text associated with the interpretation of figure 16 and 17, which instead gave numbers based on models. This is very confusing for readers.

Section 6: This section feels like it was tacked on as an after thought. In particular, these modes of variability are part of the extremes in 2019 and so are already part of the attribution analysis carried out in earlier sections. The way section 6 is written seems like these modes are in addition to the 2019 extreme conditions, which isn't the case.

Section 6.2 and Figure 19: The analysis of the IOD fails to take into account known problems with instrumental data for the Indian Ocean. Generally it is assumed that indices of the IOD are only reliable after 1958, and even after that time there are differences in how well different data products capture the upwelling signal in the eastern upwelling region of the IOD. No information is given about what dataset is used for the analysis in figure 19, so readers can't evaluate potential problems related to the dataset used, though clearly data prior to 1958 is used and this should not be part of any analysis of IOD variability.

Figure 19 and 20: The IOD and SAM have been reported in other studies to influence Tmax, which is also important to fire risk. These should also be shown along with precipitation, and the text should investigate further why this study finds no relationship to Tmax when others have.

Figure 20: ENSO and SAM also interact with a negative correlation, so for consistency in analyses the ENSO-independent SAM relationship should be used.

Line 587: 38 million hectares in 2002/03 – this isn't correct.

Line 589: Need to specify that the 1974/75 fires were grass fires, which have different drivers to forest fires.

Section 7: This section has interesting information, though important points are frequently unreferenced. However, this section reads like a separate study that is unrelated to the focus of this paper on the attribution of Australian bushfire risk to anthropogenic climate change.

Conclusions: I've run out of steam for providing specific comments on the conclusions, but based on my critiques of the previous sections of the paper significant changes are also needed here.

---

## Author Comment (AC1) · 19 May 2020

*In my view, three key questions need to be addressed before we can attribute the bushfire risk to climate change, at least quantitatively.*

We would like to thank Tim Palmer for his comments, which we anticipated and discussed during the writing of the paper. We do not think they substantially influence the validity of the attribution study, as we argue under each point below. We have been very careful in stating our results as a lower bound only, given that the model evaluation revealed the models tend to underestimate trends in the relevant quantities.

1. *The predominant cause of the bushfires were unquestionably the exceptionally (unprecedented) dry conditions in 2019. These were linked to the IOD (and other*

[Figure]

*remote factors). Did climate change makes such dry conditions more likely? A pre-requisite to answering such questions is that we can simulate extremes of the IOD, the regional heat waves and the teleconnnections between the two. Studying 7-day heat waves does not address this issue adequately. We should be looking at model fidelity on monthly to seasonal timescales, where teleconnection biases are known to be substantial. The dry conditions of 2019 persisted into 2020 partly through continuation of the IOD, and partly because of the negative SAM.*

According to the bushfire experts on our team, the Fire Weather Index, which in fact uses the last 52 days of precipitation, is a good predictor of the interannual variability of bush fires, see e.g., Dowdy et al. (2009). This is confirmed by our own analysis shown in Fig. 2 in the manuscript, so it is not 'unquestionable' that the drought of all of 2019 was the predominant cause of the extreme fire season as the reviewer states. The physics behind this is that in summer the upper soil layers are typically dry and any precipitation evaporates quickly, so even an individual dry and hot season has the potential to result in high fire risk. The longer time scale precipitation deficits are important for agricultural and hydrological droughts that depend more on deeper soil moisture. We investigated longer time scales by analysing annual precipitation and half-yearly teleconnections (section 6) in case they would turn out to be relevant for the 2019/20 fire season as well.

For the shorter time scales, a month was chosen as it enabled us to sample a larger set of large ensembles of climate models that only have monthly data available. Previous research (e.g., Hauser et al., 2017) has shown that individual models are unreliable for drought attribution and we thus prioritised investigating as many models as possible.

Statistically, the teleconnection between the Indian Ocean Dipole (IOD; represented by the DMI) and the most sensitive half year of low precipitation, July–

December, which is also the one relevant for bushfires, is only $r = -0.22$ (our Fig. 19), implying that IOD explains only 5% of the precipitation variance. Without clear evidence for non-linearity in this relationship, the historical data hence do not show that the simultaneous occurrence of a high DMI and low precipitation is a general feature of the climate system. The next highest DMI value in 1961 was in fact accompanied by well above average precipitation. For our class-based event definition the historical connection is the important one, not the specific properties of the 2019 event (which will never occur again). This is emphasised in the paper.

For our definition of heat waves, the linear correlation to the September–November DMI index is $r = 0.22$. The time delay between precipitation and temperature response suggests that there is an important role for precipitation, with drier soils enabling higher temperatures. The simultaneous correlation in summer, December–February, between the DMI index and weekly heat is $r = 0.00$. This shows there is no dynamical connection on average from the IOD to heat waves in our study area.

Given that the IOD in the real world only explains a small fraction of the variance of key fire risk factors, model shortcomings in representing the variability or extremes of the IOD in either season are therefore unlikely to explain the factor $\sim 1.7$ difference between observations and models in standard deviation and trends in heat extremes that we found. The correct representation of the IOD and its teleconnections to the heat waves, drought and FWI is therefore of secondary importance to other model uncertainties.

The same holds for the Southern Annual Mode (SAM), although the correlation of $r = 0.39$ with July–December precipitation translates into a higher explained variance, $r^2 = 15\%$. There is no teleconnection to heat extremes. However, in this case we are not aware of literature that describes large discrepancies in trends in the large-scale midlatitude Southern Hemisphere circulation, as this is

exactly the kind of phenomenon weather and climate models were designed to simulate. A check of the properties of extreme low values of the SAM in most models used (we did not have the weather@home pressure fields) shows that except CESM1-CAM5 they indeed have a tail compatible with the one fitted to the ERA5 reanalysis. We already excluded CESM1-CAM5 as unrealistic. We have added this information to the main text:

'We verified that the statistics of 10–100 yr low extremes of the Southern Annular Mode averaged over July–December are represented well in the models used for the attribution analysis except CESM1-CAM5 (scale parameter too small) and weather@home (data not available).'

In summary, we have not investigated the causes of the mismatch between observations and models of a factor almost two in standard deviation and trend in extreme, but the IOD or SAM are unlikely to be the culprits as their contributions to the variability in observations are too small for that. Determining which model deficiencies cause the discrepancy with observations is an active area of research. Some literature suggests that shortcomings in the coupling to land and vegetation in the exchange of heat and moisture with the atmosphere and the representation of the boundary layers are much more likely to be the cause of the problems than large-scale dynamics. Added

'The literature suggests that shortcomings in the coupling to land and vegetation (e.g., Fischer et al., 2007; Kala et al., 2016) and in parametrisation of irrigation (e.g., Thiery et al., 2017; Mathur and AchutaRao, 2019) in the exchange of heat and moisture with the atmosphere, and also in the representation of the boundary layers (e.g., Miralles et al., 2014) are more likely to be the cause of the problems.'

2. *The Fire Index has a dependency on temperature which presumably relates to the fact that vegetation dries out more at higher temperatures. Of course, in general terms this is entirely reasonable. However, at the beginning of summer 2020, the vegetation was already extremely dry due to the exceptional conditions*

*of 2019. It is not clear to me that under these circumstances additional temperature increased the fire risk much further than the critical value it was already at. In this sense the dependence of the fire index on temperature for these exceptionally dry conditions may not be correct.*

The Fire Weather Index has temperature dependencies in its Fine Fuel Moisture Code representing the moisture content of fine fuels and litter on the forest floor, the Duff Moisture Code (DMC) that represents the moisture content of loosely compacted decomposing organic matter and the Drought Code (DC) representing the moisture content of deep compact organic matter of moderate depth (Dowdy et al., 2009). The dryness of the vegetation above ground is not used for the index. We checked if the relation between temperature and the FWI saturates at the high temperatures observed for this event by adding and subtracting 5K in 1K increments. We found no saturation, given that the relation between temperature and FWI was still mostly linear and positive. Added to text: 'We explicitly verified that the dependence of the FWI on temperature is almost linear in a range of $\pm 5$ K around the reanalysis value (not shown).' Further, volumetric soil water (Figure attached) at multiple soil layers from ERA5 suggests that despite the soil already being very dry in 2018 and into 2019, the 2019/20 austral spring-summer drought caused a further drying of the soil in the study area. This indicates that the late 2019 drought and high temperatures did still cause an increase in fire risk.

The Australian Bureau of Meteorology also claims that extreme heat increases bushfire risks, 'The combination of prolonged record heat and drought led to record fire weather over large areas throughout the year, with destructive bushfires affecting all states, and multiple states at once in the final week of the year'(http://media.bom.gov.au/social/blog/2304/hottest-driest-year-on-record-led-to-extreme-bushfire-season/), so we see no grounds to speculate that the temperature dependence has saturated.

3. *The one factor where anthropogenic climate change may have been important, but has not been taken into account here, is the $CO_2$ fertilisation effect, i.e there was simply more vegetation to burn.*

Recent literature from Australia shows that this effect is close to zero (Ellsworth et al., 2017; Jiang et al., 2020).

*Unless we can answer these questions, then I do not think we can, or indeed should, be making quantitative estimates of the impact of anthropogenic climate change on these bushfires.*

We hope to have either satisfactorily answered the questions or shown that the reviewer's concerns do not apply to our attribution statement.

*Instead we must focus effort on developing a next generation of model where the regional dynamical effects of climate change can be simulated with much more confidence than is currently possible (Palmer and Stevens, 2019). Such models are likely to require much greater resolution than we have now - in particular allowing the convective rainfall anomalies associated with the IOD to be represented with the laws of physics rather than with relatively crude parametrisations.*
*Palmer, T.N. and B. Stevens, 2019: PNAS, 116, 24390-24395.*

We very much agree with the reviewer that there is a strong need for a next generation of climate models with sufficiently high resolution to explicitly model the relevant processes, including dynamics, clouds, boundary layer processes, land-atmosphere interactions and vegetation. However, the question is whether we wait with quantitative assessments of the role of climate change in current weather extremes until everybody is sure the models are 'good enough'. We are confident that our careful attribution statement (only lower bounds) holds given our extensive model evaluation. A next generation of high-resolution climate models will hopefully provide a narrower estimate of the role of climate change in such extremes, but these relative careful quantitative statements have in general already proven to be very informative to the general public

and important in decision making processes (e.g., James et al., 2019; Boudet et al., 2020). The timely availability of this particular result made it possible that it was included in the deliberations of the Royal and state commissions set up to investigate the causes of and response to these bushfires. The same uncertainties the reviewer points out are present in many attribution studies and projections generated, e.g., by the IPCC or in other national and regional climate change projections, which also are not suspended until improved climate models are available.

Figure: ERA5 volumetric soil water from multiple levels (1: 0–10 cm, 2: 7–28 cm, 3: 28–100 cm, 4: 100–289 cm, spatial average over the study area.

**References**

Boudet, H., Giordono, L., Zanocco, C., Satein, H., and Whitley, H.: Event attribution and partisanship shape local discussion of climate change after extreme weather, Nature Climate Change, 10, 69–76, https://doi.org/10.1038/s41558-019-0641-3, 2020.

Dowdy, A. J., Mills, G. A., Finkele, K., and de Groot, W.: Australian fire weather as represented by the McArthur Forest Fire Danger Index and the Canadian Forest Fire Weather Index, Technical Report 10, The Centre for Australian Weather and Climate Research, 2009.

Ellsworth, D. S., Anderson, I. C., Crous, K. Y., Cooke, J., Drake, J. E., Gherlenda, A. N., Gimeno, T. E., Macdonald, C. A., Medlyn, B. E., Powell, J. R., Tjoelker, M. G., and Reich, P. B.: Elevated CO2 does not increase eucalypt forest productivity on a low-phosphorus soil, Nature Climate Change, 7, 279–282, https://doi.org/10.1038/nclimate3235, 2017.

Fischer, E. M., Seneviratne, S. I., Vidale, P. L., Lüthi, D., and Schär, C.: Soil Moisture–Atmosphere Interactions during the 2003 European Summer Heat Wave, Journal of Climate, 20, 5081–5099, https://doi.org/10.1175/JCLI4288.1, 2007.

Hauser, M., Gudmundsson, L., Orth, R., Jézéquel, A., Haustein, K., Vautard, R., van Oldenborgh, G. J., Wilcox, L., and Seneviratne, S. I.: Methods and Model Dependency of Extreme Event Attribution: The 2015 European Drought, Earth's Future, 5, 1034–1043, https://doi.org/10.1002/2017EF000612, 2017.

James, R. A., Jones, R. G., Boyd, E., Young, H. R., Otto, F. E. L., Huggel, C., and Fuglestvedt, J. S.: Attribution: How Is It Relevant for Loss and Damage Policy and Practice?, chap. 5, pp. 113–154, Springer International Publishing, Cham, https://doi.org/10.1007/978-3-319-72026-5_5, 2019.

Jiang, M., Medlyn, B. E., Drake, J. E., Duursma, R. A., Anderson, I. C., Barton, C. V. M., Boer, M. M., Carrillo, Y., Castañeda-Gómez, L., Collins, L., Crous, K. Y., De Kauwe, M. G., dos Santos, B. M., Emmerson, K. M., Facey, S. L., Gherlenda, A. N., Gimeno, T. E., Hasegawa, S., Johnson, S. N., Kännaste, A., Macdonald, C. A., Mahmud, K., Moore, B. D., Nazaries, L., Neilson, E. H. J., Nielsen, U. N., Niinemets, Ü., Noh, N. J., Ochoa-Hueso, R., Pathare, V. S., Pendall, E., Pihlblad, J., Piñeiro, J., Powell, J. R., Power, S. A., Reich, P. B., Renchon, A. A., Riegler, M., Rinnan, R., Rymer, P. D., Salomón, R. L., Singh, B. K., Smith, B., Tjoelker, M. G., Walker, J. K. M., Wujeska-Klause, A., Yang, J., Zaehle, S., and Ellsworth, D. S.: The fate of carbon in a mature forest under carbon dioxide enrichment, Nature, 580, 227–231, https://doi.org/10.1038/s41586-020-2128-9, 2020.

Kala, J., De Kauwe, M. G., Pitman, A. J., Medlyn, B. E., Wang, Y.-P., Lorenz, R., and Perkins-Kirkpatrick, S. E.: Impact of the representation of stomatal conductance on model projections of heatwave intensity, Scientific Reports, 6, 23 418, https://doi.org/10.1038/srep23418, 2016.

Mathur, R. and AchutaRao, K.: A modelling exploration of the sensitivity of the India's climate to irrigation, Climate Dynamics, https://doi.org/10.1007/s00382-019-05090-8, 2019.

Miralles, D. G., Teuling, A. J., van Heerwaarden, C. C., and Vilà-Guerau de Arellano, J.: Mega-heatwave temperatures due to combined soil desiccation and atmospheric heat accumulation, Nature Geoscience, 7, 345–349, https://doi.org/10.1038/ngeo2141, 2014.

Thiery, W., Davin, E. L., Lawrence, D. M., Hirsch, A. L., Hauser, M., and Seneviratne, S. I.: Present-day irrigation mitigates heat extremes, Journal of Geophysical Research: Atmospheres, 122, 1403–1422, https://doi.org/10.1002/2016JD025740, 2017.

[Figure]

[Figure]

**Fig. 1.**

---

## Referee Comment (RC3) · Anonymous Referee #3 · 7 Jun 2020

This paper addresses a challenging issue, and provides lots of interesting details. I consider the conclusions drawn from the analysis valid and significant, and eventually the paper must be published.

However, the paper has the character of a technical report. Many technical details make it a barrage of not always relevant information, leaving the reader tired.

For instance, the section 7 is certainly interesting but irrelevant for the issue at hand – which is attribution. It may be relevant in linking the potential for fire to actual fire – but the section is not discussing this limitation of the overall result, but for instance emergency matters

The authors do not provide a clear roadmap of what they do why. For instance, the

attribution to global warming is not really related to the issue of other drivers. What is the idea of attributing how much to global warming, and how much to, say, IOD? Is there an issue with climate change due to changing concentrations of aerosols?

Also, the different data sets – the authors begin with a large list, and then every now and then, one of the data wets is removed (a little like in Agatha Christie's "And then there were none"'). The should say: "we use the sets A,B,C. . ., because they have passed the following quality checks." And they should use these data sets for all strands of attribution.

The PR is referring to two time horizons, this should be made clear by using a term like PR(1900,2020) or so. How are the significance tests other trends done? Often the figure legends are incomplete.

Going into further details is not worth it at this time.

My advice: shorten the body of the paper to 10 pages; have necessary purely technical detail in appendices; do not write it as a progress report, which includes dead end streets like the usage of Berkeley Earth, but build a story which demonstrates that the upfront noted results are plausible. Avoid in-group slang. (Maybe a short summary, of how the regional as well as the global public took the events as manifestation of global climate change, and how the new results fit to these public attributions.)
* * *

---

## Author Comment (AC2) · 8 Jun 2020

1. *The analysis in this manuscript uses a range of observational data, reanalysis products and model simulations. The authors chose to include or exclude data sets based on criteria that are neither fully documented nor properly justified. A transparent approach would be most useful.*

   *For example, two long observation-based datasets (Berkeley Earth analysis and ECMWF's coupled reanalysis of the 20th Century CERA-20C) were dismissed because of their performance during one particular week (in January 1939) out of the 110-year period dating back to 1910. The argument here was that the results were highly sensitive to that one week which might have seen very warm temperatures. CERA-20C is a reanalysis based on the latest state-of-the-art*

*coupled assimilation techniques which aims to provide a physically consistent estimate of the coupled atmosphere-ocean state and has been demonstrated to perform in general very well. If the results of the presented attribution study crucially depend on a specific week in a specific year, that should warn us about the robustness of the findings and questions the brute dismissal of the CERA-20C data set altogether.*

We thank the reviewer for their feedback and for the opportunity to expand on and clarify our dataset choices. We investigated the warm week in January 1939 extensively and are convinced that it is real: the station data (GHCN-D v2 and ACORN) also show the 10 to 15C anomalies that the Australian AWAP dataset shows in this region. There were also extensive bushfires in this region, the 'Black Friday' fires. As an extra check we did a comparison of the January 1939 monthly mean daily mean temperature, for which there are more data sources than daily maximum temperature. We compared the CERA-20C reanalysis with station data, the Australian ACORN analysis and the CRU TS 4.03 analysis. This shows that the CERA-20C land temperatures in this month are indeed unrealistic and should not be used for attribution in this region. We added a new figure showing this to the manuscript (also attached to this reply).

Because it is the warmest one-week event in our index region in the AWAP dataset and the extreme value fits have a negative shape parameter $\xi$, which implies an upper bound to the temperatures, the effect of this one week on the final results is indeed non-negligible. The 1939 value would be above the upper bound if this week were not included in the fit, so including it moves the upper bound up.

The absence of this event in the Berkeley Earth analysis can potentially be explained by the very large decorrelation lengths it employs, which dilute a relatively local anomaly such as this hot week in southeastern Australia (see our new Fig. 4, reproduced as the second figure at the end of this answer). Since the event magnitude is real and including the event is important for defining the GEV fit, we opted to not include datasets that do not show it.

The CERA-20C reanalysis also has an unrealistic cooling trend over 1900–1970 (defined as regression on global mean temperature) over the index region, counter to what the AWAP dataset shows, which points to more significant issues in the reanalysis. Note that the observations assimilated in CERA-20C are surface pressure, mean sea level pressures, surface marine winds, ocean temperature and salinity profiles. They do not include land surface temperature observations, which allows for discrepancies between the near-surface temperatures of the reanalysis and the observations. We added the time series of TX7x in CERA-20C to Fig. 3 to illustrate this issue (our new Fig. 3 attached at the end of this reply).

As a third check, we computed the correlation of the reanalysis time series with the AWAP estimate of observations. This correlation is quite low for CERA-20C at $r = 0.6 \pm 0.1$, contrary to the 20CRv3 reanalysis at $r = 0.8 \pm 0.1$ since 1910. This shows that the magnitude of one-week heat waves in the CERA-20C reanalysis does not correspond well with the magnitude in the observations, again making it unsuitable for attribution analysis. This explanation has been added to the text.

These three points, the value for January 1939, the long-term trends before 1970 and the poor synchronization of interannual variability with observations, all show that CERA-20C does not perform well enough for heat extremes in southeastern Australia to be included in this analysis.

To summarize, while we appreciate the concern for robustness of the presented analysis, we are convinced that the data issues discussed here provide ample justification to exclude certain datasets. We hope that the additional documentation of these data issues serves to convince the reviewer and reader of the validity of our approach and might even be of more general use to illustrate where reanalysis products could be improved for use in event attribution.

*No explanation has been given why ERA-20C, ECMWF's first atmospheric re-analysis of the 20th Century, has been excluded as a dataset.*

Daily ERA-20C maximum temperatures were not available to us for this attribution study and we were advised to use CERA-20C instead. They are also not available from the ECMWF but have to be computed from 3-hourly data, which in turn have to be downloaded programmatically. The whole procedure is time-consuming and error-prone and hence incompatible with a rapid attribution study.

For the revision we did include it, but found that the interannual variability of TX7x averaged over our study region correlates with the AWAP observation-based analysis at only $r = 0.5$ (see attached figure). The January 1939 event is reproduced to some extent in ERA-20C, although the monthly mean of daily mean temperature anomaly during the event is only about half as large as in AWAP, thus showing a substantial underestimation of the event magnitude. We do not know what causes the incoherence of this reanalysis with the observations in this region, but this makes it clearly unsuitable to use as a proxy for the observed record.

*Section 3.3. says that the ACORN-SAT station data were not available at the time of writing. But they are included in Figures 5-7. Clarifications should be given which data exactly are used in these figures and for which time periods.*

Section 3.3 says that ACORN station data are available, but the daily gridded analysis based on the station data, ACORN-SAT, is not. (They are still not available at the time of revision.) We clarified this further in the text.

*For the Fire Weather Index analysis, no other reanalysis products than ERA-5 from 1979 onwards have been used — why? This presents another non-transparent choice of data sets and time periods used.*

The ERA5 reanalysis was the only one that had all the required data up to the end of 2019 at the time of the analysis. We added this explanation to the manuscript.

[Figure]

2. *A related question is that of common periods between the data sets. The time periods spanned by the data sets vary considerably with some temperature data starting in the mid-19th Century, some in the early decades of the 20th Century and some only after 1950, see Table 1. The data in the FWI analysis only start around 1980 though. It is not clear to me what the impacts on the results are that stem from artefacts due to analysing data over vastly different periods, e.g., in Figure 5. If temperature trends were calculated using GEV estimates with a linear covariate relationship with global mean temperature, for data like JRA-55 assumptions must have been made for periods before the 1950s — was the temperature increase from 1900 extrapolated? Given the nonlinear nature of the trends from 1900 to today, that seems an overly strong and questionable assumption to me. The finding that ACORN, CERA-20C and ASF-20C reveal a non-stationary relationship between TX7x and GMST (presumably the others don't?) is interesting and would be worth more investigations regarding possible implications. Perhaps the authors can comment on this.*

There are two places in which the different time periods can play a role: in the model evaluation and in the attribution. For the model evaluation, as in Fig. 5, we compare the scale parameter $\sigma$ and shape parameter $\xi$ of a fit to the observations (or reanalysis) to fits to the model output. In this case the values hardly depend on the time period chosen, in fact we assume that $\sigma$ and $\xi$ are constant over time and check this assumption in the observations and models by comparing the values for different start dates (1910 and 1950).

The differing time periods obviously play a much larger role in the attribution step, in which we estimate the Probability Ratio PR and change in intensity $\Delta I$ from the transient runs for models that do not simulate a counterfactual climate. For convenience we use the smoothed GMST value at 1900 as reference climate. If the data start later we indeed extrapolate from the start date backwards in time to 1900, using the same assumption that we used for the fit. Note that the

GMST trends for pre-1950 dates are small, in particular with regards to the full trend from 1900–2019; the percentage of the 1900–2019 trend value covered by the extrapolated time period is noted for each starting date below. One sees that the extrapolations are small, at most 14% of the full trend in case of JRA, and much smaller for the model data. The period 1902–1920 is cooler than 1900 due to volcanic eruptions of Santa Maria in 1902 and a series of smaller eruptions afterwards lowering the GMST. Extrapolating in time to 1900 involves an interpolation in GMST for starting dates in that interval.

In principle it is possible that the local response to global radiative forcing approximated by the global mean temperature is different for different time periods, as the rise in the early 20th century was due to a combination of increasing concentrations of greenhouse gases, decreasing volcanic activity and increasing solar activity, whereas the rise from the 1970s onward is mainly due to greenhouse gases minus aerosols. These different forcings may well have different footprints locally. However, the only observational datasets to show negative trends over 1910–1970 have been shown to be a bad representation of heat extremes in this region on other grounds (Berkeley Earth, ERA-20C, CERA-20C) or have differing numbers of stations over this time period (ACORN stations). The positive trends in heat extremes agree with summer (DJF) mean temperatures increasing in this region in other datasets (GISTEMP, CRU TS, Berkeley) at a slightly higher rate than global mean temperature over 1910–1970 as well as the full period. To conclude, there are no indications that the negative trends over this period are real.

A final, obvious effect is that shorter time series sample less of the natural variability and give larger uncertainties per ensemble member.

The differing start dates are as follows:

- Observational analyses (AWAP, ACORN stations): 1910, introduction of Stevenson screens. Due to the volcanic eruptions, 1910 is below the full

GMST trend line of 1900–2019, so the interval 1920–2019 covers 126% of the full warming over 1900–2019.

- Reanalyses that do not assimilate land temperature observations (20CRv3): includes 1900, the results do not look reliable in this region for the 19th century so we only use the period 1900-2019.

- The JRA starting data is 1958, the period 1958—2019 covers 86% of the full warming over 1900–2019.

- For EC-EARTH, which has data starting in 1860 we use data from the beginning of the GISTEMP series (1880) and interpolate to 1900.

- The GFDL-CM3 model starts in 1920. This is still below the full GMST trend line of 1900–2019, so the interval 1920–2019 covers 109% of the 1900–2019 warming.

- The CanESM2, GFDL-ESM2M and IPSL-CM6A ensembles start in 1950, this covers 96% of the 1900–2019 warming.

- The ASF20C model starts in 1901, which is very close to 1900.

One sees that the extrapolations are small, at most 14% for JRA and much smaller for the model data. We added a sentence to the text 'The value for 1900 is determined by extrapolating the statistical fit to the data available. Note that this extrapolation represents a small fraction of the total trend due to the majority of GMST change between 1900 and the present day occurring after 1970.'

Note that we already included an analysis with an equal start date of 1950 and thus less sensitive to the potential different effects of GMST changes due to different drivers. The results are very similar to the analysis of the full period, except of course with larger uncertainties due to the shorter time period (see our Fig. 7).

3. *As the main author of the ASF-20C atmospheric seasonal hindcast data set that has been used in this manuscript, I should clarify that the statements made*

*around line 303 about ASF-20C are not correct. ASF-20C is not initialised from ocean reanalysis. Instead, it uses prescribed SST fields and is an atmosphere-only forecast product.*

Our excuses, this has been corrected.

*The subsequent sentences are misleading at best and provide no justification for the non-stationary relationship between TX7x and GMST, as mentioned in line 300.*

These lines have been deleted and replaced by 'ASF-20C is initialised from ERA-20C and run with the same SST dataset as boundary forcing (HadISST.2.1.0.0) (Weisheimer et al., 2017) and hence shows similar weaknesses as this reanalysis. It agrees well with the observations over the satellite era, but has an unrealistic negative temperature trend over the whole century. This is heavily influenced by the too warm years at the beginning, but in fact the trend is negative for start dates as late as 1935 and only becomes comparable to the observations and reanalysis from around 1960 onwards (not shown). We do not know what causes the long-term trends to be at odds with observations.'

*In line 370 it is argued that ASF-20C cannot be used for the drought analysis as the data cannot provide annual mean precipitation nor the driest month in a fire season. This is also not correct; ASF-20C does exist for all months of the year (although originating from different initialisation months).*

Our original text was maybe not precise enough and we have now attempted to further clarify the reasons for our choices. The description of ASF-20C (Weisheimer et al., 2017) does not specify how the reinitialisation works for every month, but we assumed that the new ensemble was not a continuation of the old one. This implies that there are no continuous runs covering a whole or half a year. This might imply discontinuity in slowly varying land boundary conditions such as soil moisture, making the analysis of annual mean or fire season minimum precipitation challenging. We made the text around our argument more

precise: 'The ASF-20C model only has 4-month runs starting four times per year and therefore cannot provide continuous and physically consistent annual or 6-month fire season precipitation, as slowly varying land boundary conditions such as soil moisture are discontinuous every three months when a new ensemble is initialised. Using this dataset without significant further investigation is therefore not straightforward and led us to exclude it from our analysis.'

Figures:

1. The highest 7-day running mean of daily maximum temperature of the July–June year in a) the AWAP analysis, b) Berkeley Earth, c) 20CRv3 reanalysis, d) CERA-20C ensemble mean (DJF maximum) and d) ERA-20C ensemble mean (DJF maximum). The green line indicates a 10-yr running mean. (Expanded Fig. 3 of the original manuscript).

2. The Australian monthly mean daily mean temperature anomalies relative to 1981–2010 in a) GHCN-M v3 station data, b) the ACORN-SAT analysis, c) the CRU TS 4.03 analysis, d) the Berkeley Earth analysis, e) the 20CRv3 reanalysis, f) the CERA-20C reanalysis (ensemble mean) end g) the ERA-20C reanalysis. (New figure 4).

3. Running start date regressions of AWAP, ERA-20C and ASF-20C trends as a regression on smoothed GMST. (Not included in the manuscript.)

**References**

Weisheimer, A., Schaller, N., O'Reilly, C., MacLeod, D. A., and Palmer, T.: Atmospheric seasonal forecasts of the twentieth century: multi-decadal variability in predictive skill of the

winter North Atlantic Oscillation (NAO) and their potential value for extreme event attribution, Quarterly Journal of the Royal Meteorological Society, 143, 917–926, https://doi.org/10.1002/qj.2976, 2017.

[Figure]

a)

**TX7x AWAP**

max 7-day ave Tmax [Celsius]

b)

**TX7x Berkeley Earth**

max 7-day ave Tmax [Celsius]

c)

**TX7x 20CRv3**

max 7-day ave Tmax [Celsius]

d)

**TX7x CERA-20C**

7-day ave Tmax [Celsius]

a)

**anom Jan1939**
station monthly temperature all

b)

**tave-clim8110 Jan1939**
ACORN-SATv2 Tave

c)

**tmp-clim8110 Jan1939**
CRU TS4.03 temperature

d)

**temperature-clim8110 Jan1939**
Berkeley Tavg

e)

**air-clim8110 Jan1939**
20CRv3 2m temperature

f)

**t2m-clim8110 Jan1939**
CERA-20C T2m

g)

**t2m-clim8110 Jan1939**
ERA-20C T2m

**Fig. 2.**

[Figure]

**Fig. 3.**

---

## Referee Comment (RC4) · Célia Gouveia (Referee) · 12 Jun 2020

Review of manuscript

**"Attribution of the Australian bushfire risk to anthropogenic climate change"**

by

**Geert Jan van Oldenborgh, Folmer Krikken, Sophie Lewis, Nicholas J. Leach, Flavio Lehner, Kate Saunders, Michiel van Weele, Karsten Haustein, Sihan Li, David Wallom, Sarah Sparrow, Julie Arrighi, Roop P. Singh, Maarten K. van Aalst, Sjoukje Y. Philip, Robert Vautard, and Friederike E. L. Otto**

This manuscript aims to evaluate if the exceptional fire risk associated with the bushfire during the last months of 2019 and January of 2020 was exacerbated by anthropogenic climate change, namely in the Southeastern Australia, where the fires were particularly severe. The authors analysed the exceptionally of the heatwaves and drought and how they reflected on the Canadian Fire weather Index (FWI). The analysed also, using the current climate models, the long-term trend of the above-mentioned parameter. The driver effect of the Indian ocean dipole and Southern Annular Mode was also assessed. The overall context of the subject is very important in Australia context taking in account the importance of extreme climate events, such as drought and heatwaves, for the region within the context of warming tendency. It should be noted that Australia has a large history of tragic events despite being one the countries (maybe the first one) with better management strategies. Therefore, the work seems to be appropriate for this journal.

However the manuscript is very exhaustive, very hard to follow with an excessive number of figures and details. The analysis is very repetitive and should be organized in a more effective way in order to increase readability. Otherwise, the main achievements will be lost in somewhere.

**MAJOR**

As I said the subject and results of this manuscript are of great interest for a wide range of readers. However, the reading of the paper is very tiring, the number of figures in the manuscript is very high and the results, synthesis, interpretation and and

conclusion for each topic is really tedious. Nevertheless, I recognize that present these results is a very hard task. Therefore, my next comments are suggestion that may increase the readability and increase the number or interested readers that should be attracted to the important results of the manuscript.

1. Some paragraphs of Introduction show a strong lack of references, namely the first ones that have only references to national reports. The same situation occurs in several paragraphs of the introduction and along the manuscript that seems to be more appropriated for a technical report than to a paper.

2. The entire paper should be reorganized. The structure should be less technical and descriptive and more similar to paper structure: Introduction, Data and Methods, Results, discussion and conclusions. Several section should be merged, and the figures reduced significantly in the maintext. The remaining figure should be moved for Supplementary Information.

3. I understand other factor should be included in fire risk analysis. However the present manuscript is so long and the main contribution of the authors are related with Fire Weather Risk. Therefore, I suggest removing section 7 from the manuscript. A paragraph related with the other drivers may be included in Section 8. Conclusions. Consider changing the title accordingly.

4. Clarify if Figure 1 shows the forested areas over the entire Australia or over Eastern Australia. Consider move Figure 1 to Data and Methods.

5. The option of using the Canadian Fire Weather Index (FWI) instead of the FFDI is neither presented, neither justified.

6. Consider comparing the performance of FWI with Forest Fire Danger Index (FFDI) developed and commonly used over Australia for indicating dangerous weather conditions for bushfires.

7. The author used several different datasets for reanalysis the different variables and models. I would prefer to see a less wide lack of reanalysis and gridded datasets. For instance, why do not used precipitation AND temperature from CRU.

8. The author use ERA5 from ECMWF to compute FWI. Why do not use the FWI computed using ERA5 by ECMWF and disseminated already by Copernicus?

9. Data from FFDI using ERA5 computed by ECMWF? Did the authors compare their results for FWI with the ones disseminated by Copernicus?

10. The option of using a window of 7 days for temperature and FWI is not fully presented and justified. Did the author make a sensitivity study to define the 7-days window? Why 7 days and not 5 days? Provide references and justification for the option made, including a comparison with the widely accepted definitions of heatwaves adopted by WMO or based on percentiles.

11. Figures in Figure 3 correspond to averaged values over the southeastern Australia? Information must be presented in figure caption and maintext. Consider moving Figure 3 to Supplementary information.

12. Lines 275 consider to present figure for JRA-55 in Supplementary information.

13. Consider reducing Figures 5-7 to one figure and moving the remaining for Supplementary information.

14. Line 335: Analysis of the driest month in fire season: How must dry is considered month? A difference of 2 mm/day makes has a significant impact on this DMC, DC, FFMC and FWI? The impact of a delta of precipitation in a region with very low values of daily precipitation should be assessed in terms of fire weather risk. Consider providing a sensitivity analysis on this impact.

15. Line 339: please provide quantitative information. What are the observed values and the normal values.

16. Line 351: Annual mean low precipitation analysis: information about the precipitation regime over the region is desirable over a region, i.e., information about inter and intra annual variability of precipitation on the region.

17. Line 366: What are the observed values and the normal values.

18. Consider reducing Figures 9-11 to one figure and moving the remaining for Supplementary information.

19. Lines 411: Why do you use a window of 7 days for FWI and after the MSR instead of the DSR for a window of 7 days.

20. Line 515 did the authors evaluate the formula of DSR for Australia region?

21. Figure 13. In the figure caption describe the information for dots.

22. Consider reducing Figures 15-17 to one figure and moving the remaining for Supplementary information.

**MINOR**

1) (Line 61): parenthesis is missing before 'Clarke'

2) The link for each database should be provided.

3) Figure 8: titles are not completed

---

## Author Comment (AC3) · 16 Jul 2020

*This paper describes an attribution analysis of a number of factors that are known to either contribute to or reflect wildfire risk using observational data, observationally constrained data products (reanalyses), and a collection of CMIP5 model simulations. It also considers the impacts of internal climate variability as reflected in the large-scale modes of variability that influence the Australian climate, and it includes a discussion of vulnerability and exposure factors associated with the impacts of the summer 2019/2020 wildfires.*

*I found the paper frustrating to read and evaluate. One clear impression is that the authors were in a terrible hurry, producing text that often appears not to have been*

[Figure]

*carefully proofread, not thinking carefully about how to describe their methods in a clear way, not always justifying methodological choices, not justifying choices of data products or evaluating those products with a sufficiently critical eye, and attempting to be overly comprehensive. Reading the paper is a bit like being forced to "drink from a firehose" '— there are so many details and so many small aspects that can be criticized, that is difficult to know exactly how and what to criticize in a review. The fact that all code is being made available doesn't really reassure me very much. Readers who want to understand what was done, sufficiently so that the work can be replicated, shouldn't be placed in a position of having to read code but rather, should be provided with explanations in the paper that are clear enough so that they can develop and implement their own code.*

We thank the reviewer for their comprehensive feedback on our paper. Indeed, the article was written in a hurry in order to make the results available in a timely fashion during the aftermath of the fires. For example, the study's findings were used one week after the discussion paper was published in discussions of the various state commissions and Royal Commission at the Bureau of Meteorology on the link between climate change and bush fire risk. During the analysis we focused on making sure the results were correct, which took time away from creating a more carefully written text. We would like to take the opportunity of the revisions to make the article more readable and appreciate the reviewer's detailed comments to that end.

*Some specific comments:*

**1-14** *The abstract does not mention the long section on vulnerability and exposure factors, and there is no reference to vulnerability and exposure in the title. Does that section really belong in the paper?*

Thank you for highlighting this. We have incorporated a reference to the vulnerability and exposure context in the abstract. Generally, many studies conducted by the World Weather Attribution group include a section on vulnerability and

exposure, as some of the co-authors work in this field and help us put the physical results (often focused on physical processes) into context of impacts on the ground.

**16** *The very first sentence of the paper starts by being sloppy in the way in which Australian station data are characterized. The word "homogeneous" has a very clear and well understood meaning in the context of observational data products (i.e., meaning that observations have been carefully evaluated and adjusted to ensure that they are free of artefacts resulting from changes in instrumentation, instrument siting, instrument housing, observing and reporting practices, etc., etc.), and surely the claim here is not that Australian station data is homogeneous in that sense. Clearly, avoiding the obvious inhomogeneity due to the lack of proper instrument shielding early in Australian instrumental record is necessary, but we shouldn't just accept that all of the subsequent record is homogeneous.*

This is a good point. We intended to refer to the introduction of Stevenson screens as standardized measurement equipment, so we have revised this to 'standardized'.

**26-27** *What is the source of this estimate? Is it possible to have any confidence in that number or the range that is given?*

The source is a Red Cross report, accessible via hyperlink. We have changed this to an explicit footnote, so that it becomes clear how to access the source. This report information is collected by the Red Cross, based on official state reports of damaged infrastructure. The loss of wildlife number is an estimate from Prof. Chris Dickman at the University of Sydney (https://www.sydney.edu.au/news-opinion/news/2020/01/08/australian-bushfires-more-than-one-billion-animals-impacted.html), communicated as a 'conservative' estimate based on taking the typical wildlife loss after habitat destruction based on case studies and scaling it to the burned area of this year's bushfires.

**27-29**  *Again, what are the sources?*

We are not aware of peer-reviewed studies yet quantifying economic and health impacts from these fires, but we have added a reference to insurance claims and a general reference to health impacts from wild fires.

**Figure 1**  *Is there a URL and a date for where this image was obtained?*

We made the image ourselves using MODIS satellite data (obtained from https://firms.modaps.eosdis.nasa.gov/) for the fire radiative power and Dinerstein et al. (2017) data for the forest cover. References to the data sources have been added.

**93**  *I imagine daily maximum temperatures are meant.  There are many instances in the paper where a second reading of the words, just to see if the connect logically, would have helped enormously.  There are also a large number of run-on sentences in the paper that are difficult for readers to parse and understand.*

We clarified that this refers to daily maximum temperatures.  We will revise the entire paper for readability.

**102**  *This subsection is entitled "Event definition", but it doesn't talk specifically about event definition at all. I think what is needed is a clear statement that the event of interest will be defined using the FWI. This section gives some justification for doing that by considering the relationship between FWI and area burned, but event definitions per se are not discussed in this subsection.*

We have substantially revised this subsection to clarify that it mainly deals with general parameters of the event definition (such as fire season, spatial domain, way of aggregating), while specific event definition details for each driver variable (temperature, precipitation, fire indices) are given at the beginning of their respective sections later on in the paper.

**Figure 2 caption** *Please tell me what is meant by a "one-sided confidence interval about zero". I assume you mean the interval from -1 to the expected 95th quantile for the correlation coefficient under the null hypothesis that the correlation is zero. If this is correct, then it would be better to call this the 5% significance critical value for a one-sided test of the null hypothesis that the correlation is zero against the alternative hypothesis that the correlation is positive.*

We have revised the caption, including the wording suggested by the reviewer, which was helpful and greatly appreciated.

**129** *Often, acronyms like ASF20C appear before they are defined.*

We thank the reviewer for flagging this. We have scanned the whole text to remove instances of this problem.

**137-152** *Some careful justification for the distributional choices would seem to be in order. These distributions emerge in statistical extreme value theory as limiting distributions under idealized conditions, where the limit is taken either as block length increases without bound in the case of the GEV, or as the exceedance threshold increases without bound in the case of the GPD. Given the way the data are processed, we are likely a long way from being able to be satisfied that the actual distributions are well approximated by these limiting distributions. Indeed, it seems likely that the relative quality of the fit will diminish as you go deeper into the tail, even if quantile plots look to be ok. In particular, one should be worried about extrapolating beyond the available data. Some aspects of this are discussed later in the paper, but those limitations don't really seem to prevent the authors from referring to values that appear to correspond to very long return periods in some instances. In the case of precipitation deficit, any of a number of possible candidate distributions could presumably be considered if using as much as 30% of sample values. These would have different deep tail characteristics, affecting calculations of probability ratios, but might not be discernably better or*

*worse than the GPD based on standard diagnostics of the fit. So how does one proceed in a careful way take this source of structural uncertainty into account? It might be as important as the structural uncertainty represented by the spread between models.*

We agree that this is an important point and have spent quite some time in the past investigating which distributions fit the data well enough for common extremes. In this case we decided to use the following for the three types of extremes we study.

**Heat extremes** The highest weekly average of the year is a block maximum. Even though the number of independent blocks in a summer is small, as the reviewer mentions, a GEV distribution fits the observational data well, see Fig. 4. The uncertainties are indeed rather large. More convincing evidence is the agreement in models with large numbers of simulated years, all of which show no deviations from the GEV fits in the return time plots up to the return times of a few hundred years that we use. CESM (4000 years of data) shows a thinner tail for the highest values, whereas CanESM (3500 years of data) shows a fatter tail above 1000 yr return times, EC-Earth and GFDL-ESM2M (2000 years each) are described well by the GEV. We assume that the very hottest events involve nonlinear physics, but apparently the models do not agree which sign the deviation from the GEV should be. However, we evaluate the distribution at 100 years for the current climate, so we use the distribution to interpolate, not extrapolate, and the differences in the PDFs are mainly due to the (considerable) differences in the parameters of the function and not to deviations from the assumed form of the GEV. Because of the large uncertainties in the tails of these distributions with negative shape parameters we take care to never claim much accuracy of the numbers coming out of the fits anyway.

**Drought extremes** We verified that these are not described well by either a normal

distribution nor by a gamma distribution (indeed, we have not found a single case where a gamma distribution described precipitation data well enough). The GPD appears to fit the data well enough, partly due to the extra fit parameter that gives more flexibility. The low number of data points and hence strong dependence on the threshold in the observational case does limit its usefulness. We have attempted to take this uncertainty into account by using both the 20% and 30% threshold results. In this case, model experiments with larger numbers of data points show that the GPD fits the data well up to the return times of thousands of years, which are actually sampled in the climate model cases. Note that we evaluate the distribution at a return time of only around ten years.

The functional form of the dependence of the covariate is just a convenience that makes sure we never get negative values, there is no theoretical justification for assuming Clausius-Clapeyron scaling for droughts. However, as the changes with the covariate are small (in fact, mostly compatible with zero), the exponent is close to a linear function anyway and this choice allowed us to use a ready-made routine. The main problem was technically fitting the function in a parameter space where only a very small area satisfied the constraints of no negative precipitation and we had to change to a different version of the simplex minimisation routine we used to do the log-likelihood maximisation in order for the fit routine to find the maximum (the GSL version could not do it).

**Fire Weather Index** We again use block maxima, like in the heat extremes analysis. The GEV fits all models well in a return time plot (equivalent to a Q-Q plot), but with very different parameters. We again conclude that the differences between the distributions of the different models are dominated by the parameter differences and not by deviations from the GEV.

As for droughts, there is no theoretical justification for the functional form of the dependence on the covariate (smoothed global mean temperature)

beyond the requirement of a positive-definite distribution (which would be violated with a simple linear dependence that is used in many other articles).

We have added a summary of these arguments into the article and supplementary material.

**160** *Why 4-years and not some other degree of smoothing? Exactly how is the smoothing done, and how is time referenced to the smoothed values? For example, if using a 4-year running mean, which year is the value associated with in covariate dependent functions?*

The 4 years are a previously developed compromise (King et al., 2015) between the typical time scales of ENSO and more decadal-scale variability and a well-defined value at the end of the record. The smoothing is a running mean. Year 3 of 4 is used as covariate. Critically, uncertainties from these choices are typically very small. We have added clarifications on all of these points.

**165-172** *Choices for how the GEV and GPD distributions are parameterized should be justified and carefully argued, not just stated. For precipitation, exponential scaling might make sense at the upper end of the precipitation distribution, but why would I consider that to be reasonable at the lower end of the distribution, and why, in that case, should the scale parameter be linked to GMST? Building in something that scales like Clausius-Clapeyron might not be the best idea for the dry end of the precipitation distribution.*

See the answer to the comment on lines 137–152 above.

**187-189** *Is it obvious that this is the best way to proceed? If the analysis was literally performed as described here, the effective block size for the models would be 5- or 10-times the block size used for the observations. That means that for the models, the block maxima used to fit GEV distributions would sample a much deeper part of the tail than is possible with the observations since the distributions*

*for the model output would have been fitted to what are effectively 5-year or 10-year blocks rather than 1-year blocks as for the observations. How then, can I make sense of differences in parameter estimates between fits to observed and simulated parameter estimates.*

Indeed it would have been better if both fits would have been done with the same block size. However, this is not possible due to the paucity of data in the reanalysis and the functional form of the model data not agreeing with a GEV function all the way down to very low return times in one climate model that is used in the final synthesis (CanESM2, the others have distributions that are compatible with a GEV for all return times). The description in the methods section has been updated to mention that it only refers to one model. The method described is the one that gave a good fit in the region of the return time of interest, 31 years. The main goal of the method, to establish lower bounds on the probability ratio, can therefore be achieved with it.

Regarding the test on the fit parameters that the reviewer comments on, indeed the return times for which these are defined differs between observations and models when excluding the shortest ones (below 5 yr). However, we are interested in return times, around 31 years in the current climate, which correspond to longer return times in the climate of 1900. The assumption we make is that the data in this range are described well by the GEV we fitted to. We do not use the range of return times below 5 years that are excluded in the CanESM2 fit. The parameter values can therefore still be compared to the 'observational' ones.

**191-192** *I think this is all that is said about bias correction in the paper except for another brief mention at line 442, but surely this is important and should be discussed (and defended) in some detail. Exactly what was done, and how does this avoid overusing the observational data?*

All the bias correction we do is to evaluate the extreme value function at the same return time as the return time of the observational analysis. As the reviewer

comments, it is very important not to overuse the observational data so we do not attempt to correct the PDF further than this one-parameter correction (see also the reply to Tim Palmer's comment). The usual minimal bias correction in climate model analys is a correction of the mean, which is implicit in all IPCC change plots. We found that the effective inclusion of some effects of the biases in variability and skewness by evaluating at the same return time rather than just shifting the mean gives more realistic results, especially in heat extremes. These have distributions with a negative shape parameter in which the return time is very sensitive to the return value for which it is evaluated. Evaluating at the same return time as the observations removes this sensitivity. We have updated the bias correction description to better reflect this.

**200** *Exactly what do you mean by the $\chi^2$/dof statistic (what is calculated, and what is the basis for the interpretation given to this statistic)?*

The goal is to determine whether the apparent uncertainty across models is just due to internal variability or whether it is indeed indicative of actual model structural differences. To that end, we simply compare the spread of the model results with the spread expected from their estimate of uncertainty due internal variability $\Delta x$, $\chi^2 = \sum[(x - \overline{x})/(\Delta x)]^2$. This should be roughly equal to the number of degrees of freedom, $N - 1$. If it is larger, we interpret this as evidence that we have to take into account another source of uncertainty, the model spread (which is part of the model uncertainty). The same method is used by Aurélien Ribes et al. (2020), although he agrees with us that it is a rough estimate as any discrepancy could still be due to chance. However, it is the best we can do given the information available. We have extended the discussion in the methods section with the explicit formula and more details.

**240** *For each observational product, the paper should draw attention to the key limitations that would affect the analysis in this paper. For example, although Stevenson screens begin to be used in 1910, there could be many other reasons to*

*be concerned about the homogeneity of temperature observations, such as variations in station coverage over time (e.g., spatial sampling in 1910 would undoubtedly have been different than in the 1970s). Also, the paper should make a clear distinction between observational products on the one hand, and observationally constrained products (re-analyses) on the other. The latter are clearly non-homogeneous, with inhomogeneity due to changes in data sources, quality and quantity over time being of particular concern in the southern hemisphere where the observational constraint is much weaker. Ensemble reanalysis products, such as the 20th century reanalysis may be able to provide information about the strength of the observational constraint and how it varies in space and time (if the spread between ensemble members is large, the constraint is obviously weak or non-existent; if the spread is small, one has further work to do to determine if it is small because the analysis is being effectively constrained by the observations or whether this is coming about for another reason). Further, it should be noted that surface variables are often not very well constrained in reanalyses. The classification of variables by strength of observational constraint that is given in Appendix A of the Kalnay et al paper describing the original NCEP 40-year reanalysis (BAMS, 1996, https://doi.org/10.1175/1520-0477(1996)077<0437:TNYRP>2.0.CO;2) still largely holds and should be considered.*

This is certainly a valid point and we attempted to provide more justification for our dataset choices in response to this and other reviews received.

We have added the number of stations entering the analyses to the information already given. Beyond that, the only information on the inhomogeneities would come from the daily gridded ACORN dataset that has been corrected for this, but unfortunately it is not yet available.

We have indeed attempted to distinguish between observational datasets and reanalyses. JRA-55 is the only dataset with adequate coverage in the southern

hemisphere before 1979. We have added a section discussing the quality of the longer reanalyses, showing that the ECMWF products ERA-20C and CERA-20C indeed are unusable in this region. The 20CRv3 reanalysis does surprisingly well despite the limited constraints in the early part of the record, but is not used in the analysis either given the availability of good observational datasets.

We disagree that the classification in Kalnay et al. (1996) is still useful. Modern models are good enough to assimilate near-surface temperature observations and hence reanalyses like JRA-55 and ERA5 have become much better at simulating these variables than NCEP/NCAR R1. This obviously does not hold for long-term reanalyses (ERA-20C, CERA-20C, 20CRv3), of which only 20CRv3 approaches a reasonable simulation of the weekly maximum temperature extremes in southeastern Australia.

**Figure 3** *Use the same vertical scale on both panels (or better yet, plot the two time-series on the same graph).*

At the request of Antje Weisheimer we have expanded Fig. 3 to five panels so showing all lines in one plot is not an option anymore. We have replotted all these panels on the same vertical scale (attached).

**251-255** *A number of reanalysis products are mentioned here, but the paper also uses others (e.g., ERA-5).*

This section only refers to the reanalyses that were used in the heat attribution. ERA5 is not used there due to its brevity and availability of better (longer) alternative datasets. For the FWI, where more variables are required, we do rely on ERA5, despite its brevity, as explained in the relevant section. The results from ERA5 for the hottest week of the year are compatible with the other observational and reliable reanalysis datasets. There is a higher trend over the period since 1979, but the difference with the long-term trends from the century-scale datasets is not statistically significant due to the larger uncertainties resulting

from the lower number of data points. We have included this in text. We do not show this in the summary plots because the period over which the trend is estimated is so different from the observational datasets and models.

**240-269** *An overview of the strategy for using the different observational and reanalysis products would be useful. This would demonstrate that there is some overarching reasoning that knits the selection of products together and that has informed the choice of products. I have to say that the choices are really confusing, both for reanalyses and for the observational products. For example, GMST is apparently from GISTEMP (mentioned at line 123, but not in this observational data section), but the gridded global surface temperature dataset that is used is Berkeley Earth (line 242), and other well studied and documented global gridded temperature data products such as HadCRUT4 are not mentioned at all. Why these particular choices? For the gridded products, the infilling strategy and error models, which vary between choices, are presumably important considerations, particularly in the southern hemisphere and especially when considering a relatively small land area in the southern hemisphere that is sandwiched between ocean to the east and a very dry, sparsely observed continent to the west.*

We will add more structured text to explain the dataset choices, as this has been a consistent concern across reviews.

The heat events are defined as the maximum of 7-day maximum temperature, so these can only be extracted from a dataset with daily resolution. Neither GIS-TEMP nor HadCRUT4 has daily resolution, these are monthly datasets with very coarse resolution designed to study global and large-scale temperature changes. The only place were these are useful is as a the covariate in the extreme value function that acts as a proxy for the radiative forcing of global warming. For this we take the (smoothed) GISTEMP global mean temperature, as HadCRUT4 un-derestimates the warming trend due to its undersampling of the polar regions (although the differences are so small that in practice it makes very little differ-
ence).

As we mention in the text, there is only one global long-term daily maximum temperature dataset that we are aware of, Berkeley Earth. We checked this dataset against local Australian datasets of daily maximum temperature, of which the AWAP dataset is a full analysis, which interpolates the station data in space. The ACORN station dataset does not, but provides a useful qualitative check. Reanalyses of maximum temperature are somewhat less reliable but again serve as useful checks for the observational datasets. As the reviewer mentioned, some are not reliable in the southern hemisphere before the satellite era.

We revert to monthly and daily mean data to check whether the January 1939 event is real or not, but prefer to use datasets with high spatial resolution for this. The $5 \times 5°$ resolution of HadCRUT4 is not sufficient to resolve this relatively small-scale event.

We have clarified this discussion in the text.

**270** *I find it very surprising that the entire observational discussion for TX7x, including results from AWAP and mention of one of the reanalyses, is limited to only 6 lines of text. Statistical model fitting results are shown in Figure 4, but are really not discussed in any meaningful way '— and Figure 4 itself is not explained in a way that most readers would be able to understand. Specifically, cumulative frequency distributions for 1900 and 2019 are shown, but there is no explanation in the text or in the figure caption explaining how the points that are shown are derived from the observations. Evidently observations are adjusted to particular years using smoothed GMST values for those years to make adjustments via the fitted distribution. Shouldn't one be concerned that this could induce some circularity, particularly if one of the intents of the figure is to illustrate the fit of the statistical model to the observations? Results from one reanalysis are mentioned, but silence concerning other reanalyses begs a question about whether they did not "tell a similar story" '— do they tell a similar story?*

The same method has been used numerous times in previously published literature going back to van Oldenborgh et al. (2015) and King et al. (2015). We might have assumed that most readers would be familiar with this approach and have thus omitted some of the details. Indeed the observations have been shifted to the climate of 2019 and 1900 using the functional dependence on the smoothed global mean surface temperature. We have added references to earlier work using these methods to the methods section and a more complete explanation of the figure to the caption.

We are not sure how this can induce some circularity: the 4-year smoothed annual mean global temperature is not affected in any meaningful way by the local highest one-week maximum temperature in southeastern Australia.

We did not want to discuss the 20CRv3 results quantitatively as it does not assimilate near-surface temperature observations and therefore cannot be expected to reproduce these reliably. We added a sentence discussing this: 'As the 20CRv3 reanalysis does not assimilate near-surface temperatures we do not expect the quantitative results for TX7x to reflect reality, but note that qualitatively they agree well with the other estimates of the observations.' The other long reanalyses have been rejected as unreliable at this stage.

**281** *See my comment concerning lines 187-189. What explains the apparently much narrower uncertainty bounds on the climate model-based parameter estimates as compared to the model-based estimates? Is the explanation that the model-based analysis actually uses annual blocks rather than blocks constructed by pooling data for a particular simulated "year" across ensemble members (which is literally what lines 187-189 appear to say)? In this case, samples of annual maxima are 5- or 10-times as large as from observations, which, all else being equal, should result in confidence intervals that are about $5^{-0.5}$ or $10^{-0.5}$ as wide as for observations (i.e., $\sim 45\%$ or $\sim 32\%$ as wide, respectively). But this interpretation also doesn't seem quite right because the model confidence intervals*

*seem narrower than these expectations.*

First, we assume the reviewer means 'observation-based' instead of 'model-based' in his second sentence.

In this case, all fits are based on annual maxima, larger blocks are only used for the Fire Weather Index (FWI) and Monthly Severity Rating (MSR). Indeed, the models have more ensemble members than the one realisation of the real world, leading to smaller uncertainties due to natural variability. The number of ensemble members per climate model is listed in Table 1 and varies from 15 to 50 for the models with daily maximum temperatures. They also have differing record lengths, from 55 to 170 years compared to the 110 years of the AWAP dataset. For models with reasonable variability, which we have checked in the model validation section, this gives uncertainties due to natural variability of about $\sqrt{110/65}/\sqrt{15} \approx 34\%$ (HadGEM3-A) to $\sqrt{110/70}/\sqrt{50} \approx 18\%$ (CanESM2) of the observational estimates, which agrees by eye with the length of the bars in Figure 5.

We have added a clarifying sentence to this end.

*I have many largely similar comments about sections 4 and 5 that I won't repeat here. Hopefully the message that the paper needs to document the work and justify choices and interpretations much more carefully has come across.*

We have updated these sections with similar clarifications as the heat event attribution section.

*Regarding Section 6 '— a very strong conclusion is drawn on lines 565-566, but it is not obvious to me that the strong quantitative evidence and supporting modelling experiments that would be required for such statement has really been presented. Quantitative evidence seems to be restricted largely to estimates of correlation coefficients which, if considered as simple regression diagnostics (i.e, focusing on $r^2$ rather than $r$), would correspond to explained variance amounts of the order of 5-15%.*

We estimated the contribution of the IOD and SAM from the scatter plots in figures 19 and 20 as the fraction of the anomaly of 2019 described by the (purple) linear trend line from zero DMI-ENSO or SAM to the value observed in 2019, which in both cases is about one third. Mathematically the explained fraction is

$$\frac{dP}{dI}\Delta I/\Delta P \approx \frac{1}{3}, \tag{1}$$

with $P$ the July–December precipitation and $I$ the DMI-ENSO or SAM index. However, we are deliberately unspecific about exact numbers given the large approximations in the value of the relationship and the linearity of it as the reviewer points out. In response, we weakened the statement in the paper to 'More quantitative estimates will require further analysis and dedicated model experiments, as the linearity of the relationship between these indices and the regional climate is not verifiable from observations alone.'

**References**

Dinerstein, E., Olson, D., Joshi, A., Vynne, C., Burgess, N. D., Wikramanayake, E., Hahn, N., Palminteri, S., Hedao, P., Noss, R., Hansen, M., Locke, H., Ellis, E. C., Jones, B., Barber, C. V., Hayes, R., Kormos, C., Martin, V., Crist, E., Sechrest, W., Price, L., Baillie, J. E. M., Weeden, D., Suckling, K., Davis, C., Sizer, N., Moore, R., Thau, D., Birch, T., Potapov, P., Turubanova, S., Tyukavina, A., de Souza, N., Pintea, L., Brito, J. C., Llewellyn, O. A., Miller, A. G., Patzelt, A., Ghazanfar, S. A., Timberlake, J., Klöser, H., Shennan-Farpón, Y., Kindt, R., Lillesø, J.-P. B., van Breugel, P., Graudal, L., Voge, M., Al-Shammari, K. F., and Saleem, M.: An Ecoregion-Based Approach to Protecting Half the Terrestrial Realm, BioScience, 67, 534–545, https://doi.org/10.1093/biosci/bix014, 2017.
Kalnay, E., Kanamitsu, M., Kistler, R., Collins, W., Deaver, D., Gandin, L., Iredell, M., Saha, S., White, G., Woollen, J., Zhu, Y., Leetma, A., Reynolds, R., Chelliah, M., Ebisuzaki, W., Higgens, W., Janowiak, J., Mo, K. C., Ropelewski, C., Wang, J., and Jenne, R.: The

NCEP/NCAR 40-year reanalysis project, Bull. Amer. Met. Soc., 77, 437–471, https://doi.org/ 10.1175/1520-0477(1996)077<0437:TNYRP>2.0.CO;2, 1996.

King, A. D., van Oldenborgh, G. J., Karoly, D. J., Lewis, S. C., and Cullen, H.: Attribution of the record high Central England temperature of 2014 to anthropogenic influences, Environmental Research Letters, 10, 054 002, https://doi.org/10.1088/1748-9326/10/5/054002, 2015.

Ribes, A., Thao, S., and Cattiaux, J.: Describing the Relationship between a Weather Event and Climate Change: A New Statistical Approach, Journal of Climate, 33, 6297–6314, https://doi.org/10.1175/JCLI-D-19-0217.1, 2020.

van Oldenborgh, G. J., Haarsma, R., De Vries, H., and Allen, M. R.: Cold Extremes in North America vs. Mild Weather in Europe: The Winter of 2013–14 in the Context of a Warming World, Bulletin of the American Meteorological Society, 96, 707–714, https://doi.org/10.1175/ BAMS-D-14-00036.1, 2015.
* * *
[Figure]

Fig. 1.

---

## Author Comment (AC4) · 26 Oct 2020

*This paper addresses a challenging issue, and provides lots of interesting details. I consider the conclusions drawn from the analysis valid and significant, and eventually the paper must be published.*

*However, the paper has the character of a technical report. Many technical details make it a barrage of not always relevant information, leaving the reader tired.*

We thank the reviewer for their review and acknowledge that the text in particular was not easy to read. This is a comment common to all the reviews received, so it justifies and motivates a thorough revision and restructuring of the text, which we will undertake.

*For instance, the section 7 is certainly interesting but irrelevant for the issue at hand*

*— which is attribution. It may be relevant in linking the potential for fire to actual fire — but the section is not discussing this limitation of the overall result, but for instance emergency matters.*

Thank you for highlighting this. Generally, many studies conducted by the World Weather Attribution group include a section on vulnerability and exposure, as some of the co-authors work in this field and help us put the physical results (often focused on ocean-atmosphere processes) into context of impacts on the ground. We consider that a disaster is not only caused by extreme meteorology (that may be influenced by anthropogenic climate change), but also by how exposed and vulnerable people are. Trends in those factors may be harder to quantify but also contribute to the overall change in risk. They are also where some of the solutions can be found as society has clearly chosen a path of both mitigation and adaptation. We therefore would like to keep the content. However, we recognise that Section 7 is too long, in particular given the length of the rest of the paper. Therefore, we will condense Section 7 and attempt to cross-reference it better in the rest of the paper.

*The authors do not provide a clear roadmap of what they do and why. For instance, the attribution to global warming is not really related to the issue of other drivers. What is the idea of attributing how much to global warming, and how much to, say, IOD? Is there an issue with climate change due to changing concentrations of aerosols?*

The method we use to attribute the increase in fire weather risk to anthropogenic climate change (documentation to appear as Philip et al. (2020)) includes all pathways, including through shifts of modes of variability such as the IOD. Everything but the trends is considered 'noise', including the unforced variability of potentially forced changes of these modes. Splitting up the trends or noise into modes and other factors is a scientifically interesting question, but is not relevant for the overall attribution statement we derive in this analysis.

Concerning aerosols, these are also included in the analysis as part of the anthropogenic forcing if they give rise to changes in event probability from one time period to another. Generally, these effects are much smaller in the southern hemisphere compared to the northern hemisphere. The ozone forcing has a stronger impact, but again has been included in the model runs we use.

Another reviewer suggested to better justify, but also shorten the section on modes of variability and so we will revise this section to be clearer and more concise, including answers to above questions.

*Also, the different data sets — the authors begin with a large list, and then every now and then, one of the data sets is removed (a little like in Agatha Christie's 'And then there were none'). The should say: 'we use the sets A,B,C..., because they have passed the following quality checks.' And they should use these data sets for all strands of attribution.*

This comment is indeed consistent with the review comment by Antje Weisheimer. We will revise the data set discussion to make it easier to see what we used and why. However, we added century-scale daily temperature and monthly precipitation datasets to be able to give much more reliable statements on trends in heat extremes and drought than would be possible on the basis of just the 1979–2018 period, which is the period over which we can reliably compute the Fire Weather Index. So there are some justifiable deviations from using a fixed collection of datasets for all strands of the analysis. Limiting ourselves to a consistent collection of datasets would possibly greatly reduce the robustness of the conclusions for some attribution statements, which are currently based on multiple datasets and lines of evidence on multiple time scales.

*The PR is referring to two time horizons, this should be made clear by using a term like PR(1900,2020) or so. How are the significance tests other trends done? Often the figure legends are incomplete.*

We have made sure in the revised text that the time horizon is always included with the PR.

The significance tests are done using a 1000-member non-parametric bootstrap method, just taking the 2.5% and 97.5% percentiles as representing the 95% confidence interval.

We have gone through the figure captions to make sure they are more complete.

*Going into further details is not worth it at this time.*

We hope that by addressing these comments and those of the three other reviewers plus the two comments we have addressed all the concerns of the reviewer.

*My advice: shorten the body of the paper to 10 pages; have necessary purely technical detail in appendices; do not write it as a progress report, which includes dead end streets like the usage of Berkeley Earth, but build a story which demonstrates that the upfront noted results are plausible. Avoid in-group slang. (Maybe a short summary, of how the regional as well as the global public took the events as manifestation of global climate change, and how the new results fit to these public attributions.)*

We have taken this comment to heart and will revise and restructure the paper with a main part and supplementary information.

Philip, S. Y., Kew, S. F., van Oldenborgh, G. J., Otto, F. E. L., Vautard, R., van der Wiel, K., King, A. D., Lott, F. C., Arrighi, J., Singh, R. P., and van Aalst, M. K.: A protocol for probabilistic extreme event attribution analyses, Advances in Statistical Climatology, Meteorology and Oceanography, accepted, 2020.

---

## Author Comment (AC5) · 4 Nov 2020

**1   Summary comments**

*I was really looking forward to reviewing this paper, but it was a terrible disappointment. Overall, this comes across as a very poorly prepared manuscript, possibly related to an effort to do the work quickly and publicize it prior to peer review. The work lacks clear text to describe the analysis and clear and considered messaging of the main results; it lacks a robust assessment of attribution of fire risk is even feasible with current resources; it lacks a considered approach to climate processes relevant to fires in Australia; and it lacks clear, precise and visually appealing figures that help readers to understand the analysis and outcomes. As a reviewer it was an incredibly frustrating*

[Figure]

*experience to assess this manuscript and a clear demonstration of why careful science and the review process should be prioritized over releasing rapid results for media attention. The topic of this paper is very important, but the very poor job that has been done here undermines efforts of climate science communication and attribution science in general. My recommendation is that this manuscript should be rejected and that the authors should do a much more careful analysis and writing prior to submitting this work again.*

We thank the reviewer for their honest feedback and for providing a comprehensive and thorough review despite the frustration. Indeed, the article was written relatively quickly in order to make the results available in a timely fashion during the aftermath of the fires. For example, the study's findings were used one week after the discussion paper was published in discussions of the various state commissions and Royal Commission at the Bureau of Meteorology on the link between climate change and bush fire risk, where the main messages were generally perceived as helpful and later accepted by for instance the NSW government. Our colleagues from the Red Cross / Red Crescent Climate Centre also highlighted repeatedly that a timely study is extremely helpful to inform decision-makers as windows of political opportunity often open in the brief timespan after the event but may have closed again by the time a peer reviewed study could be available. It is in the light of these windows of opportunity and with the goal to equip those decision-makers with scientific evidence that a rapid time frame becomes necessary. It of course is never thought of or aimed to replace peer-review. And in fact, this manuscript crucially builds on many peer-reviewed studies that carefully describe the methodologies that have been developed by the community.

However, it is indeed clear that we have failed to clearly highlight in the text where we use exactly the same methods explained elsewhere and clearly signpost where to find them. The current analysis is focused too exclusively on the results. Given, however that those are solid, novel and have already proven extremely useful in decision-making contexts, we are convinced that we can revise the manuscript to satisfy the concerns

by, inter alia, highlighting assumptions and their justifications and more carefully describing the methodology and clearly highlight where in the peer-reviewed literature these have been applied before with broad acceptance from the scientific community. We very much appreciate the reviewer's detailed comments which helped to identify where the reasoning is not transparent enough. We found that these comments in the end did not affect any of our conclusions.

**2  Specific comments**

- *In all cases it is important to state the direction of excursions of the IOD and SAM, as opposite phases have different impacts. E.g. line 17: "...a strong Indian Ocean Dipole from the middle of the year..." should be "...a strong positive Indian Ocean Dipole from the middle of the year..."*

  We have revised all such instances in the revised manuscript.

- *Lines 16-18: It isn't correct to describe the IOD and SAM conditions as being in addition ("as well as") to the warmest and driest year on record. These drivers of variability were part of why conditions were so hot and dry, not an additional factor.*

  Thank you for pointing this out. We have corrected this in the revised version.

- *Line 20: Was fire activity unprecedented across all of these states individually (that is how it reads), or was the combined level of fire unprecedented? And in what way was it unprecedented? 1974/75 saw a far larger area burnt. 2009 saw a far greater number of lives lost.*

  The unprecedented aspect was the extent of the fires in densely populated areas. The larger area burned in the 1974/75 fire season was mainly in sparsely populated regions. The lower number of casualties in 2019/20 in spite of more fire

activity can hopefully be attributed to better warnings and protection measures. However, we agree that the statement is ambiguous and have specified in which aspect the fires were unprecedented: 'The bushfire activity across the states of Queensland (QLD), New South Wales (NSW), Victoria (VIC), South Australia (SA), Western Australia (WA) and in the Australian Capital Territory (ACT) was unprecedented in the area burned in densely populated regions.'

- *Lines 23-41: The information in here is poorly supported by references. It is also not central to the background of this attribution study. Suggest shortening substantially and refocusing on the introduction required for setting up the background for this specific study.*

  We have followed the reviewer's suggestion and shortened this part. We also added more references.

- *Figure 1: This is a very misleading and inaccurate figure. The size of the dots on this image may misrepresent fire size. The blue shading doesn't only show forested areas of eastern Australia as the caption says. Why is none of Tasmania shade blue? The blue shading seems to be a very loose definition of "forest". Why start this analysis in July 2019 where it will capture many of the normal winter savannah fires which are unrelated to the fire crisis being discussed here?*

  This figure is intended to be an introduction to the study domain, including some information on vegetation cover and fire occurrence during the event. It is not used in the later analysis. We added a note that the figure includes grass fires on the savannahs with completely different characteristics. Further, we now only show the fires from October 2019 to January 2020 to reduce the number of grass fires.

  Thank you for spotting the problems in the blue shading. We have included the forest in Tasmania and deleted the 'eastern Australia' in the caption.

- *Paragraph starting line 45: This is an enormous paragraph that moves between many different concepts. Editing required.*

This paragraph has been revised and restructured together with the entire manuscript.

- *Line 55-76: It is stated that FFDI is the index that is operationally used in Australia. Yet this index isn't used in this study. No explanation is given as to why FFDI isn't used in this study, and this would seem the most sensible choice given that it is the index used for fire danger in Australia.*

The reasoning for not using FFDI in favour of the Fire Weather Index (FWI) is given a bit further down. We will point the reader towards that section in the revised manuscript. We argue that FWI is more physically-based and a better measure of forest fire risk than the FFDI citing (Dowdy et al., 2009), an employee at the Australian Bureau of Meteorology (BOM). The problem with FFDI is that the input variables to these indices are dependent. A more statistical approach is not able to disentangle the temperature dependence from the absolute humidity influence, so a more physically-based index is needed for attribution when the main changing factor is temperature. There is (unpublished) evidence again provided by BOM that the temperature dependence of the FFDI is too high for these reasons.

*Line 84: What does "in-situ sites" mean? Some of the examples given in the previous sentence seem pretty small scale (e.g. Brisbane heat during November 2014).*

We reformulated this to be clearer, 'However, at small spatial scales, human influence on extreme heat is sometimes less clear, as in Melbourne in January 2014 (Black et al., 2015).'

- *Line 92: How do you take a 7-day moving average of annual maximum temperatures?*

[Figure]

Thanks for catching this. We clarified that this refers to the annual maximum of a 7-day moving average of daily maximum temperatures.

- *Line 96: Why are these two time windows chosen for looking at precipitation, and are these choices defendable? The FFDI uses instead an accumulated precipitation deficit. Process-wise, soil moisture is important in Australia for determining the dryness of living fuels, while temperature and humidity are important for drying of dead fuels, so it isn't clear that the precipitation indices chosen are the best options for relating to fire risk. Also, how is the "fire season" defined?*

The Fire Weather Index uses a precipitation window of a few months (a relaxation time of 52 days to be exact, as explained in the detailed description in section 4.1 and now added to the text here). We used one month as our short time scale as it greatly enlarges the number of models that can be considered (not all provide the necessary daily output); previous studies have shown the importance of including as many models as possible for drought studies to study the systematic uncertainties due to different model formulations (see, e.g., Hauser et al., 2017).

The annual time scale was included because the FWI is also sensitive to (part of) the rain season and at the time it was argued by many climate scientists that this was the relevant time scale, partially based on the a posteriori argument that these severe forest fires followed the driest calendar year on record. This time scale should correspond better to deeper soil moisture during the fire season, which is also approximated by the Drought Code in the FWI.

The fire season is defined as September–February in our study region. We moved the definition to an earlier section of the manuscript and also clarified the choice of the different time scales.

- *Figure 2: The "Log(burned area)" is not clear. What are the units? It would also be better to show burned area in true units but give the y-axis a log scale. 2019/20 conditions should be added to figure 2-right, so that readers can see*

*whether an extrapolation of this relationship to 2019/20 is appropriate (line 199). What are the green and grey lines in Figure 2-right?*

We have updated the figure as recommended by the reviewer. The units are $^{10}\log(\text{km}^2)$.

Unfortunately, the MODIS burned area data for 2019/20 are not yet available so we cannot add that data point to the graph yet.

The grey lines denote the regression lines for the individual months, the green line for all months in the fire season. We have added this to the caption.

- *Section 2.2: Why have an observational data section that then only says that the observation data used are described elsewhere in the manuscript? It would be appropriate to describe all of the data sources, and choices made, here.*

As requested by other reviewers we have reorganised the manuscript so that the heat and drought analysis are in supplementary material. We therefore present the datasets used in the main text here, and those that are only used in the supplementary material there.

- *Section 2.3: It needs to be described why these models were chosen and why others were not. These choices need to be objective, and the reader needs to know what these objective choices were. My guess is that the choice is based on starting by selecting all models CMIP5 and CMIP6 models where large ensembles ($n >$?) exist, but that is just a guess and needs to be explicit in the manuscript. Is single forcing or omitted forcing within these ensembles also a requirement?*

Thank you for pointing out this omission, the criterion to select an initial set of models is now described more clearly. It is ultimately an opportunistic choice, guided by what models are at our disposal that have all the necessary variables at the required resolution. One recent definition of large ensembles is $n \geq 15$

(Deser et al., 2020), but there is no established definition. All model evaluation criteria are subjective to some degree, hence making them explicit is essential. We use a set of statistical tests, described in Philip et al.. This is now stated in the manuscript.

- *Line 156: Are these autocorrelations based on monthly data, annual data, seasonal Data?*

We have clarified this in the text and refer to the definition above. . We consider the annual mean or driest month of the fire season, so these are annual values and the autocorrelations are year-on-year.

- *Line 160: Why 4-year smoothed? What guides this choice of length?*

The 4 years are chosen based on a previously developed compromise between the typical time scales of ENSO that we want to filter out and the time scales in the evolution of the global mean temperature that are externally forced and that we want to keep. See, e.g., (van Oldenborgh et al., 2009) for more background. We have added a clarification to that end: 'The smoothing is the shortest that reduces the ENSO component of GMST, which is not externally forced and therefore not relevant, but leaves as much of the forced variability as possible (Haustein et al., 2019). In particular, a longer smoothing time scale can cause issues extrapolating to the 'current climate' year, which is ususally a last one in the series.'

- *Line 202: Are the model spread and natural variability necessarily independent?* They do not have to be independent, but there is no evidence that they are not. At the regional scale, models that have a large sigma are not necessarily the ones that warm the most, and vice versa. For the seven models that pass our quality control it is indeed the case that they appear independent by showing a non-significant negative association (see Fig. 1 at the end of this reply). Equivalent conclusions are reached for the other variables in the paper.

[Figure]

More importantly, we effectively evaluate the models on their variability by requiring a value of the scale parameter $\sigma$ in the extreme value function fit that is within the uncertainties of the scale parameter found when fitting the same GEV to the observations. This ensures that the different models we retain do not have an unrealistically wide range of sigmas. In the unlikely event of a dependence between variability and spread, this would ensure that the influence of such a dependence would be very small. We added 'largely' before 'independent' to reflect these considerations.

- *Wouldn't this assumption be broken if model spread is at least partly due to differences in how models represent modes of variability. Wouldn't forced changes of the modes of variability (of which there is evidence) also violate this assumption?*

Modes of variability in general only represent a small fraction of the natural variability (see for ENSO van Oldenborgh and Burgers (2005), in which the explained variance typically does not exceed 20% ($r = 0.45$); similar figures are easily made for other modes like the IOD and SAM). We therefore consider all natural variability together rather than just the modes.

Concerning the second question, to second order, the forcing may affect the mean and variability of the natural variability. If it affects the mean, the effect is included in our analysis method. If it affects the variability the assumptions that the variability is constant (for temperature) or the variability over the mean is constant (for precipitation) is violated and our analysis is not valid. We test for this by computing the variability over time applying a running window of varying length. These tests based on observations are often too noisy to give much information, but the model experiments with much more data show whether the assumption holds or not. In the case of this analysis we found the variability did not change significantly. In other analyses we found that the variability indeed changes, albeit only noticeably in the future (Vautard et al., 2020). This test is part of the standard procedure employed in our analysis and described in detail

in Philip et al. (2020), we also added it to this manuscript.

- *Lines 220 to 226: It is notable that neither of the intervals of extreme heat discussed here coincided with the worst fire events that fell around New Years Day. . .*

  We do not necessarily expect a week-by-week correspondence of heat and fire, as many additional factors can lead to an intensification of ongoing fires in a given week (e.g., wind, local fuel availability, etc.). Originally, this section attempted to contextualize the different heat records broken in 2019, but we recognize that much of section 3.1 is not needed as motivation for section 3, so in the revised version we have greatly shortened this section and merged it with the old section 3.2.

  It should be noted that although the analysis is done on the hottest week of the year, similar temperature increases are expected for the slightly less hot weeks and those weeks did coincide with the worst fires in the 2019/2020 season. This provides general support for the heat-fire relationship, which can be described with the entire warm tail of the distribution, not only the hottest events. It follows that the relationship between heat and fire risk is more general and not tied exclusively to the hottest week.

- *Lines 217 and 228: Numbers for temperature anomalies in summer 2018/19 are given of 2.61oC and 1.52oC. I think that the second may be related to mean temperature and the first related to maximum temperature — but the text is not clear.*

  This has been clarified in the revised version of this section.

- *Line 232: What is the threshold behaviour of global warming referred to here?*

  Thank you for pointing out that this was poorly written. We simply refer to the finding that heatwave changes often scale linearly with global temperature increase

(Perkins-Kirkpatrick and Gibson, 2017). No threshold behavior was implied. We have revised this accordingly.

- *Figure 3: Is this data averaged for the SE Australia region? Why is there data through to 2020 if this is showing July-June years? This needs to be specified in the caption. It would also be nice if a bit of extra effort went in to making this figure attractive.*

Yes, this is data averaged for the SE Australia region, which has been added to the caption. At the time of writing the peak heat season had already passed, so we were reasonably sure that the value up to that point would be the highest in the July 2019 to June 2020 year. This turned out to be correct.

We will spend some time making the figures better-looking, as we explained in the beginning of this reply we estimated that publishing the discussion paper quickly had clear benefits in it being more useful for society, which came at the expense of having less attractive figures.

- *Line 258: It is also notable that 1938/39 was a big bushfire year in southeast Australia.*

Thank you. We have added this information.

- *Line 271: Make it explicit here that you are talking now about southeast Australia. I was initially confused because earlier a value of 40oC was given for the hottest week, but this was nation-wide.*

We have added this clarification and have also moved much of the discussion on the national heat record to the Supplementary Material.

- *Lines 286 to 289: Does this then imply that previous studies that have attributed heat extremes in Australia, and have given quantitative results, are flawed/affected by this same model capability problem? If so, this is an important finding that should be made clearer.*

We checked this for one model (EC-Earth), and found that both discrepancies, in trends and variability, are mostly confined to southeastern Australia, so would not necessarily affect the other studies done for Brisbane, Adelaide, etc. The higher trends in southeastern Australia are also clearly visible on maps of trends in one-day heat, e.g. at https://worldweatherattribution.wordpress.com/analyses/trends-in-weather-extremes-2018. This is one of the reasons why we have to take the effort of doing event attribution on a particular event, rather than just quoting general results for much larger regions.

- *Figure 4-7: How sensitive are these results to the choice of the 4-year smoothing for the global mean temperature dataset? I think that it is important to show results with a longer smoothing that could better account for differences in interannual to interdecadal variability, which will be random in models and cancel out over a large ensemble, com- pared with the single realisations of observations/reanalysis where the influence of variability will be a real part of the signal.*

The 4-yr smoothed GMST is just used as a measure of the externally forced climate signal. It is already highly correlated with the CO2 concentration and total anthropogenic forcing ($r > 0.96$). We repeated the most sensitive fits presented in the paper, that of TX7x, with a 10-yr running mean GMST and found only small differences in the fits compared to using a 4-year smoother (Fig. 2 at the end of this reply). Note that the model fits use the simulated ensemble mean global mean temperature, which is an even more robust estimate of the forced response.

Recent publications have shown that there is very little unforced variation in the global mean on these longer time scales (e.g., Haustein et al. (2019)). Also, very little of the variability over land is driven by teleconnections from lower-frequency modes of the climate system (e.g., (van Oldenborgh et al., 2012), Fig. 2). Consistent with several other papers in the event attribution literature, we concluded that a 4-year smoothing is adequate to reduce the influence of the two largest

causes of interannual variability of the GMST: ENSO and the high latitude winter temperatures.

- *Lines 301 to 302: Should ACORN be included if there are issues from changing data coverage? Surely then it would be better just to use AWAP for the Australian observational dataset as it specifically resolves this issue.*

Indeed the AWAP dataset resolves the issue of the varying station density (although the coastal effects may not be resolved adequately by the interpolation algorithm). However, the ACORN station dataset resolves another issue: the homogeneity of individual station time series. We prefer to show both to convince the reader that neither issue significantly affects the estimated warming trends. Leaving out the observational dataset with the lowest trend would also open the analysis up to claims of cherry-picking.

- *Line 303: The 1929 event????*

This was a typo and should read "1939", but we will clarify in the revised version that we refer to "the exceptional heat wave of January 1939, which was not reproduced in certain datasets, but confirmed by station data."

- *Figures 6 and 7 have data labels cut off.*

Thanks for pointing this out. We will revise the figures before resubmission, including clear labels etc.

- *Lines 335-342: This is an area where I have questions over the suitability of FWI for this study, which only considers precipitation over the last 52 days. The FFDI used op- erationally in Australia uses a longer-term drought index, which is particularly relevant given the multi-year timescale of droughts in Australia and the changes in soil mois- ture that influence fuel dryness. Multi-year drought was a factor in the 2019 extreme conditions and so should not be ignored in this way.*

[Figure]

The FWI considers in its DC code, which describes the dryness of the deepest inflammable soil layer in the FWI, past precipitation with an exponential time scale of 52 days (van Wagner, 1987). This implies that part of the winter rains are indeed included to some extent. In fact, the last three years were so dry that the winter, in which evaporation is less even in the absence of a seasonal cycle in precipitation, does not manage to replenish the soil moisture and it shows a steady decline, enhancing the FWI in the study season 2019/20 (see figure 3 at he end of this reply). We have rephrased the manuscript to make this clear.

In contrast, in our study we consider more general drought events, none of which was as extreme in the observed record. Of course those are generalities and do not rule out an influence that was only present in 2019/20.

The question how relevant 3-yr droughts were in general is separated into two parts: are multi-year droughts associated with more severe bushfires in south-eastern Australia? Do their properties diverge from consecutive independent one-year events?

Considering the first, we did a correlation analysis of the logarithm of Sep–Feb MODIS area burned against precipitation on various time scales (an extension of Fig. 2 in the article). This confirms the strong correlation with Sep–Feb precipitation: $r = -0.74$. The correlation with Jan–Dec annual mean precipitation is $r = -0.49$, so already only half as strong of a predictor, even though it includes much of the simultaneous connection and the FWI does include some winter precipitation. Finally, the correlation with the preceding 3-yr mean precipitation is $r = -0.01$. Although the uncertainties are high (95% CI on this is $-0.5$ to $+0.5$), this suggests that multi-year meteorological droughts are in general not necessarily a good predictor of area burned in the final year. While this doesn't rule out the possibility that in the special case of 2017-2019, the multi-year drought contributed to drying out fuels in advance of the 2019/20 fire season, it is not a relationship that is generally supported by the data.These conclusions are corroborated by Dowdy et al. (2009), who showed that the Fire Weather Index represents fire risk in Australia better than the FDDI.

Second, we did analyse one-year droughts (low precipitation in the calendar year ending in the fire season) and found that they showed no attributable trend in the study area in southeastern Australia, so it is unlikely that they influenced the long-term trend that we see in fire weather risk. Looking at the autocorrelation and spectrum of annual low rainfall in southeastern Australia (Fig. 4 at the end of this reply) we see no evidence of multi-year droughts being anything but random collections of annual droughts. The lag-1 autocorrelation is 0.15, well within the 95% CI around zero and, even if it were real, would explain only 2% of the variance. The lag-2 autocorrelation is 0.01. These numbers do not support investigating trends in multi-year droughts separately either.

Based on these results we decided that studying multi-year droughts separately would not give any new insights and have larger uncertainties. We have added a short paragraph to the text discussing the analysis here (suppl. mat.).

For completeness, preliminary results for 3-year drought illustrating the points made above are attached. The results are not different from the annual drought analysis in the paper but with larger uncertainties: neither observations nor models show a statistically significant trend. The fits are also not very stable with only 8 or 12 degrees of freedom to fit four parameters.

- *Line 348: The suitability of only using the lowest 20% or 30% of annual mean precipitation observations needs to be better justified. It is not clear that this is a suitable measure to be using for the attribution testing. Figure 8 still seems to show all annual data, but these should only show the 20% and 30% of data points that are actually being used in the analysis.*

This is simply based on the minimum number of points needed to reliably fit the 3-parameter GPD function, about 20, while still allowing to look at the low tail of

the distribution, which is the one relative for the question of whether there is a trend in low precipitation years. We added this to the text.

In Fig. 8, the part that was fitted to the GPD can be deduced by where the thick lines stop and from the return time (1/20% = 5 yr). We thought it was clearer to show all points for reference how the tail connects to the rest of the distribution. We plan to make the points not used in the fit a paler shade of blue to show the difference.

- *Line 350: What is the justification for scaling precipitation relative to smoothed global mean surface temperature? I understand this for looking at temperature attribution, but it isn't well justified in the text that this should also apply for precipitation.*

First, we investigate to what extent the trends in any variable are due to global warming. To do this we use the smoothed global mean surface temperature as covariate, which shows the part of the trend most related to the global warming trend. Next we use the climate models to check whether the trend is indeed attributable to global warming.

Regarding the exact form in which the covariate is used in the fit, *scaling* the distribution with the covariate is not justified for temperature, we just *shift* the distribution for that variable, so assume the scale parameter that describes variability is constant (and check that in the observations and model data). For precipitation, the scaling (i.e, assuming $\mu$ and $\sigma$ vary the same way with the covariate so that the dispersion parameter $\sigma/\mu$ is constant) is justified on two grounds: to avoid negative precipitation and on the theoretical arguments given in Hanel et al. (2009). It is an approximation that is widely used in the hydrological community.

- *Figure 8: again text is cut off. Titles are poorly designed. My frustration at the lack of care in preparing the manuscript is rising. . .*

As we mentioned before, we had what we perceived as a choice between a polished manuscript and the discussion paper being available rapidly and thus being useful to the adaptation community. We chose the latter in the hope that we will be granted the time to fix these issues in revision.

- *Lines 357 to 359: This long sentence is not well constructed and hard to read/understand.*

  We have split up and revised this sentence.

- *Figures 5, 6, 7, 9 , 10, 15, 16 and 17 all have unlabelled x-axes.*

  Apologies. The caption mentions that these are Probability Ratios. We will add these labels in resubmission.

- *Section 4.5: The precipitation fails to take into account the seasonal patterns of rain- fall change that are well described by Australia's Bureau of Meteorology (increasing warm season rainfall in northern regions, decreasing cool season rainfall in southern regions). Decreases in cool season rainfall may not directly influence the months when fire occurs, but is important in the context of multi-year droughts and soil moisture deficits that influence fuel dryness.*

  See reply to comment on lines 335-342.

- *Section 4.6: Reiterating earlier concerns that this analysis doesn't actually do anything related to meteorological drought. In Australia droughts are multi-year events and so aren't and can't be described by a rainfall deficit over the scale of months to a year. The analysis also fails to address the way droughts in this region are being viewed, with southeast Australia being a region that is usually dry (often in drought) and occasionally experiences drought-breaking rain events.*

  See reply to the comment on lines 335-342 above.

- *Section 5: Part of the reason why the 2019/20 fires were so devastating was because of the number of extreme fire events where pyrocumulonimbus activity occurred. The metrics here do not account at all for the factors that influence pyroCb risk.*

No, we restrict ourselves to the large-scale factors in the weather that give favourable conditions to bush fire development. The small-scale feedbacks that allow the fires to grow larger cannot yet be investigated. We have added a remark on this caveat in the revised version.

Section 5.2: The equation for DSR is given, but I don't see anywhere the equations given for FWI or for MSR.

The FWI has a complex formulation that is given in the literature cited. The MSR is defined one sentence above the DSR as the monthly average of this quantity.

- *Section 5.5, including Line 459 and Figure 16 and 17: Because all models severely underestimate the risk compared to ERA-5, surely this is an indication that accurate quantification of attributed changes in fire risk is not possible. It seems very misleading to then go on and give percent increases for 2019 and for 2C warming based on the model output. Similarly, at line 464, it seems unwise to make a statement about the trends being non-significant when there is such disagreement between observations and models, with the models being a severe underestimate.*

Indeed, which is why we do not provide a synthesised number for the increase in Fire Weather Index, but only separate numbers for ERA5 and the models and emphasise how different they are. In the MSR-SM discussion, we added the phrase 'in the models' to the first sentence, where it was erroneously left out. The reanalysis-model discrepancy and the fact that we only use the lower bound as a conservative estimate has been emphasised more in the revised version. As the reviewer notes, all other information is not reliable due to this discrepancy.

- *General comment: I didn't specifically keep track, but it seems that many acronyms are not defined.*

We will do a thorough check of the introduction of all acronyms before resubmission.

- *Paragraph at line 476: The precipitation contribution can also be seen to be very much underestimated in these models compared with ERA-5. This should not be overlooked.*

The trend in precipitation during the short 1979–2019 period says very little about long-term trends, one needs a century of data to be able to start to study drought trends. We added a remark that the precipitation trend in ERA5 may well be due to natural variability but does contribute to the trend in fire weather risk.

- *Paragraph at line 489: The factor of four and factor of nine numbers I think come from the lower end of ERA-5? But these values aren't the focus of the text associated with the interpretation of figure 16 and 17, which instead gave numbers based on models. This is very confusing for readers.*

Thanks for catching this. We have revised the paragraph discussing Figs 16 and 17 to also emphasize the observational numbers, so they do not appear out of nowhere when discussed later.

- *Section 6: This section feels like it was tacked on as an after thought. In particular, these modes of variability are part of the extremes in 2019 and so are already part of the attribution analysis carried out in earlier sections. The way section 6 is written seems like these modes are in addition to the 2019 extreme conditions, which isn't the case.*

There were a lot of questions about the specific situation in 2019 and the role of the IOD and SAM, especially from Australia where the Bureau of Meteorology had been emphasising these drivers. We thought it would be useful to address

these, even though, as the reviewer remarks, they are implicitly included in the analysis of the previous sections. We emphasized that it is merely for situational context given the extreme nature of these modes in 2019, but that it is not additive to the attribution analysis presented earlier in the paper.

- *Section 6.2 and Figure 19: The analysis of the IOD fails to take into account known problems with instrumental data for the Indian Ocean. Generally it is assumed that indices of the IOD are only reliable after 1958, and even after that time there are differences in how well different data products capture the upwelling signal in the eastern upwelling region of the IOD. No information is given about what dataset is used for the analysis in figure 19, so readers can't evaluate potential problems related to the dataset used, though clearly data prior to 1958 is used and this should not be part of any analysis of IOD variability.*

We thank the reviewer for the additional information. The DMI index is based on ERSST v5. We added his information to the text and the caption. We have also replaced this figure with one starting in 1958. This makes the correlation to Jul–Dec precipitation slightly weaker, $r = -0.18$ instead of $-0.22$ and does not change any of our conclusions. The correlations to TX7x are now all non-significant.

- *Figure 19 and 20: The IOD and SAM have been reported in other studies to influence Tmax, which is also important to fire risk. These should also be shown along with precipitation, and the text should investigate further why this study finds no relationship to Tmax when others have.*

The one publication that we are aware of for the SAM that considers the same region also does not find a correlation (Perkins et al., 2015). For the IOD, the difference is that we avoid double-counting ENSO and IOD teleconnections by subtracting the influence of ENSO on the IOD before investigating the IOD connections with Australia. The ENSO forcing of part of the IOD has been extensively

documented, we consider the part of the IOD that is independent of ENSO. As we mention there are significant ENSO teleconnections to this region.

We have added the temperature figures as requested.

- *Figure 20: ENSO and SAM also interact with a negative correlation, so for consistency in analyses the ENSO-independent SAM relationship should be used.*

We thank the reviewer for reminding us of this. After doing this, the correlation in DJF becomes stronger but does not change our conclusion that there is no significant connection with TX7x. The Jul–Dec correlation is weaker and makes the connection to rainfall slightly weaker. The text and figures have been updated based on this analysis.

- *Line 587: 38 million hectares in 2002/03 — this isn't correct.*

Ellis et al write 'over 54 million hectares were affected by bushfires'. We have adopted this phrase in our text. (The 38 million hectares appears to come from Wikipedia, not Ellis et al. 2004.)

- *Line 589: Need to specify that the 1974/75 fires were grass fires, which have different drivers to forest fires.*

We added this information to the text.

- *Section 7: This section has interesting information, though important points are frequently unreferenced. However, this section reads like a separate study that is unrelated to the focus of this paper on the attribution of Australian bushfire risk to anthropogenic climate change.*

We think it is important to discuss other factors than climate that have changed the risk of the impacts of the bushfires, the vulnerability and exposure. Even though climate change has increased the risk of bushfires, these other factors,

together with meteorological factors we could not analyse such as the pyrocu-mulinumbus feedback, also influence the severity of the disaster. Short-term, the vulnerability and exposure are also the factors that can be addressed to decrease the risks, as stopping climate change will (at best) take a long time. This section has also been documented in the accepted companion paper, Philip et al. (2020).

- *Conclusions: I've run out of steam for providing specific comments on the conclusions, but based on my critiques of the previous sections of the paper significant changes are also needed here.*

We are grateful for the many helpful and valuable comments by the reviewer, which undoubtedly will help us in our effort to polish this paper and make it more readable. While we acknowledge that the initial presentation of the study was suboptimal, we are glad to report that none of the adjustments to the text and the additional analyses affect the conclusions of the study.

**References**

Black, M. T., Karoly, D. J., and King, A. D.: The contribution of anthropogenic forcing to the Adelaide and Melbourne, Australia, heat waves of January 2014, Bull. Amer. Met. Soc., 96, S145–S148, https://doi.org/10.1175/BAMS-D-15-00097.1, 2015.

Deser, C., Lehner, F., Rodgers, K. B., Ault, T., Delworth, T. L., DiNezio, P. N., Fiore, A., Frankignoul, C., Fyfe, J. C., Horton, D. E., Kay, J. E., Knutti, R., Lovenduski, N. S., Marotzke, J., McKinnon, K. A., Minobe, S., Randerson, J., Screen, J. A., Simpson, I. R., and Ting, M.: Insights from Earth system model initial-condition large ensembles and future prospects, Nature Climate Change, 10, 277–286, https://doi.org/10.1038/s41558-020-0731-2, 2020.

Dowdy, A. J., Mills, G. A., Finkele, K., and de Groot, W.: Australian fire weather as represented by the McArthur Forest Fire Danger Index and the Canadian Forest Fire Weather Index, Technical Report 10, The Centre for Australian Weather and Climate Research, 2009.

Hanel, M., Buishand, T. A., and Ferro, C. A. T.: A nonstationary index flood model for precipitation extremes in transient regional climate model simulations, J. Geophys. Res. Atmos., 114, D15 107, https://doi.org/10.1029/2009JD011712, 2009.

Hauser, M., Gudmundsson, L., Orth, R., Jézéquel, A., Haustein, K., Vautard, R., van Oldenborgh, G. J., Wilcox, L., and Seneviratne, S. I.: Methods and Model Dependency of Extreme Event Attribution: The 2015 European Drought, Earth's Future, 5, 1034–1043, https://doi.org/10.1002/2017EF000612, 2017.

Haustein, K., Otto, F. E. L., Venema, V., Jacobs, P., Cowtan, K., Hausfather, Z., Way, R. G., White, B., Subramanian, A., and Schurer, A. P.: A Limited Role for Unforced Internal Variability in Twentieth-Century Warming, J. Climate, 32, 4893–4917, https://doi.org/10.1175/JCLI-D-18-0555.1, 2019.

Perkins, S. E., Argüeso, D., and White, C. J.: Relationships between climate variability, soil moisture, and Australian heatwaves, Journal of Geophysical Research: Atmospheres, 120, 8144–8164, https://doi.org/10.1002/2015JD023592, 2015.

Perkins-Kirkpatrick, S. E. and Gibson, P. B.: Changes in regional heatwave characteristics as a function of increasing global temperature, Scientific Reports, 7, 12 256, https://doi.org/10.1038/s41598-017-12520-2, 2017.

Philip, S. Y., Kew, S. F., van Oldenborgh, G. J., Otto, F. E. L., Vautard, R., van der Wiel, K., King, A. D., Lott, F. C., Arrighi, J., Singh, R. P., and van Aalst, M. K.: A protocol for probabilistic extreme event attribution analyses, Advances in Statistical Climatology, Meteorology and Oceanography, accepted, 2020.

van Oldenborgh, G. J. and Burgers, G.: Searching for decadal variations in ENSO precipitation teleconnections, Geophys. Res. Lett., 32, L15 701, https://doi.org/10.1029/2005GL023110, 2005.

van Oldenborgh, G. J., Drijfhout, S. S., van Ulden, A. P., Haarsma, R., Sterl, A., Severijns, C., Hazeleger, W., and Dijkstra, H. A.: Western Europe is warming much faster than expected, Clim. Past., 5, 1–12, https://doi.org/10.5194/cp-5-1-2009, 2009.

van Oldenborgh, G. J., Doblas-Reyes, F. J., Wouters, B., and Hazeleger, W.: Decadal prediction skill in a multi-model ensemble, Clim. Dyn., 38, 1263–1280, https://doi.org/10.1007/s00382-012-1313-4, 2012.

van Wagner, C. E.: Development and Structure of the Canadian Forest Fire Weather Index, Forestry Technical Report 35, Petawawa National Forestry Institute, Ottawa, Canada, 1987.

Vautard, R., van Aalst, M. K., Boucher, O., Drouin, A., Haustein, K., Kreienkamp, F., van Oldenborgh, G. J., Otto, F. E. L., Ribes, A., Robin, Y., Schneider, M., Soubeyroux, J.-M.,

Stott, P. A., Seneviratne, S. I., Vogel, M. M., and Wehner, M. F.: Human contribution to the record-breaking June and July 2019 heat waves in Western Europe, Environ. Res. Lett., https://doi.org/10.1088/1748-9326/aba3d4, 2020.
* * *
**Fig. 1.** Variability (scale parameter) versus model spread (best fit to trend) for the seven models that passed the quality control tests.

[Figure]

**Fig. 2.** Comparison of the $\Delta$T using a 4-yr running mean and 10-yr running mean in the GMST covariate.

Interactive
comment

**ERA5 DC (NSW) 2010-2019 15-day run. avg. / yearly mean**

**Fig. 3.** ERA5 estimate of the DC component of the FWI with 12-month running mean.

Interactive
comment

[Figure]

[Figure]

[Figure]

[Figure]

**Fig. 4.** Autocorrelation function and spectrum of annual mean precipitation averaged over the study area in southeastern Australia (CRU TS 4.04).

**Fig. 5.** As Fig. 11 in the manuscript but for 3-yr low precipitation in the study area.

---

## Author Comment (AC6) · 10 Nov 2020

*This Manuscript aims to evaluate if the exceptional fire risk associated with the bushfire during the last months of 2019 and January of 2020 was exacerbated by anthropogenic climate change, namely in the Southeastern Australia, where the fires were particularly severe. The authors analysed the exceptionally of the heatwaves and drought and how they reflected on the Canadian Fire weather Index (FWI). The analysed also, using the current climate models, the long-term trend of the above-mentioned parameter. The driver effect of the Indian ocean dipole and Southern Annular Mode was also assessed. The Overall context of the subject is very important in Australian Context Taking in account the importance of extreme climate events, such as drought and heatwaves, for the region within the context of warming tendency. It should be noted that Australia*

[Figure]

*has a large history of tragic events despite being one the countries (maybe the first one) with better management strategies.Therefore, the work seems to be appropriate for this journal.*

*However the manuscript is very exhaustive, very hard to follow with an excessive number of figures and details. The analysis is very repetitive and should be organized in a more effective way in order to increase readability. Otherwise, the main achievements will be lost in somewhere.*

We thank the reviewer for their review and acknowledge that the text in particular was not easy to read. This is a comment common to all the reviews received, so it justifies and motivates a thorough revision and restructuring of the text, which we will undertake. In particular, we are planning to follow the suggestion to move certain analyses to a supplementary material and just reference their results in the main text.

**1  MAJOR**

*As I said the subject and results of this manuscript are of great interest for a wide range of readers. However, the reading of the paper is very tiring, the number of figures in the manuscript is very high and the results, synthesis, interpretation and conclusion for each topic is really tedious. Nevertheless, I recognize that to present these results is a very hard task. Therefore, my next comments are suggestions that may increase the readability and increase the number of interested readers that should be attracted to the important results of the manuscript.*

1. *Some paragraphs of Introduction show a strong lack of references, namely the first ones that have only references to national reports. The same situation occurs in several paragraphs of the introduction and along the manuscript that seems to be more appropriate for a technical report than to a paper.*

We will shorten this part of the introduction and add more references for the parts that we retain.

2. *The entire paper should be reorganized. The structure should be less technical and descriptive and more similar to paper structure:Introduction, Data and Methods, Results, discussion and conclusions. Several section should be merged,and the figures reduced significantly in the maintext. The remaining figure should be moved for Supplementary Information.*

We will restructure the paper following these recommendations, which were also expressed by the other reviewers.

3. *I understand other factors should be included in fire risk analysis. However The present manuscript is so long and the main contribution of the authors are related with Fire Weather Risk. Therefore,I suggest removing section 7 from the manuscript. A paragraph related with the other drivers may be included in Section 8. Conclusions. Consider changing the title accordingly.*

Generally, many studies conducted by the World Weather Attribution group include a section on vulnerability and exposure, as some of the co-authors work in this field and help us put the physical results (often focused on ocean-atmosphere processes) into context of impacts on the ground. We therefore would like to keep some of the content. However, we recognize that Section 7 is too long, in particular given the length of the rest of the paper. Therefore, we will condense Section 7 and attempt to cross-reference it better in the rest of the paper.

4. *Clarify if Figure 1 shows the forested areas over the entire Australia or over Eastern Australia. Consider Move Figure 1 to Data and Methods.*

We have redrawn Fig. 1 to show forested areas in all of Australia. Ww think it is important to show this figure in the main text to justify the choice of the study region.

5. *The option of using the CanadianFire Weather Index (FWI) instead of the FFDI is neither presented, neither justified.*

   The choice of FWI was born out of the data availability and the generally good correspondence between FWI and fires in the domain as documented in Dowdy et al. (2009). The general behavior of the two indices is also similar. However, we agree that this can be stated more explicitly, which we will do in the revised version.

6. *Consider comparing the performance of FWI with Forest Fire Danger Index (FFDI) developed and commonly used over Australia for indicating dangerous weather conditions for bushfires.*

   A comprehensive comparison of the indices for the domain of interest was conducted by Dowdy et al. (2009). They generally behave similarly, even if not identically across the full distribution of fire risk values. One general conclusion is that FFDI is slightly more temperature-sensitive than FWI. Thus, given the important role of temperature in driving the secular trend in fire weather in Australia, we again interpret our choices to be conservative and in line with our goal to provide a lower bound for the role of anthropogenic climate change in SE Australia's fire risk. Given existing extensive comparisons of the two indices in the literature, we would like to refrain from adding another index to the study. The literature also suggests that the uncertainty from the choice of index is negligible compared to the uncertainty from observational datasets and climate models used, which we are already illustrating and quantifying in detail.

7. *The author uses several different datasets for analysis the different variables and models. I would prefer to see a less wide lack of reanalysis and gridded datasets. For instance, why do not used precipitation AND temperature from CRU.*

   The CRU TS temperature dataset is monthly, whereas we needed a daily dataset to study the heat extremes, so we could not use it. More generally in our analyses

we find it useful to compare different datasets to make sure that the signals we find are not artefacts of the analysis methods of these datasets. For instance in this study, we had started off using the Berkeley daily temperature dataset, and only by comparing it to other datasets such as AWAP did we find out that it did not represent the weather of this region well enough to be used. If we would have based the results only on the Berkeley dataset without using others they would have been much more clear-cut but not correct.

8. *The author use ERA5 from ECMWF to compute FWI. Why do not use the FWI computed using ERA5 by ECMWF and disseminated already by Copernicus?*

   For consistency across our study, we use our own FWI code and apply it to all datasets and model output consistently. In practice, our ERA5 FWI implementation is identical to the one provided directly by ECMWF (after they fixed a bug that gave a difference). Also, the version provided by ECMWF is not currently updated beyond 2018, so could not be used for our attribution study.

9. *Data from FFDI using ERA5 computed by ECMWF? Did The authors compare their results for FWI with the ones disseminated by Copernicus?*

   We assume the reviewer refers to the FWI, since we are not using the FFDI (see earlier reply above). We did compare our implementation of FWI with the ERA5 implementation and they are identical during the period of overlap.

10. *The option of using a window of 7 days for temperature and FWI is not fully presented and justified. Did the author make a sensitivity study to define the 7-days window? Why 7 days and not 5 days? Provide references and justification for the option made, including comparison with the widely accepted definitions of heatwaves adopted by WMO or based on percentiles.*

    The 7-day averaging interval in the Fire Weather Index was chosen to obtain a good correlation between the highest FWI value of the year and the area burned

in the 2017 Sweden forest fires. A much shorter interval does not correlate well and a much longer one invalidates the use of the GEV for the statistical fit. Of course any definition has a degree of arbitrariness in it, but this one appears to work well (Krikken et al., 2019). Changing it slightly, e.g., to 5 or 9 days, did not make any difference compared to all the other uncertainties in the analysis.

Concerning heat wave definitions, in previous attribution studies we found that the ETCCDI definitions of heat waves are often not the most useful ones to describe the impacts. In particular, the percentile-based one (Tx90p) denotes warm episodes in other seasons than high summer also as heat waves, even though their impacts are very different and usually much less severe. The other definition, TXx, is more useful but does not take into account that for some impacts, e.g., mortality in countries where the vulnerable population is indoors or indeed forest fire risks, a single hot day has much less impact than a string of hot days. We chose a definition here that connects with the time scale on which we evaluate the FWI.

We will add some discussion of these considerations in the revised version to clarify our choices of indices.

11. *Figures in Figure 3 Correspond to averaged values over the southeastern Australia? Information must be presented in figure caption and main text. Consider moving Figure 3 to Supplementary information.*

We will significantly revise all text and figure captions to be more self-explanatory. In response to other review comments, Fig. 3 has been expanded to include more datasets. At the same time, we will move most of the temperature and precipitation analysis to the Supplementary Material, as suggested by several reviewers.

12. *Lines 275 consider to present figure for JRA-55 in Supplementary information.*

[Figure]

We have made a new Figure 3 including all datasets discussed (attached) and indeed moved it to the Suppl. Mat.

13. *Consider reducing Figures 5-7 to one figure and moving the remaining for Supplementary information.*

Thanks for this suggestion. Indeed, we will move the figures to the Suppl. Mat., although we may keep them separate for clarity. See reply to point 11.

14. *Line 335: Analysis of the driest month in fire season: How must considered month? Difference 2mm/day makes has a significant impact on this DMC, DC, FFMC and FWI? The impact of a delta of precipitation in a region with very low values of daily precipitation should be assessed in terms of fire weather risk. Consider providing a sensitivity analysis on this impact.*

We are not completely sure whether we fully understand the reviewer's questions. First, this region of Australia does not have a strong seasonal cycle in precipitation; summer precipitation is very similar to winter precipitation in mm/dy. Second, we chose to use a measure based on monthly precipitation in order to maximize the number of observational datasets and model runs available, as trends in low precipitation are very model sensitive, see, e.g., Hauser et al. (2017).

We tested two definitions: (i) for each year the driest month of the six months in the fire season September–February was fitted to a GEV function and (ii) the lowest 20% of all months in the fire season were fitted to a GPD function. Both gave very similar results.

15. *Line 339: please provide quantitative information. What are the observed values and the normal values.*

Added the number from the BOM publication and a climatology for all Australia: '277 mm/yr (...) against a climatology of 418 mm/yr over 1900–2019'. Also added

the numbers for the study area: 'with 1.28 mm/dy (467 mm/yr), against a 1900–2019 climatology of 2.25 mm/dy (820 mm/yr) in the same AWAP dataset.'

16. *Line 351: Annual mean low precipitation analysis: information about the precipitation regime over the region is desirable over a region, i.e., information about inter and intra annual variability of precipitation on the region.*

We added a sentence about the lack of seasonal cycle and large variability to the previous paragraph: 'Note that mean monthly precipitation in the study area is almost constant through the seasonal cycle, but inter-monthly variability is very large compared to the mean, $\sigma/\mu$ = 49% on average, larger in summer and smaller in winter.'

17. *What are the observed values and the normal values.*

Added 'with 0.49 mm/dy (15 mm/mo) against a normal value for December of 2.44 mm/dy (76 mm/mo).'

18. *Consider reducing Figures 9-11 to one figure and moving the remaining for Supplementary information.*

In line with our plans to split the paper into main text and Supplementary Information, we will most likely follow this suggestion and move the analysis for temperature and precipitation to the Supplementary Information.

19. *Lines 411: Why do you use a window of 7 days for FWI and after the MSR instead of the DSR for a window of 7 days?*

The DSR has been constructed to linearly scale with fire severity and therefore better suited for averaging over longer time scales (van Wagner, 1987). We've tested the sensitivity to either using a 7-day rolling averaged FWI or DSR and found only very little sensitivity. Hence, we choose to use the 7-day rolling averaged FWI because it is much more widely known and used. For averaging over

longer time scales we did use the monthly averaged DSR (MSR) as this metric is, as previously stated, better suited for monthly and seasonal averages and also frequently used to asses fire risk on longer time scales (e.g., Malevsky-Malevich et al., 2008).

20. *Line 515 did the authors evaluate the formula of DSR for Australia region?*

No, and this also does not make much sense as it would lump together very disparate regions with different vegetation (rain forest, Mediterranean forest, grasslands) and seasonal cycles. We are also not aware of any analysis in the published literature.

21. *Figure 13. In the figure caption describe the information for dots.*

Dots are individual annual values. We have added this to the caption.

22. *Consider reducing Figures 15-17 to one figure and moving the remaining for Supplementary information.*

We will consider merging these figures if it is possible to maintain readability of the figure. Generally, we are planning to keep most of Section 5 in the main paper (incl. figures), but move a lot of the material from Sections 3 and 4 to the Supplementary Material.

**2  MINOR**

1. *(Line 61): parenthesis is missing before 'Clarke'*
   Done.

2. *The link for each database should be provided.*

As indicated in the manuscript, all time series used in the analysis are available from one page on a public web site. These time series in turn contain links to the datasets from which they were generated.

3. *Figure 8: titles are not completed*

   Figures, captions, and titles will be revised throughout the paper.

**References**

Dowdy, A. J., Mills, G. A., Finkele, K., and de Groot, W.: Australian fire weather as represented by the McArthur Forest Fire Danger Index and the Canadian Forest Fire Weather Index, Technical Report 10, The Centre for Australian Weather and Climate Research, 2009.

Hauser, M., Gudmundsson, L., Orth, R., Jézéquel, A., Haustein, K., Vautard, R., van Oldenborgh, G. J., Wilcox, L., and Seneviratne, S. I.: Methods and Model Dependency of Extreme Event Attribution: The 2015 European Drought, Earth's Future, 5, 1034–1043, https://doi.org/10.1002/2017EF000612, 2017.

Krikken, F., Lehner, F., Haustein, K., Drobyshev, I., and van Oldenborgh, G. J.: Attribution of the role of climate change in the forest fires in Sweden 2018, Natural Hazards and Earth System Sciences Discussions, 2019, 1–24, https://doi.org/10.5194/nhess-2019-206, 2019.

Malevsky-Malevich, S. P., Molkentin, E. K., Nadyozhina, E. D., and Shklyarevich, O. B.: An assessment of potential change in wildfire activity in the Russian boreal forest zone induced by climate warming during the twenty-first century, Climatic Change, 86, 463–474, https://doi.org/10.1007/s10584-007-9295-7, 2008.

van Wagner, C. E.: Development and Structure of the Canadian Forest Fire Weather Index, Forestry Technical Report 35, Petawawa National Forestry Institute, Ottawa, Canada, 1987.

[Figure]

**Fig. 1.** The highest 7-day running mean of daily maximum temperature of the July–June year in a) the AWAP analysis, b) Berkeley Earth, c) 20CRv3 reanalysis, d) CERA-20C ensemble mean (DJF maximum), e) ERA-20C

---

## Author Response (AR3)

Reply to reviewers second revision

**Report #1**

*This paper corresponds to the revised version of the paper "Attribution of the Australian bushfire risk to anthropogenic climate Change" previously submitted to Natural Hazards and earth System Sciences. The authors have made a great effort to answer the main issues raised in my previous evaluation. In particular, they revised the introduction, results and discussion section, shortening the manuscript significantly.*

*I am satisfied with the new version of the manuscript itself. However, the solution adopted by the authors to shortening the manuscript was to send the entire section 3. to supplementary information. In fact, the supplementary information has more similarities to a second paper than to a supplement material. I strongly suggest reducing information in the supplement and reorganize it in order to fit the usual format of this kind of material.*

*Therefore, I am glad to give my approval on the publication of the improved version of the paper after the necessary changes on supplementary information.*

We agree that the supplementary information is long and contains introductory (SI) paragraphs which were taken from the original main text. We do believe that the majority of the SI is indeed needed to corroborate and support the results in the main text. However, we have revised the SI by starting with a summary and justification for the necessity of this supplementary analysis and have shortened some other parts of the SI to hopefully avoid the character of a stand-alone paper.

**Report #2**

*I appreciate the revisions that have been made to this paper. This version is much improved over the original paper – and indeed, is closer to the paper that should have been submitted in the first instance.*

*I disagree with the implied suggestion that allowances should be made due to time pressure associated with the goal of rapid attribution – indeed, if the rapid analysis of events has now evolved into a quasi-operational activity, then I would argue that results should be published elsewhere than in the peer reviewed scientific literature unless specific cases have something to teach us that was not previously known about the methods and tools that are used or the mechanisms that produced the event in question.*

We agree completely with the reviewer for operational attribution analyses. However, this study is definitely not yet in this category. Only heat and cold events, and large-scale precipitation events are in our opinion ready to be operationalised because there are enough studies published, so that following a protocol will give the correct answers. This is the first wildfire analysis to be published from this team. There have not been many by other authors and some of those have in our view serious shortcomings, like analysing only a single climate model or only the reanalysis period, both of which can give spurious trends. We want to give an example of a more thorough analysis in the peer-reviewed literature that future operational studies can refer to and refine.

*While the paper is improved, I do have some additional comments that are listed below. Many of these comments call on the authors to think a bit more carefully about how they are communicating with readers since they do not necessarily share the vocabulary that the authors use to communicate amongst themselves.*

Thank you, it is always useful to get these types of comments from reviewers outside the field for a paper that probably will attract many readers from outside the attribution community.

*51-52: The manuscript mentions in a few places that "ignition sources and type of vegetation[, which are] factors largely independent of meteorology[,] play an important role", but surely this is not true. Lightning is certainly a major ignition source, and the type of vegetation in a given location is certainly dependent on the climatology of that region. Better wording would be appropriate.*

We agree and have reworded the text to 'In addition, ignition sources and type of vegetation play an important role. The types of vegetation depend on the long-term climatology, but do not vary on the annual and shorter time scales we consider, and the dry thunderstorms providing a large fraction of the ignition sources are too small to analyse with climate models. In this analysis we therefore only consider ... '

*128: Do you mean similarity in the correlation coefficients (rather than confidence intervals)? The discussion goes on to mention explained variance, which implies that it refers to the square of the correlation coefficient.*

Indeed, this was an oversight. We have revised this to read: 'Given the similarity in the correlation coefficients ($r$) within their confidence intervals, the log-linear relationship appears to explain equal variability ($r^2$) to that of the ranks.'

*169-170: Exactly what is an "annual mean low precipitation" value? This seems a confusing combination of words.*

'low annual mean precipitation'

*185: I think what is meant is that the iterative maximization of the likelihood function does not converge. It is not the "fit", per se, that doesn't converge.*

I am afraid I do not see the difference but I am not an expert on the terminology here so we revised according to the suggestion of the reviewer.

*193: What is meant by the statement that precipitation is "positive-definite"? Most readers would think about a matrix when they see this term, and wonder whether they missed something in the description of the methods that involved the creation of a matrix of some kind. A few might think about other notions of positive-definiteness as defined by mathematicians, and I think they would also wonder what is meant. Maybe you just mean that precipitation observations are always non-negative?*

Changed to non-negative (twice)

*196-198: I have two comments on this new sentence.*

*First, ENSO (and other modes of internal variability that affect the region) are certainly relevant since it would be hard to think that the probability of extreme temperature, precipitation and FWI are all insensitive to the phases of these modes of variation.*

We define a covariate that describes the trend as well as possible. This should exclude ENSO, as ENSO has no trend, hence we use a low-pass filter that excludes the variability due to ENSO that should not be included in the trend. Added 'not relevant for the trend'.

*Second, what kind of externally forced variability are you talking about here? If the interest is in anthropogenic forcing, which has a largely monotonic response over time, wouldn't a longer time average that better isolates the signal by removing more of the high frequency variation (mostly internal, but perhaps also due to episodic volcanic forcing) be better?*

It is not really possible to include volcanic forcing and exclude the internal ENSO variability by temporal filtering as they have the same timescale (Lehner et al., 2016, *Geophysical Res. Letters*). The length of the filter is a trade-off. In principle a longer time scale would be better, but as we mention the event that is being attributed is usually at the end of the unfiltered time series or just beyond it, so that a long filter becomes increasingly ill-defined for that part of the time series. A 4-yr running mean requires an estimate of the next two years, which can be assumed to be similar to the current year (e.g., van den Dool and Barnston, 1996), but a 21-yr running mean or LOESS filter requires estimates for the next 10 years, over which GMST will continue to rise.

*201: Here and throughout, make it clear that all probabilities are ESTIMATED, and thus subject to uncertainty.*

Certainly, added this three times.

*204-212: There is something here that I seem to be missing. For example, if the GEV location parameter is a linear function of T', then how can it also be an exponential function of T', as in equation (3)?*

It is a linear function for temperature and an exponential one for the non-negative parameters precipitation, FWI and MSR. Clarified the text.

*251: The responses to my previous comments promised that this shorthand ($\chi2/dof$) would be clarified – but that appears not to have been done (I could not find a definition of the statistic that is referred to here in either the main text nor the supplement.*

We have added the definition and a clearer explanation how it is used.

*271: How did you determine that the factor was "at least two"?*

This is the lower bound of the model synthesis, which is unfortunately not shown in Figs. S6. We have added this bar to the figures S6 and S7 and mentioned it in the text. We rounded the lower value of the PR in the model world from nominally 1.84 to two in order not to suggest too much accuracy.

*Note also that there is an important nuance of the communication of probability ratios that the paper seems to be sloppy about. When PR=2, p1 = 2p0, and thus the event is estimated to be 2 times AS likely in the current climate as in the counterfactual climate, or equivalently, 1 times MORE likely. If you say "at least two times more likely", then my interpretation would have to be that PR ≥ 3. This vagueness of interpretation seems to crop up in several places.*

We assume the reader knows that the PR is declared REAL and not INTEGER.

*330-332: This convoluted sentence will be very hard for readers to understand. I suggest breaking this up into two or three sentences that explain in bite-size chunks the differences between the analyses of the observations and that of the climate model output (rather than trying to make the readers swallow the entire sandwich at one go).*

Thanks, done.

*Also, here and elsewhere, be careful with the word "models". For example, on line 330, the text reads "the models use as covariate the model GMST". Evidently the first use of "model(s)" refers to the statistical models used in this study, whereas the second use of "model", only 5 words later, refers to climate models. That implicit distinction will be clear to some readers, but many other readers will be puzzled.*

We attempted not to use the word 'model' to refer to statistical models but only to climate models. In this sentence that was the intention also, reformulated to make this clear.

*Figure 3 caption: Describe the grey shaded band that is shown in the two right-hand panels.*

Done: ''.. and the grey bands the 95\% uncertainty ranges.'

*Figure 5 caption: Describe the shaded bands that are shown!*

Done: 'The bars denote the 95% uncertainty ranges'.

*343-344: The statement that "all model dispersion and shape parameters lie within the large observational uncertainties" seems a bit of a stretch. There is some overlap in the spread of the dispersion parameters from the climate models with that from the observations for 3 of the 4 climate models, but that overlap doesn't necessarily mean that for those climate models, one would accept the null hypothesis that the dispersion parameter from the climate model is the same as the dispersion parameter from the observations.*

We agree that the correct way to do this would be to formulate the null hypothesis that the values are compatible and do the straightforward statistical analysis from there. However, due

to a miscommunication within the group the procedure has been simplified in our published protocol (Philip et al, 2020) to whether the ranges overlap, which gives a slightly more lenient test. We made the choice to follow the published protocol here and update it in the near future.

*353: Results from ERA5 are confounded with the impact of low-frequency internal variability during the single 41-year realization of the observed climate that ERA5 attempts to reconstruct…*

This is exactly the point we try to make and why we also did the long-term drought analysis using a much larger ensemble (cf Goss et al, ERL, 2020 on the Californian fires). We added a sentence re-emphasising that here as well: ''Note that the ERA5 value is probably biased high as the positive contribution of trend towards a drier climate over 1979--2019 is not present over 1900--2019, see sections S2 and 5.6.'

*355-356: Here is another example of ambiguity in the interpretation of the probability ratio. Judging from the figures, I think you should be saying seven rather than eight times MORE likely, and for the lower bound, perhaps only 2.5 times MORE likely.*

The unrounded numbers are 7.96 and 4.31.

*367: This might not be as much of a climate model deficiency as implied here (which is what "underestimation" implies). As very briefly discussed in Section 6, and further elaborated in Section S3, apparently the IOD and SAM did play a role.*

Using our class-based event definition, the variability of the SAM and IOD is included in all these results, so does not explain the discrepancies. It could be that the discrepancies come from problems representing the IOD and SAM, but we show that this is unlikely to be the case in the supplementary material due to the small contribution of these modes to the historical variability.